# Fungal community assembly in drought-stressed sorghum shows stochasticity, selection, and universal ecological dynamics

Cheng Gao [1], Liliam Montoya [1], Ling Xu [1,2], Mary Madera[1], Joy Hollingsworth[3], Elizabeth Purdom [4], Vasanth Singan [5], John Vogel[5], Robert B. Hutmacher[6], Jeffery A. Dahlberg[3], Devin Coleman-Derr[1,2], Peggy G. Lemaux [1] & John W. Taylor [1]*

Community assembly of crop-associated fungi is thought to be strongly influenced by deterministic selection exerted by the plant host, rather than stochastic processes. Here we use a simple, sorghum system with abundant sampling to show that stochastic forces (drift or stochastic dispersal) act on fungal community assembly in leaves and roots early in host development and when sorghum is drought stressed, conditions when mycobiomes are small. Unexpectedly, we find no signal for stochasticity when drought stress is relieved, likely due to renewed selection by the host. In our experimental system, the host compartment exerts the strongest effects on mycobiome assembly, followed by the timing of plant development and lastly by plant genotype. Using a dissimilarity-overlap approach, we find a universality in the forces of community assembly of the mycobiomes of the different sorghum compartments and in functional guilds of fungi.

[1] Department of Plant and Microbial Biology, University of California, Berkeley, CA 94720, USA. [2] Plant Gene Expression Center, US Department of Agriculture-Agricultural Research Service, Albany, CA 94710, USA. [3] University of California Kearney Agricultural Research & Extension Center, Parlier, CA 93648, USA. [4] Statistics Department, University of California, Berkeley, CA 94720, USA. [5] Department of Energy Joint Genome Institute, 1 Cyclotron Rd., Berkeley, CA 94720, USA. [6] University of California West Side Research & Extension Center, UC Davis Department of Plant Sciences, Five Points, CA 93624, USA. *email: jtaylor@berkeley.edu

I n nature, the intimate, symbiotic association of fungi with living plants is well appreciated, whether as mycorrhizal partners or parasites[1], as is their role as drivers of plant community structure[2–4]. Equally well appreciated is their association with dead plants where their role in ecosystem carbon and nitrogen cycling has been highlighted[5–7]. In contrast, studies of the total communities of fungi associated with plants are few and far fewer than those of total communities of bacteria[1], despite the fact that, in terrestrial environments, fungi account for more biomass than bacteria[8,9]. Here we tackle this gap in knowledge of fungal communities associated with plants by adding the fungal mycobiome to studies of the bacterial microbiome[10,11]. We use an approach that accounts for the variation in plant compartment (leaf, root, rhizosphere and soil), plant development (from seedling emergence to grain maturation), physical environment (irrigation and drought), and host genotype (drought response to either retain or suppress photosynthesis) as summarized in Supplementary Data 1 and 2. We chose drought as the key environmental variable for our study because it will be a defining feature of this century[12,13], and will not only directly affect plants and their fungal communities, but also indirectly affect plants through changes in the fungal community[1,14]. Positive, indirect effects of fungal communities on plants could best be harnessed by modern agriculture[15], if the mechanisms of community assembly were better understood.

The assembly of communities rests on the activity of the four processes that influence constituent species: selection, drift, diversification, and dispersal[16]. Selection is wholly deterministic and drift is wholly stochastic[17], but both dispersal and diversification have stochastic and deterministic components[17]. Of the four processes, drift, the stochastic extinction caused by random species abundance fluctuation[18], is the most difficult to demonstrate because one must first rule out the other three processes, two of which have stochastic components, although one of these —the evolutionary process of diversification—can reasonably be ignored for fungi over a period as short as a season.

Ecological drift has been estimated in ecological model fitting as the 'unexplained' compositional variation, or has been estimated from empirical data as the dispersion of beta diversity (an approach also termed compositional variance)[18]. However, neither of these methods of estimation necessarily represent the consequences of ecological drift due to possible undermeasurement of the processes of selection, dispersal and diversification[18]. Drift is thought to most strongly influence community assembly when (i) communities are small or when (ii) they have recently been released from selection imposed by a stress such as drought[19,20]; However, due to the difficulty of detecting drift, even in these two situations its demonstration in nature remains rare[17,21–23].

Adding our study to those already in existence raises the question of universal features of mycobiome assembly and temporal change[24], in particular, are the underlying ecological dynamics of microbiomes universal across all communities or unique to individual communities? To address this point, we employed the recently developed, dissimilarity-overlap curve (DOC)[24]. The DOC approach has been applied to recent studies of human-associated bacterial microbiomes, finding universal ecological dynamics across all communities[24,25]. Although many attempts have been made to assess the universality of microbial community assembly, assessment of universality by the DOC method has been applied to just two types of fungal mycobiomes, both of which are arbuscular mycorrhizal (AM) fungal communities. These studies found both universal and unique ecological dynamics depending on ecosystem type (natural v. agricultural) and phosphorus availability (low v. high)[26,27].

To thoroughly address the mechanism of community assembly, we combined a simplified environment with one crop species, sorghum [*Sorghum bicolor* (L.) Moench] grown in homogenous soil with intensive sampling from thrice-replicated plots for all possible combinations of two genotypes, four plant compartments, three water treatments, and 17, weekly time points, giving a total of 1026 samples (Fig. 1). The discovery that the effect of host genotype is negligible allowed us to use six replicates in most of our analyses. Our study complements recent reports from the same agricultural system on the bacterial microbiome (756 samples)[10], the AM fungal mycobiome (312 samples)[11], and the sorghum transcriptome[28].

We test two hypotheses concerning fungal communities of sorghum plants; $H_1$, that drift will be important when fungal communities are small, as expected early in the development of sorghum plants when microbes should be rare on newly formed roots and leaves[29], and, $H_2$, that drift will be important after drought stress is relieved by restoring irrigation to sorghum plants that had been deprived of water before flowering. To test these hypotheses, we characterized fungal communities associated with sorghum plants growing in the field from seedling emergence to plant senescence under conditions of regular irrigation and also when stress imposed by pre-flowering drought was relieved by resuming irrigation.

Here, we present results that support, as hypothesized ($H_1$), a significant role for drift early in fungal community assembly by observing a negative correlation between the strength of stochasticity on one hand and community size on the other, and note that we cannot rule out some role for the stochastic aspect of colonization. Conversely, our hypothesis, $H_2$, that release from pre-flowering drought stress would enhance the importance of stochasticity was falsified, likely due to the strong selection exerted by the plant host in the sorghum system. Moreover, where DOC analysis shows a reduced dissimilarity among fungal communities as the fraction of shared taxa increases and where this reduced dissimilarity is supported by the bulk of all possible pairwise comparisons, we find that the underlying ecological dynamics are largely universal across mycobiomes of the different sorghum compartments and also universal across functional guilds of fungi, which include diffusion-feeding yeasts as well as filamentous plant pathogens, AM fungi, and saprotrophs.

## Results

**Sorghum mycobiome structure**. First, we characterized overall fungal communities in an agricultural field of sorghum from samples taken before sorghum seed was sown and over the ensuing seventeen-week period from seedling emergence to grain maturation (Fig. 1). We characterized fungal operational taxonomic units (OTUs) from samples of soil, rhizosphere, roots and leaves taken from a field that had never previously experienced sorghum and most recently had been planted to oats. In total, we recognized 1070 OTUs (Supplementary Data 3) detected from DNA sequence of fungal internal transcribed spacer 2 (ITS2) amplified by dual-barcoded, fungal specific 5.8SFun and ITS4Fun primers[30], and sequenced by Illumina Miseq (34 541 758 reads; Supplementary Figs. 1–4; Supplementary Software 1).

**Host dimensions of time, compartment, and genotype**. To examine the dimensions in which the host shapes the mycobiome, we explored the host effect in terms of compartment, time and plant genotype. As described in the statistical methods section, we analyzed the data both as counts (dataset 1) and compositionally (dataset 2), to both recognize the compositional nature of high-throughput sequencing (HTS) microbiome data[31] and to permit comparisons of our results with previous studies that treat HTS data as counts. By either method of analysis, the largest effect on the total mycobiome was exerted by the

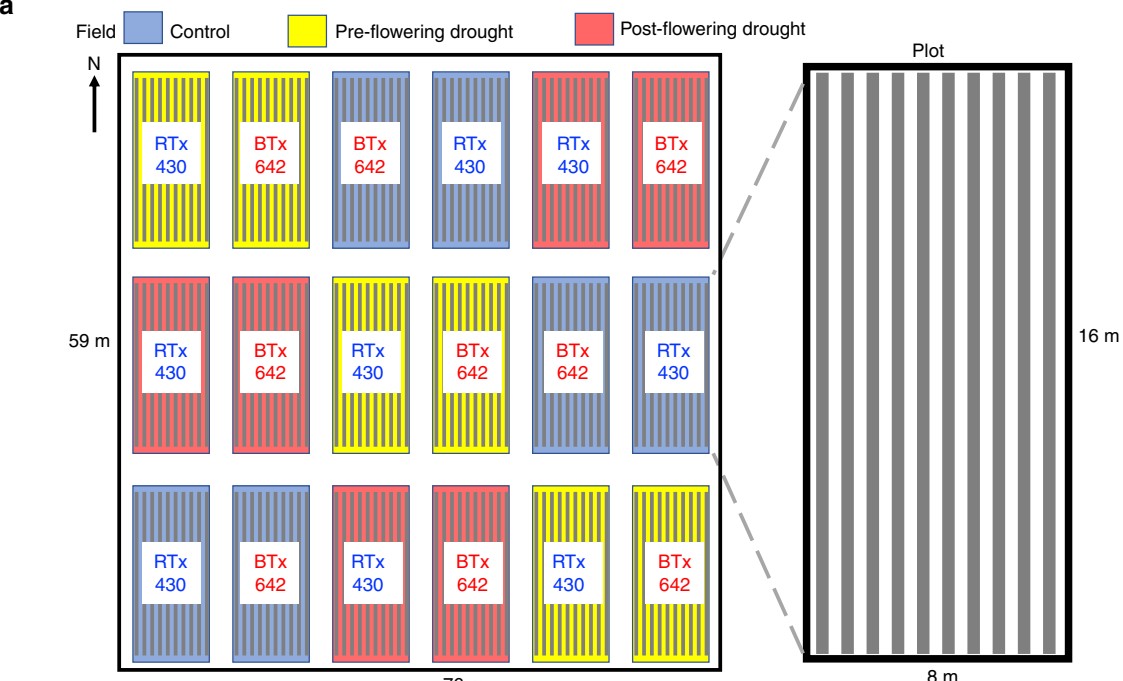

**Fig. 1 Experimental design. a** Field layout of the 18 plots (16 × 8 m² each) in a random block design of three treatments (control, pre-flowering drought and post-flowering drought) and two sorghum cultivars (RTx430 and BTx642) with three replicates in a 76 × 59 m² field. The discovery that the effect of host genotype is negligible allowed us to use six replicates in most of our analyses. Each plot consisted of ten, 16 m long rows, each containing approximately 200 plants spaced 8 cm apart. **b** Irrigation scheme and sampling strategy. Irrigation for all treatments was identical until week 3 when pre-flowering drought was initiated. Irrigation for control and post-flowering drought was identical until week 10 when post-flowering drought was initiated. Soil samples were collected 20 cm from the plant stem to a depth of 15 cm with a soil corer, while leaf, rhizosphere and root samples were collected from plants extracted by shovel to a depth of approximately 20 cm.

compartments of: soil, rhizosphere, root and leaf (dataset 1: $R^2 =$ 0.421***; dataset 2: $R^2 = 0.371$***), followed by time (weeks 1–17 of plant development) (dataset 1: $R^2 = 0.107$***; dataset 2: $R^2 = 0.055$***), and lastly by host genotype (cultivars BTx642 and RTx430) (dataset 1: $R^2 = 0.002$***; dataset 2: $R^2 = 0.002$*; Fig. 2a; Supplementary Fig. 5; Supplementary Data 4). This result is different from that seen with AM fungi, alone, where temporal change had the greatest impact ($R^2 = 0.438$***), followed by the host compartment (soil, rhizosphere and root) ($R^2 = 0.094$***) and, lastly, by the host genetical dimension[11]. This difference between the total sorghum mycobiome and that of AM fungi, alone, may be due to the ability of AM fungi to simultaneously occupy three distinct compartments (compartments of root, rhizosphere and soil)[32], whereas most other fungi can effectively occupy only one of the four compartments.

The first effect of the sorghum host on fungal community function became apparent shortly after planting as evidenced by a decline in the abundance of saprobes relative to the abundance of

mycorrhizae, plant pathogens, endophytes, and yeasts (Supplementary Figs. 2, 4, 6). Subsequent temporal change in community composition, measured using a Mantel test of Bray–Curtis community dissimilarity, was seen throughout the ensuing 17 weeks and was strong in leaf, root and rhizosphere and weak in soil (Fig. 3a; Supplementary Fig. 7). Visual confirmation of these trends was seen in comparison of OTUs (Fig. 2b) and functional guilds (Supplementary Fig. 2), again strongly in leaves (where a signal for temporal change can be detected even at the phylum level; Supplementary Fig. 3), roots and rhizosphere, and weakly in soil.

Having observed a host effect on fungal community composition, we wondered if the strength of host-driven community turnover was constant over time. Given that the developing sorghum plant is involved in driving these temporal changes in community composition, the length of time that it takes for this effect to reach a temporally stable community can be used to gauge the strength and timing of sorghum's influence (Fig. 3b;

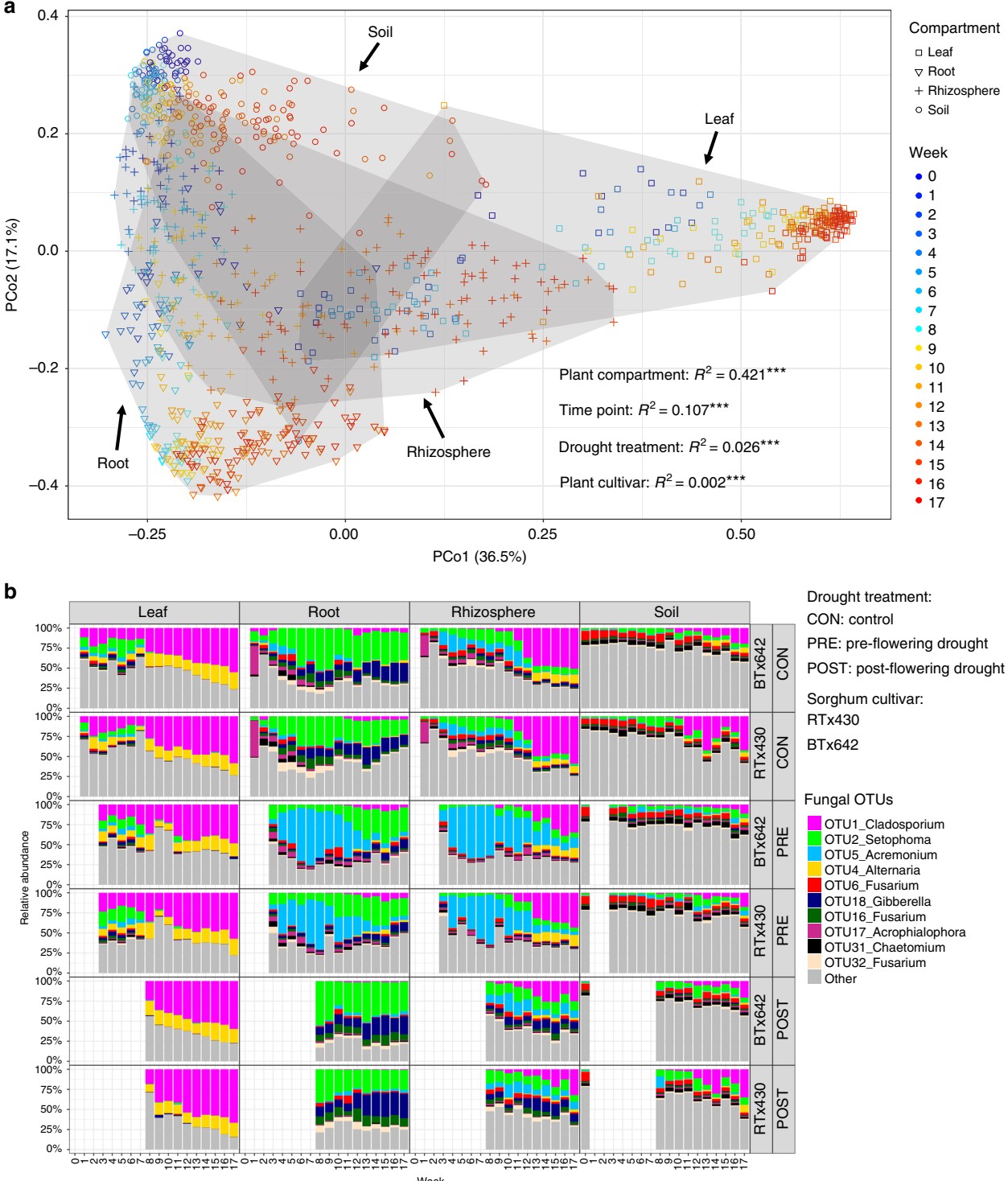

**Fig. 2 Structure of sorghum mycobiome. a** Principal coordinate (PCo) analysis of fungal community Bray–Curtis dissimilarity with permutational analysis of variance (PERM ANOVA) showing significant association of fungal community composition with, in order of importance, compartment, time point (TP), drought treatment and sorghum cultivar (***$P$ < 0.001). Note that results of principal component (PC) analysis of Aitchison distance is presented in Supplementary Fig. 5. **b** Temporal change in relative abundance of fungal operational taxonomic units (OTUs) at each TP in the four compartments, three treatments and two sorghum cultivars. To avoid redundancy, pre-flowering treatment sampling began at the third week and post-flowering sampling began at the 8th week. Source data are provided as a Source Data file.

Supplementary Figs. 8, 9). Here, turnover of community composition was depicted by Simpson dissimilarity, a metric that is independent from species richness variance[33]. With this approach, we found that each compartment had a different period for maximum turnover of each fungal community. In

leaves, compositional variance turns over markedly until the ninth week, after which it ceases to change (Fig. 3b; Supplementary Figs. 8, 9). In roots the turnover of compositional variance is more predictable than in leaves and turns over until the 12th week followed by stability (Fig. 3b; Supplementary Figs. 8, 9). In

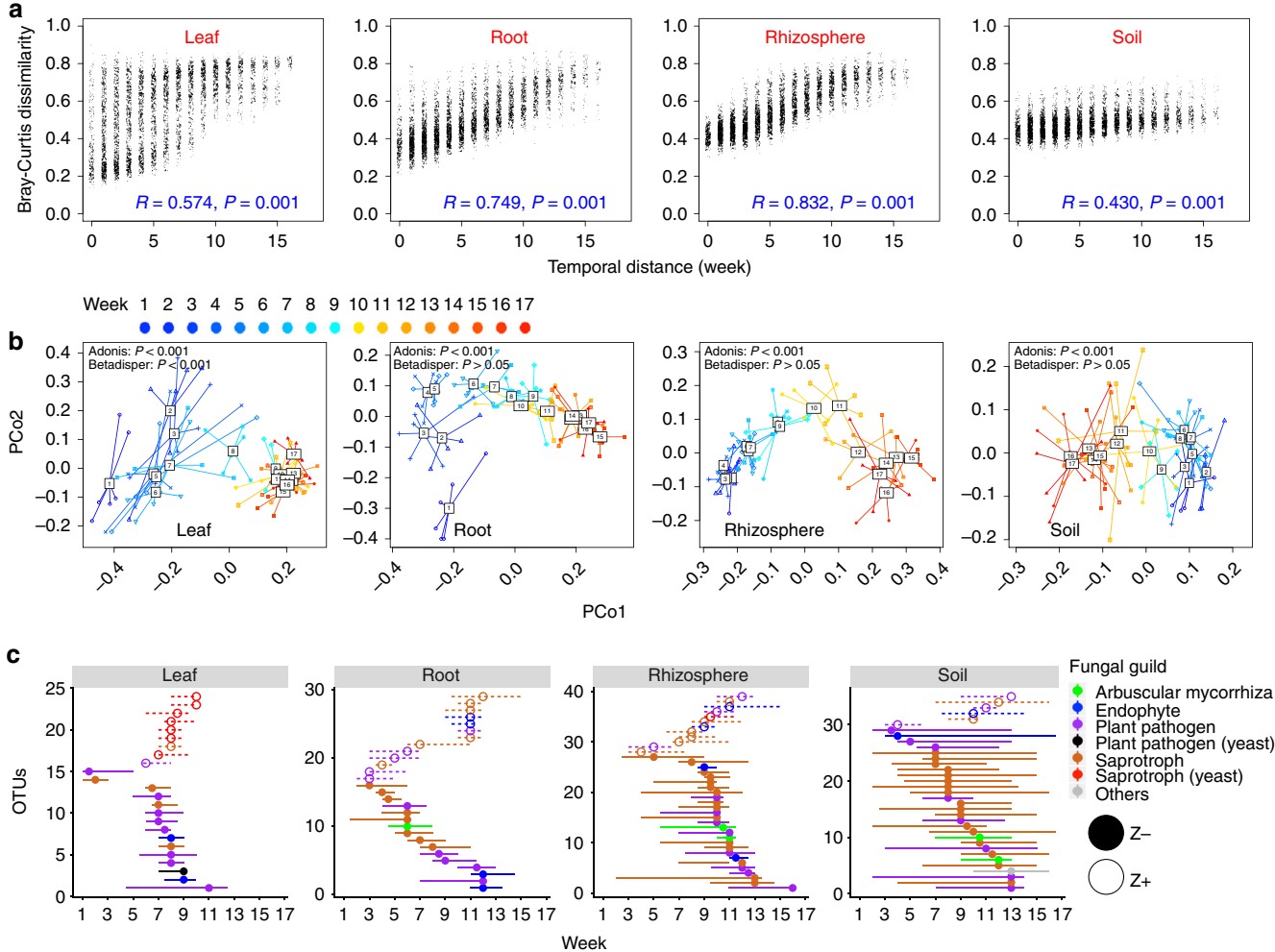

**Fig. 3 Temporal dynamics of the sorghum mycobiome. a** Succession is strong in leaf, root and rhizosphere, and weak in soil, based on Mantel testing of the correlation between temporal distance and Bray–Curtis community dissimilarity. **b** Turnover of fungal community composition demonstrated by Simpson dissimilarity among replicate plots and sampling times. Note strong fungal compositional turnover among weeks 1–9 in leaves followed by significantly less turnover for weeks 9–17. Similar strong turnover is seen among weeks 1–12 in roots, weeks 7–13 in rhizosphere, and weeks 9–13 in soil, but none is significantly different from other time points in these compartments. The Simpson metric differs from the Bray–Curtis and Jaccard metrics in that it is free from richness variance, as seen in our analysis of the sorghum root mycobiome (Supplementary Figs. 11–12) and our re-analysis (Supplementary Fig. 13) of the rice root microbiome of Edwards et al.[35]. **c** Temporal change of individual OTU abundance shown by threshold indicator taxa analyses (TITAN). For each OTU, functional guild is noted by color and filled symbols show declining abundance (z−), and open symbols show increasing abundance (z+). Genus names of the OTUs in each guild can be found in Supplementary Fig. 10. Source data are provided as a Source Data file.

contrast to leaves and roots, the situation in rhizosphere and soil is different with little turnover from weeks 1 through 7 or 9, and again from weeks 13 through 17, the two periods of relative stability bridged by one of turnover (Fig. 3b; Supplementary Figs. 8, 9).

We also found evidence for turnover using a second analytical approach, threshold indicator taxa analysis (TITAN), which is based on increases (z+) or decreases (z−) in the abundance of indicator fungal OTUs[34]. With the leaf mycobiome, our results showed that indicator fungal OTUs showed both z+ and z− beginning in the 7th to 8th weeks which largely ceased by the 9th week (Fig. 3c; Supplementary Fig. 10). In roots, indicator fungal OTUs showed significant z+ and z− from the third week until the 12th week, while in the rhizosphere obvious temporal turnover did not begin until the 8th week and continued until the 12th week (Fig. 3c; Supplementary Fig. 10). In published analyses of bacterial and AM fungal communities of these same sorghum plants, the five-week delay in the initiation of turnover of rhizosphere compared to roots was not seen and, instead, the

bacterial microbiome of both root and rhizosphere stabilized after the 6th week[10], and AM fungal communities of both root and rhizosphere continued to turn over from the 1st week to the 17th week[11]. Turning to studies of bacteria in other systems, compared to our fungal results, in a bacterial community associated with rice, turnover started early (1st week) and stabilized after 8 or 9 weeks when vegetative plant growth had ceased, again with root and rhizosphere following the same temporal pattern[35]. Why turnover should cease earlier for bacterial microbiomes than fungal mycobiomes is not clear, although the better taxonomic precision for fungal ITS compared to bacterial 16S may play a role.

The level of compositional variance among replicated samples of microbial communities has been reported to change over time. For example, bacterial microbiomes associated with plants have exhibited higher compositional variance early in the plant life cycle than in later stages in both rice roots and *Arabidopsis* leaves[35,36], and similar results have been published for fungi in crop roots, albeit with sampling intervals longer than one week

(10–55 days)[37,38]. However, compositional variance in such studies, whether measured by the Bray–Curtis (incorporating abundance data) or Jaccard (limited to presence/absence data) metrics, can be confounded by variation in richness[33]. As noted above, this influence can be avoided by using the Simpson metric[33]. With our sorghum data, Bray–Curtis and Jaccard metrics showed similar results: higher compositional variance (assessed from pairwise dissimilarity values) in early than in later samples for leaves and roots (Supplementary Figs. 11, 12). However, when the variance due to richness was removed, significantly higher compositional variance (Simpson dissimilarity) was found only at early time points in leaves, and not in roots (Fig. 3b; Supplementary Fig. 8). Using this approach, we also reexamined the result that early bacterial communities of rice roots are more variable than later communities[35], finding that the higher compositional variance was no longer significant for early roots when richness variance was removed (Supplementary Fig. 13). Finding higher compositional variance in emerging leaves than emerging roots may reflect the density of microbes surrounding each plant organ, which must be vastly greater for roots.

The strong effect of the host compartment seen in our sorghum data (42.1%) was also seen in two studies of ectomycorrhizal (EM) *Populus* in which compartment (again; root, stem, leaf, and soil) accounted for 58%[39] or 24%[40] of compositional variation. While acknowledging that comparison among studies is never simple, comparison of our results with those two studies may help explain the more than twofold difference in *Populus* compartment effects. The strong compartment effects seen in our study and the first of the *Populus* studies[39] rely, in part, on an abundance of yeasts in leaves, which was smaller in the second *Populus* study[40] (Supplementary Fig. 2). The gap between the very strong compartment effect in the first *Populus* study and our results may be explained by the low overlap of EM fungal OTUs in *Populus* roots and soil[39,41] compared to the greater overlap of AM fungi in these two compartments in sorghum[11].

The effect of the two host genotypes (post-flowering drought tolerant cultivar BTx642 and pre-flowering drought resistant cultivar RTx430), although weaker than that of compartment or time, was still significant in root ($R^2 = 0.021$, $P < 0.001$) but not in leaf ($R^2 = 0.004$, $P = 0.284$) (Supplementary Fig. 14). This host genotype effect in sorghum roots was due to the presence of two pathogens (OTU19_*Sarocladium* and OTU20_*Monosporascus*) and one saprotroph (OTU34_*Achroiostachys*), all of which were significantly more abundant in the roots of sorghum cultivar BTx642 than in the roots of cultivar RTx430 (Supplementary Fig. 14), plus the presence of another pathogen, (OTU207_*Magnaporthiopsis*), whose pattern of abundance was the reverse, that is, significantly more abundant in the roots of cultivar RTx430 than in cultivar BTx642 (Supplementary Fig. 14). Prior work has frequently demonstrated the effects of host plant genotype for root communities of EM fungi and we believe that our results now extend the effect to communities of plant pathogenic and saprotrophic fungi[42,43] (Supplementary Data 1). Although we found no effect of host genotype on leaf fungal communities when considering all 17 time points as a whole ($R^2 = 0.004$, $P = 0.284$), we did find a significant effect of host genotype on leaf fungal communities for the subset of weeks 10–17 ($R^2 = 0.038$, $P = 0.002$; Supplementary Fig. 14A). This significant effect was due to the differential presence of yeasts (Supplementary Fig. 14B), a result in line with other recent findings[21,44] (Supplementary Data 1). Perhaps differences between cultivars in diffusible substrates differentially influence yeast growth in the period following sorghum flowering. In contrast to our results, a study of bacterial microbiomes in leaves and roots of Brassicacae found evidence for host genetic control of the leaf but not the root

microbiome[45] (Supplementary Data 2). Whether we should expect to see similar results when both the type of microbe and the host are different is a question that cannot be answered by the few existing studies and, obviously, more are needed.

**Testing H1 – Stochasticity and fungal community size.** As noted in the introduction, four ecological forces shape fungal community composition, selection, dispersal, evolutionary divergence, and drift, only one of which is wholly deterministic, selection[16]. Also as mentioned at the outset, one of the forces with a stochastic component, divergence, can be ignored for fungi over a 17-week period[46], leaving two forces with stochastic components, wholly stochastic drift and partly stochastic dispersal. Thus, should we detect stochasticity, it could be attributed to drift or stochastic dispersal in the period of initial colonization. Whereas stochastic aspects of dispersal, in this case the initial colonization, are expected to be unaffected by community size, drift is clearly enhanced in small communities when individuals are prone to extinction by chance[18] and the probability of extinction is expected to increase as the community size shrinks[47–50]. In practice, this rigorous test of drift is hampered by (1) the difficulty to capture microbial communities small enough to detect the action of ecological drift, and (2) the lack, until recently, of statistical tools that can retrieve the stochastic component of compositional variance.

In this study, we captured small fungal communities in early leaves and roots by weekly sampling beginning with seedling emergence as evidenced by estimating community size using three different methods, all of which were strongly correlated ($R > 0.7$, $P \ll 0.001$; Supplementary Fig. 15): (i) the percentage of fungal reads found in PCR amplifications of ITS2 from fungal and host DNA, (ii) the fungal abundance as assessed by real-time PCR amplification of rDNA small subunit (SSU) and (iii) the percentage of fungal reads found in the transcriptomes of sorghum leaves and roots (Supplementary Figs. 16–18). As implied above, statistical tools have recently been developed that can retrieve the stochastic component of compositional variance. These methods compare the matrix of dissimilarities calculated for observed communities to those calculated from communities randomly assembled by sampling, with replacement, from the pool of all observed taxa. The proportion of the matrix of dissimilarity calculated from observed data with respect to the distribution of those calculated from multiple rounds of resampling determines the probability that the null hypothesis of stochasticity can be falsified. The two tools that we employed are the beta Nearest Taxon Index (βNTI) and the Raup-Crick Index (RCI), which are unable to reject stochasticity when the |βNTI| < 2 (reflecting stochastic turnover in phylogenetic composition[51,52]) and when the |RCI| < 0.95 (reflecting stochastic turnover in species composition[53,54]). The stochasticity indicated by |βNTI| < 2 and |RCI| < 0.95 is obtained when communities are not dominated by either the two dimensions of dispersal (homogenous dispersal, dispersal limitation) or the two dimensions of selection (variable selection, homogeneous selection)[55]. Thus, the stochasticity detected by |RCI| < 0.95 and |βNTI| < 2 is not likely to be explainable by hidden variables. When the null hypothesis can be rejected, observations smaller than the null estimations indicate underdispersion of phylogenetic (βNTI < −2) and species (RCI < −0.95) composition in which community pairs are homogenous, and observations larger than the null estimations indicate overdispersion of phylogenetic (βNTI > 2) and species (RCI > 0.95) composition in which community pairs are heterogenous[51,53,54]. The phylogeny used to calculate βNTI from ITS2 OTUs relied on a fungal phylogeny based on 18S + 28S rDNA sequences (taxonomy_to_tree.pl script of Tedersoo et al.[56], Supplementary Fig. 19).

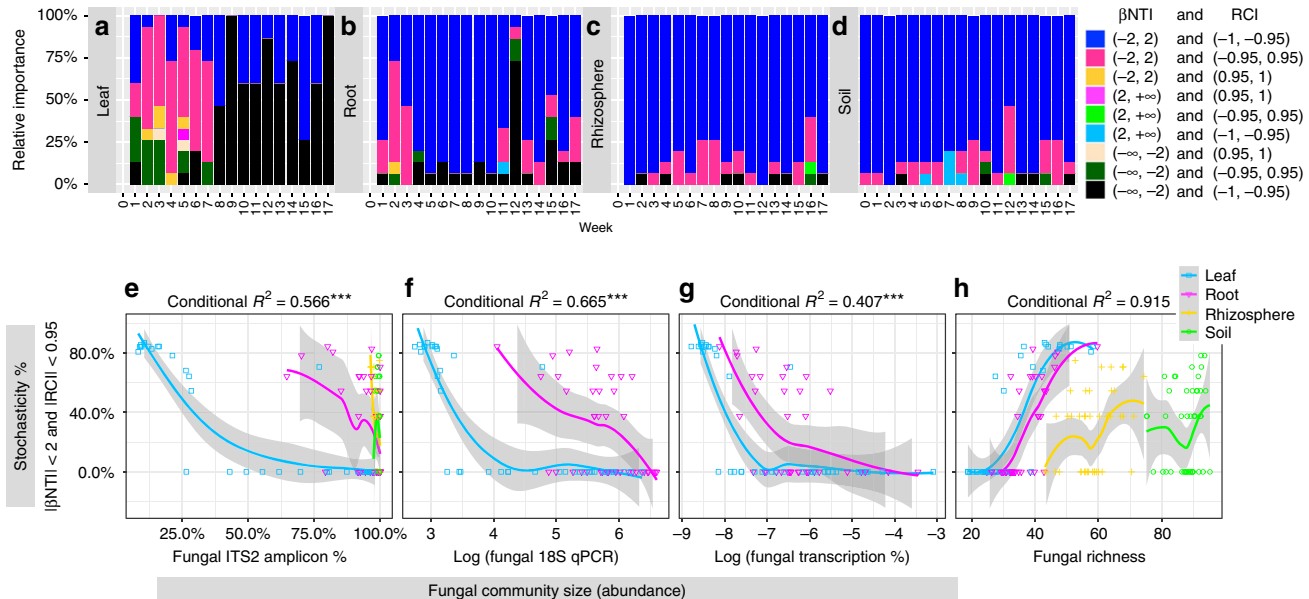

**Fig. 4 Stochasticity and ecological drift in community assembly. a–d** Stochasticity is detected from values of two indices, β-Nearest Taxon Index (βNTI) and Raup-Crick Index (RCI). Where |βNTI| < 2 and |RCI| < 0.95 (red bars), compositional variance is most likely due to stochasticity, which in this range is independent from dispersal and selection. Note that stochasticity is dominant in early time points in leaves and roots. Selection is favored as the force for community assembly where βNTI ≤ −2 and RCI ≤ −0.95 (black). Here, communities are more clustered than expected by chance both phylogenetically and ecologically, likely due to homogenous selection. Where |βNTI| < 2 and RCI ≤ −0.95 (blue), communities are ecologically more clustered than by expected by chance but phylogenetically stochastic. In this case, community composition can be a result of either homogenous selection or homogenous dispersal and, of the two choices, homogenous selection is preferred due to the observed strong selection by host compartment and time, as demonstrated in Fig. 2A and Supplementary Data 4. (**e–g**) Stochasticity and community size. Ecological drift can be strengthened when the stochastic component of compositional variance is negatively correlated with community size. Here, we demonstrate this correlation when community size is estimated in three different ways: **e** fungal rDNA, internal transcribed spacer (ITS2) reads as a percentage of total ITS2 reads amplified from fungal plus sorghum host DNA, **f** fungal 18S rDNA amount assessed by real-time or quantitative PCR amplification and **g** fungal RNA as a percentage of fungal plus plant reads found in the transcriptomes of sorghum leaves and roots. **h** Stochasticity and richness. Ecological drift is suggested by a positive correlation between fungal richness and the stochastic component of compositional variance, as shown here. Source data are provided as a Source Data file.

Early in the season, when fungi were shown to be rare on newly emerged leaves (weeks 1–7) and roots (weeks 1–3), we found that stochasticity shaped the fungal community as shown when |βNTI| < 2 and |RCI| < 0.95 (Fig. 4a–d). It is expected that stochasticity will expand community compositional variance[16] and, consistent with this expectation, we found that community compositional variances measured by Bray–Curtis, Jaccard, turnover and Fst metrics were all high for early leaves, and at least partially high for early roots (Supplementary Figs. 8, 11, 12, 20).

Intuitively, stochastic community assembly would be suspected when one is confronted by many small communities, each containing just a few species that show little overlap with the others (resulting in a high combined richness). Quantitatively, having observed stochasticity in early leaves and roots, we ascribe a significant fraction of it to drift because the percentage of sample pairs showing |βNTI| < 2 and |RCI| < 0.95 is strongly negatively correlated with fungal community size as detected by amplicon-, qPCR- and transcriptome- based methods (Fig. 4e–g). Stochasticity can also be ascribed to ecological drift if it is positively correlated with richness when community sizes are small, because the probability of extinction is expected to increase when the total species pool is large relative to the local community size[57]. Additionally, a consequence of ecological drift is high beta diversity, which can lead to high, joint richness (Fig. 4h). Again, we find that the strength of stochasticity is positively correlated with mean fungal richness for the six replicates that constitute each of the 171 samples (week, compartment, treatment) (Fig. 4h).

Having ascribed a significant fraction of stochasticity to drift, can we rule out the action of stochastic dispersal in the colonization of emerging plants? Given the same level of stochastic dispersal, greater variability in change of species compositions is expected for smaller communities than larger communities. However, no clear evidence in support of stochastic colonization was found because the first leaf and root fungal communities were dominated in replicate samples by a single species, either OTU42 (*Actinomucor*) for leaves or OTU17 (*Acrophialophora*) for roots (Fig. 2b; Supplementary Fig. 3), and both of these OTUs were rare in soil sampled prior to planting (Supplementary Fig. 4A). However, despite the dominance of OTU42 and OTU17 in the first week, we cannot rule out the stochastic colonization of other OTUs, nor can we rule out any stochastic colonization that might occurred during the two weeks between planting and our first sampling (Fig. 1). Concerning events that occur prior to sampling, we did not detect any priority effects involving selection in early fungal communities because OTU42 and OTU17, although abundant in all replicates in the 1st week (TP01), were largely replaced by other fungi by the 2nd week (TP02) (Fig. 2b; Supplementary Fig. 3).

The ability of early colonizing fungi to produce large amounts of hydrolytic enzymes might help explain their selection as first colonizers of roots and leaves. The early colonizers, OTU42 (*Actinomucor*) and OTU17 (*Acrophialophora*), are known to produce large amounts of enzymes that hydrolyze the protein, lipid, starch, and xylan components of sorghum seeds[58]; i.e., protease, lipase and amylase from *Actinomucor* and xylanase

from *Acrophialophora*[59,60]. The ability to produce abundant hydrolytic enzymes suggests that these fungi are adapted to quickly mobilize nutrients released from seeds at germination, but not to persist once the nutrients are exhausted. In the case of one of these initially abundant leaf fungi, OTU42 (*Actinomucor*, Mucoromycota), its large phylogenetic distance from the Ascomycota and Basidiomycota fungi that dominate other time points (Supplementary Fig. 3) explains why leaf samples of the first week are substantially different from later samples in the phylogenetic-based ordination (Supplementary Fig. 21A).

Stochasticity is never prominent in rhizosphere or soil and, as noted above, in leaves it gives way at week 8 (and in roots at week 4) to other processes for community assembly. This decline in stochasticity is evidenced by the RCI dropping below −0.95 and approaching −1, and with βNTI < 2 (Fig. 4). RCI approaches its lower limit (−1) because replicate plots show significantly less stochasticity than expected by chance, indicating that the same processes (homogenous selection or homogenous dispersal) are affecting all plots[54]. Homogenous dispersal is unlikely to be the driver of community assembly because we find evidence of environmental selection due to the strong effects of host and drought on the mycobiome (Fig. 2a; Supplementary Data 4). Where environmental selection is observed, homogeneous dispersal cannot be strong because immigration cannot exceed emigration, that is, there is no mass effect[61]. Alternatively, direct evidence of homogenous selection is found for at least the subset of community pairs that show βNTI ≤ −2 (Fig. 4), a condition consistent with phylogenetic underdispersion that can only be caused by homogenous selection[54,62]. Therefore, it is most likely that homogenous selection is the process that displaces stochasticity and, therefore, the most plausible driver of community assembly as sorghum matures.

We found no evidence for heterogeneity (βNTI > 2 or RCI > 0.95), whether caused by variable selection or dispersal limitation, as expected by our finding of stochasticity and homogeneity[55]. In line with these results, we detected no substantial effect of geographic distance on fungal community dissimilarity as shown by the flat slope of the change in dissimilarity over distance in leaves, roots, rhizosphere, soil and air (Supplementary Figs. 22, 23). Although we detected no dispersal limitation for communities, we cannot rule out dispersal limitation of particular species, although these would not be numerous or abundant.

**Testing H₂—drought stress and the sorghum mycobiome**. Where agriculture does not involve irrigation, drought may occur and be relieved at any stage of the plant life-cycle[14]. Here, we both imposed stress in the form of pre-flowering drought and then removed it by providing irrigation, or imposed stress in the form of post-flowering drought, having provided water for the first half of the sorghum growth cycle (Fig. 1). Release from the stress of pre-flowering drought enabled us to use the sorghum mycobiome to test our second hypothesis (H₂) that, in semi-arid California, drought would impose selection on fungal community composition and that relief from stress would then favor stochastic effects on community assembly. Relief from stress has been shown to lead to a rise in compositional variance among replicate microbial communities, whether the stress was due to drought[63–65], salinity[62], pH[66], nutrient limitation[67,68], removal of perennial plant functional groups[69], or predation[70]. In light of our test of H₁, which was consistent with small populations promoting stochasticity, we also considered the possibility that drought might reduce population size and allow stochasticity to be important before release from drought as well as after.

Contrary to expectations, the hypothesis (H₂) that release from pre-flowering drought would favor stochasticity was not supported because stochasticity, as indicated when |βNTI| < 2 and |RCI| < 0.95, did not increase when drought was lifted at week 9 (Supplementary Fig. 24). However, the prediction that stochasticity would be favored under pre-flowering drought was supported in leaves where stochasticity as judged by |βNTI| < 2 and |RCI| < 0.95 was generally higher under pre-flowering drought as compared to control plots, with the strongest difference just before and shortly after drought was lifted at weeks 8–9 (Supplementary Fig. 24). Because fungal community size was smaller under pre-flowering drought than irrigated controls (Supplementary Figs. 16–18), these results are consistent with the results of the test of our first hypothesis (H₁), that ecological drift is favored when community size is small. Our rejected hypothesis, H₂, was based on the thought that drought would be a strong, deterministic factor in fungal community assembly, however our results show that plant compartment ($R^2 = 42.1\%$) and development ($R^2 = 10.7\%$) exert much stronger effects on fungal communities than drought ($R^2 = 2.6\%$) (Fig. 2a; Supplementary Data 4). Given the dominant effect of plant compartment and development, our results can be explained by pre-flowering drought lessening the plant effect and, thereby, leading to smaller fungal populations and rising stochasticity. Similarly, drought relief would restore the plant-driven selection to pre-drought levels, obscuring any stochasticity (Supplementary Fig. 24). In line with these explanations, pre-flowering drought also delayed the development of sorghum in terms of both flowering time and biomass accumulation[10], as well as delaying dynamics of leaf fungal community size and abundance of individual fungal species, principally yeasts (Supplementary Figs. 2, 25).

Our test of H₂ necessarily focused on pre-flowering drought because its stress could be relieved, unlike that of post-flowering drought. To compare the effects of pre- and post-flowering drought on fungal community composition over time, we turned to Random Forest (RF) models. Given that the key difference was most likely to be temporal, we needed to identify the most important age-discriminant fungal OTUs (Supplementary Fig. 26) to use them to develop a more accurate sparse RF model, i.e., the trained model. To identify the most important age-discriminant fungal OTUs and use them to develop a trained model, we used a random subset of 50% control samples to construct a full RF model that treated sampling age as the response variable and all fungal OTUs as independent variables. Subsequently, the trained model was used to predict the ages of both drought samples and a second subset of 50% control samples, and the discrepancy between predicted ages of control and drought samples was then used to assess the extent of the drought effect. For fungi, similar effects of pre- and post-flowering drought were seen in all four compartments (leaf, root, rhizosphere, and soil) because the discrepancies between sampling ages and the predicted ages for pre-flowering drought communities were not larger than that predicted for post-flowering drought (Fig. 5e). Neither was a stronger effect of pre- than post-flowering drought seen by the results of permutational analysis of variance (PERMANOVA) (Fig. 5a, b). Our fungal results differ from those of bacterial microbiomes characterized from the same DNA samples used for our study, which showed that pre-flowering drought exerted a stronger effect on bacterial community composition than that post-flowering drought[10].

Although similar effects on fungal communities were seen for both pre- and post- flowering drought, we found asymmetrical changes of fungal community composition in roots and rhizosphere subjected to the two types of drought (Supplementary Fig. 27). This result is unlike the situation in bacteria, where

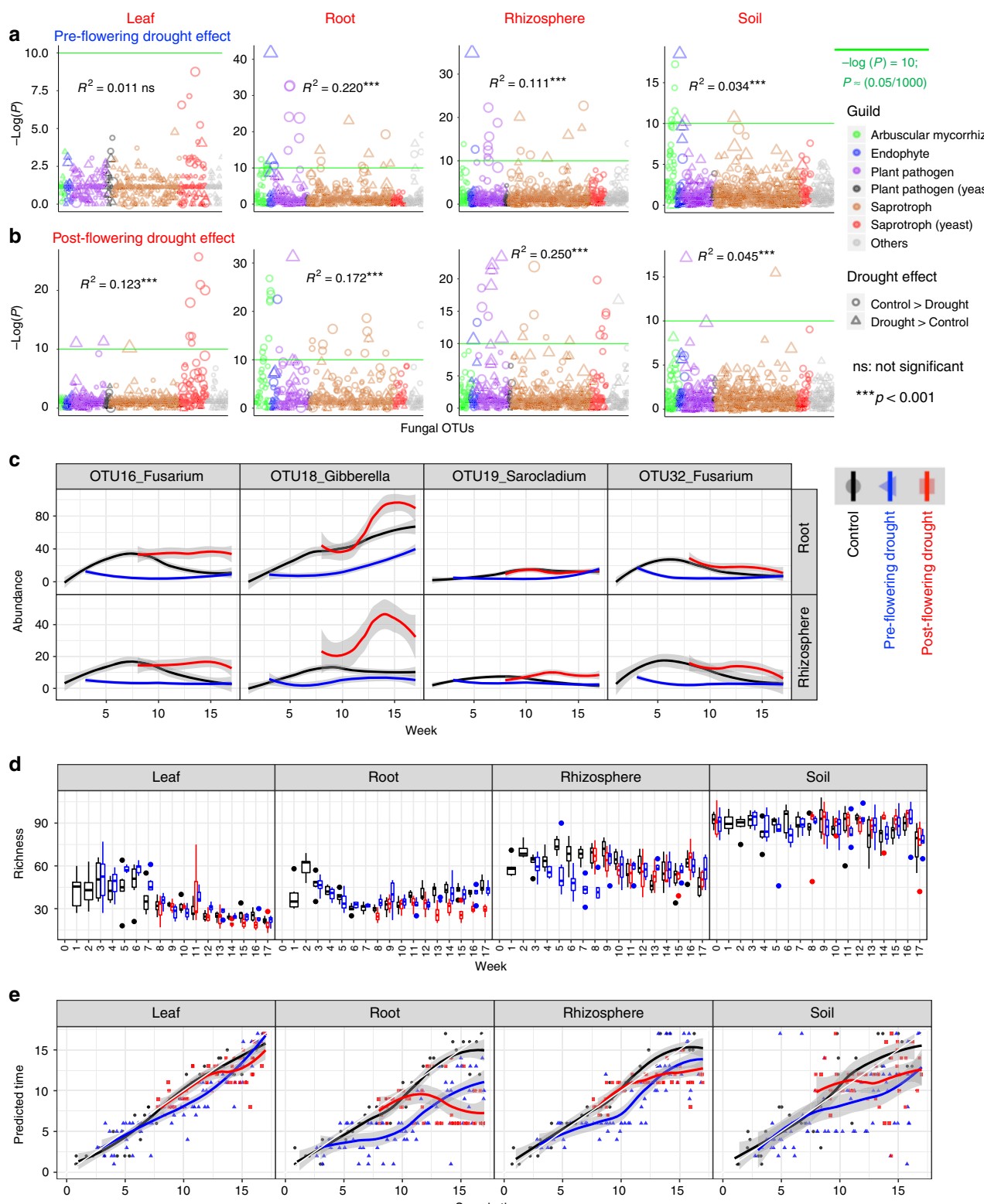

communities in both types of drought were enriched for the same actinobacteria[10]. For fungal communities, we found that the abundances of plant pathogens in the genera *Fusarium*, *Gibberella* and *Sarocladium* (OTUs 16, 18, 20, 32) were decreased by pre-flowering drought but increased by post-flowering drought in rhizosphere and at least partially in roots (Fig. 5a–c; Supplementary Fig. 28). Whereas bacterial richness was decreased by both pre- and post-flowering drought in roots and

rhizosphere[10], in the rhizosphere, fungal richness was decreased by pre- but not by post-flowering drought and, oppositely, in roots, fungal richness was decreased by post- but not by pre-flowering drought (Fig. 5d). Continuing the finding of different effects with the two types of drought and moving from richness to evenness, in the rhizosphere, fungal evenness was decreased by pre-flowering drought but increased by post-flowering drought (Supplementary Fig. 29).

**Fig. 5 Drought responses of sorghum mycobiome. a** Pre-flowering drought effect on OTU abundance and fungal community composition. Note the strongest effects ($R^2$) on root and rhizosphere. **b** Post-flowering drought effect on OTU abundance and fungal community composition. Note strong effects ($R^2$) on all compartments except soil. OTUs above a false discovery threshold [green horizontal line with $P < 0.00005$ ($\approx 0.05/1070$ OTUs) or $-\log(P) = 10$] show significant bias between drought and control. The symbol size corresponds to OTU abundance (log transformed) and color corresponds to fungus functional guild. Genera of the significant OTUs can be found at Supplementary Fig. 28. The $R^2$ is the difference in fungal community composition between control and drought treatments, as determined by permutational analysis of variance (PERM ANOVA). **c** Plant pathogenic fungal OTUs significantly affected by drought. In the compartments showing the strongest drought effects, root and rhizosphere, under post-flowering drought, plant pathogenic fungal OTUs become significantly more abundant than control, but under pre-flowering drought the pathogens are never more abundant than controls. **d** Boxplot showing OTU richness of fungal communities was significantly affected by drought in rhizosphere and roots. Note decreased richness under pre- but not under post-flowering drought in the rhizosphere and, oppositely, decreased fungal richness under post- but not pre-flowering drought in roots. **e** Delay of fungal community development by drought. Random Forest modeling of fungal community age shows that both pre- and post-flowering drought delayed the development of fungal communities to a similar extent. Random Forest modeling was used because pre- and post-flowering droughts are inherently temporal partitioned (Fig. 3), making it improper to simply compare the temporally variable, community compositional variance. Source data are provided as a Source Data file.

**Universal ecological dynamics.** Here we address the question whether the underlying ecological dynamics of microbiomes are universal across all communities or unique to individual communities? We firstly assessed universality using pairwise comparisons of all 1026 different fungal communities and then, again, as their several, component guilds, i.e., saprotrophs, plant pathogens, endophytes and yeasts, as well as just the AM fungal communities. In each pairwise comparison, dissimilarity is calculated for just the shared OTUs, those that overlap between the two communities. These dissimilarities are then plotted against the fraction of taxa that overlap to create a DOC[24]. Where the DOC dips as the overlap grows, universality is supported and the level of support is determined by the fraction of pairwise comparisons found where the DOC slope is negative (termed the fraction negative slope, Fns).

For our sorghum mycobiomes, the DOCs had significant negative slopes with Fns of 63.3% for the total fungal community comparisons (Fig. 6a), 36.1% for endophytic community comparisons, 91.0% for plant pathogenic fungal community comparisons, 44.4% for yeast community comparisons, 89.2% for saprotrophic fungal community comparisons, and 75.3% for the AM fungal community comparisons (Fig. 6d–h). The initiation of negative slope in each case represents the median of initiation of negative slopes calculated from DOCs of 1000 bootstrapped data sets. Thus, the results with sorghum mycobiomes and AM fungi in an agricultural field are similar to each other and to those reported for human-associated, bacterial microbiomes (Fns = 0.23*–0.99*)[24,25], and partially similar to those reported for AM fungi in natural and agricultural fields (Fns = 0.28*–0.94*)[26,27]. These comparisons suggest the existence of general universal ecological dynamics from human-associated ecosystems to agricultural ecosystems, and from bacteria to fungi and fungal guilds. Still, given the small number and scope of the studies, universality needs to be evaluated by more studies in more complex ecosystems and at larger scales[71–73]. The universal population dynamics suggest that a microbiome manipulation method valid in one system is likely to also work in other systems[24].

As a check on the approach, we applied it to our early leaf and root communities, which we had found to be formed by stochastic forces and for which universal behavior should be absent. When we calculated the DOCs for these communities, their Fns values were weak for both early leaves (Fns = 0.085) and early roots (Fns = 0.130), albeit significant owing to existence of weak selection (βNTI < −2 and RCI < −0.95) (Fig. 6b, c), in line with the predictions of neutral model simulation[74].

The detection of universal population dynamics raises the question of which species tend to co-exist[75]. The sorghum mycobiome in this study formed a co-abundance (φ < 0.1, R > 0.8,

$P \ll 0.001$) network embracing 12 clusters; in these clusters, fungal OTUs belonging to the same functional guild, such as, AM fungi, yeast, pathogen or saprotroph, tended to co-occur (Fig. 6i). Our results are in line with previous studies where positive correlations were demonstrated among yeasts of wheat leaves[44], and AM fungi of sorghum root, rhizosphere, and soil[11].

## Discussion

In early leaves and roots, we detected stochasticity (|βNTI| < 2 and |RCI| < 0.95) characteristic of communities lacking dominant effects of dispersal (homogenous dispersal, dispersal limitation) or selection (variable selection, homogeneous selection)[55]. This stochasticity, negatively correlated with fungal community size, was likely caused by ecological drift, but we cannot rule out a contribution by the initial, stochastic aspects of dispersal, i.e., stochastic colonization. Our likely detection of ecological drift in sorghum mycobiome almost certainly rests on our ability to capture and detect small fungal communities in early leaves and roots by weekly sampling beginning with seedling emergence, the choice of an extremely simple agricultural system with one species of plant and mechanical homogenization of soil, and the recent development of statistical methods that allow us to retrieve the stochastic component of compositional variance[55]. Our likely detection of drift in a very simple system raises the possibility that drift is also important in community assembly in truly natural systems. However, in these systems, which are complex, detection of drift would be far more difficult. The action of drift and stochastic colonization early in the development of leaves and roots, when selection and competition are relatively weak, may provide a temporal window to artificially introduce beneficial microbes and suppress harmful ones[53]. Later in the growth period, the sorghum mycobiome and its component functional guilds, e.g., plant pathogens, yeasts, AM fungi, and saprotrophs, are largely shaped by homogenous selection exerted by host over temporal, spatial and genetical dimensions, as well as by drought at pre- or post-flowering stages. At critical times in the growth cycle, such as drought, the detection of flowering-dependent responses of fungal pathogens in root and rhizosphere might be useful in agricultural management. Our detection of a plant genotype effect on fungal pathogens could also be useful in plant breeding. The different temporal behaviors of fungal communities found with leaves, roots and the rhizosphere highlight the underappreciated diversity of ecological patterns in temporal community assembly[29]. Our results also show that the fungal mycobiome is different from the bacterial microbiome in patterns of temporal structure, drought response and heritability at different compartments, despite the finding of universal underlying ecological dynamics. As noted at the outset, the number of studies of total

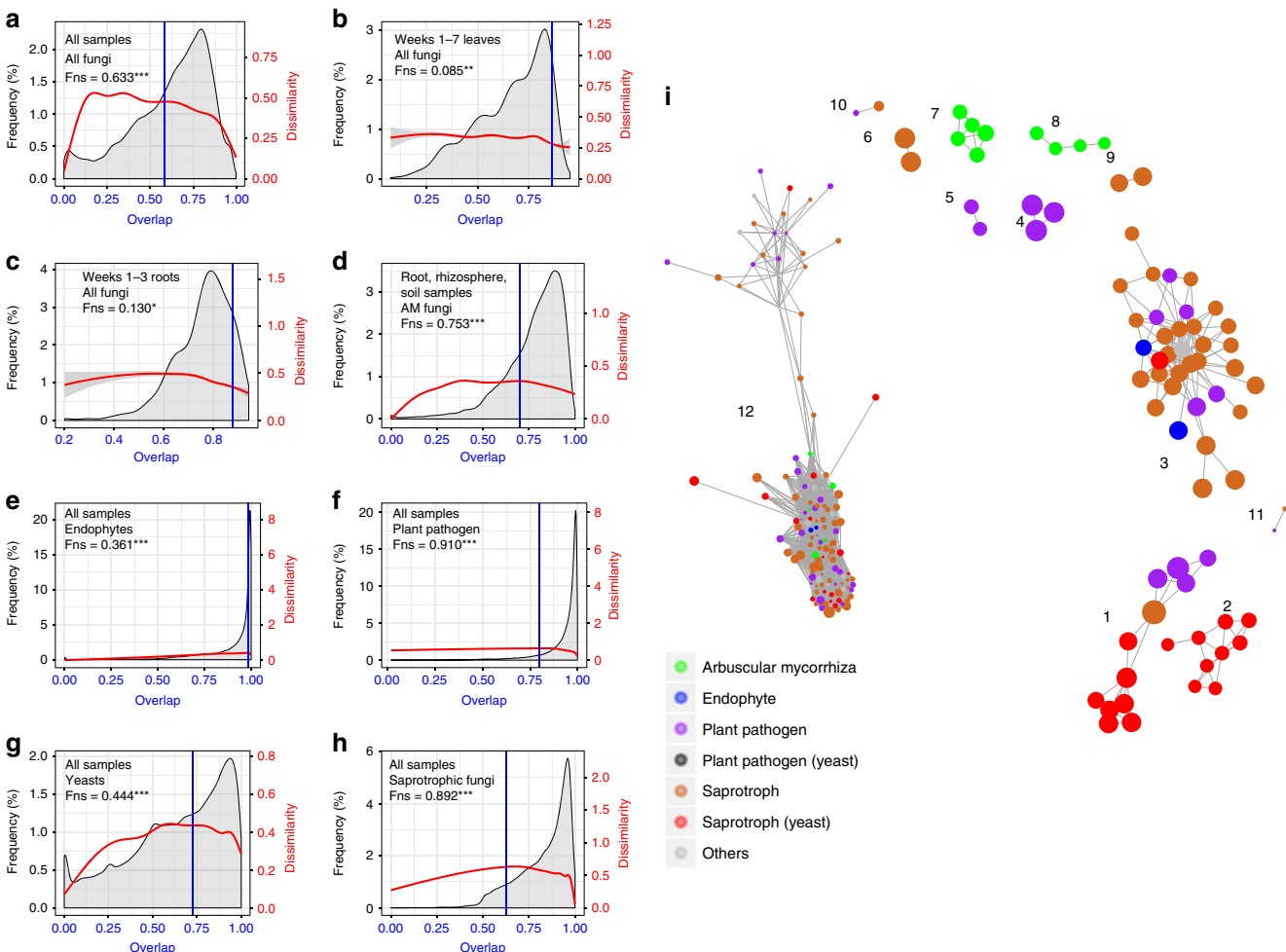

**Fig. 6 Universality of ecological dynamics of the sorghum mycobiome. a–h** Dissimilarity-overlap curves (DOC) for **a** all samples of all fungi, **b** leaf samples of weeks 1–7 for all fungi, **c** root samples of weeks 1–3 for all fungi, **d** root, rhizosphere and soil samples for all weeks of arbuscular mycorrhizal (AM) fungi, **e** all samples of endophytic fungi, **f** all samples of plant pathogenic fungi, **g** all samples of yeasts, and **h** all samples of saprotrophic fungi. For DOCs, the dissimilarity-overlap curve is in red, the distribution density of sample pair overlap is in gray, and the point at which a negative DOC is first observed is marked by a vertical blue line (chosen by median of 1000 bootstraps). The fraction of negative slope (Fns) is simply the fraction of data points in the interval where the DOC has a negative slope and significance is as follows: *$P < 0.05$, **$P < 0.01$, ***$P < 0.001$. **i** Co-occurring network of fungal guilds of sorghum Mycobiome. Each dot represents a fungal OTU, and each edge represents a positive correlation with $R > 0.8$ ($P \ll 0.001$). The different color represented the functional guilds of fungi. The dot size corresponds to OTU abundance (log transformed). Note for the 12th cluster, co-occurrence may be exaggerated among the large number of rare OTUs due to the dominance of zeros for rare OTUs in most samples. Source data are provided as a Source Data file.

bacterial communities far exceeds those of total fungi. We hope that our results indicate that fungi not only can be included in studies of plant-associated microbes, but should be included to better integrate fungi into the practices and tools of modern agriculture.

## Methods

**Experiment design and sampling.** The methods described here reflect those in our previous publications of bacteria and AM fungi from the same study site[10,11]. Our experiment is a random block design of three replicates of three treatments (control, pre-flowering drought and post-flowering drought) and two sorghum [Sorghum bicolor (L.) Moench] cultivars [the pre-flowering, drought resistant sorghum cultivar RTx430, and the post-flowering, drought tolerant (or 'stay green') cultivar BTx642] for which we collected a total of 1026 samples (Fig. 1). The sorghum seeds were sown into pre-watered fields and left unirrigated for two weeks. Prior to planting, at time point 0 (TP00), and for each of the subsequent 17 weeks, in each of the 18 plots, ten soil cores (6" depth using 6" soil collection tubes) were randomly collected and pooled (Fig. 1). From the 3rd week until the final harvest, the plants were either regularly watered in the control treatment, or were not watered until the ninth week in the pre-flowering drought treatment at which time regular watering was initiated, or were regularly watered until the 10th

week in the post-flowering drought treatment at which time watering ceased (Fig. 1). The trial was planted on May 27, 2016 and plant emergence was recorded on 1 June 2016[11]. Weekly samples of leaf, root, rhizosphere and soil were taken in 2016 for control plots on June 8, 15, 22, 29; July 6, 13, 20, 27; August 3, 10, 17, 24, 31, and September 7, 14, 21, 28[11] (Fig. 1b). To avoid redundancy, pre-flowering treatment sampling began at June 22 (TP03) and post-flowering sampling began at 27 July (TP08) (Fig. 1b)[10]. Between 10:00 and 14:00 of every sampling date, at least ten individual sorghum plants were removed from randomly chosen locations within one of the central eight rows in each plot. To sample leaves, the 3rd and 4th youngest, fully expanded leaves of the ten plants were removed, put into an aluminum packet and frozen in liquid nitrogen. The sampling of root, rhizosphere and soils is described in detail in our previous publications[10,11].

DNA was extracted from leaf, root, rhizosphere, and soil samples using the MoBio PowerSoil DNA kit (MoBio, Carlsbad, CA, USA). DNA concentration was measured using a Qubit dsDNA HS kit (Life Technologies Inc., Gaithersburg, MD, USA). Fungal internal transcribed spacer 2 (ITS2) was PCR-amplified from DNAs diluted to 5 ng/µl with ddH$_2$O, using dual-barcoded 5.8SFun (AACTTTYRRCAAYGGATCWCCT) and ITS4Fun (AGCCTCCGCTTATTGATATGCTTAART)[11,30]. The yields of PCR products were quantified using a Qubit dsDNA HS kit (Life Technologies Inc., Gaithersburg, MD, USA) and 200 ng of DNA from each of the 1026 samples were randomly assigned to four different pools, purified using AMPure magnetic beads (Beckman Coulter Inc., Brea, CA, USA), checked for concentration and amplicon size using the Agilent 2100 Bioanalyzer (Agilent

Technologies, Santa Clara, CA, USA), and sequenced on the Illumina Miseq PE300 sequencing platform (Illumina, Inc., CA, USA) at the Vincent J. Coates Genomics Sequencing Laboratory (GSL, University of California, Berkeley, CA, USA).

Detailed description of bioinformatic analysis can be found in our previous publication[11]. Briefly, raw fastq sequences were subjected to quality evaluation using FastQC v0.11.5[76], removal of primers using cutadapter v1.9.1[77], merging of forward and reverse reads, control of quality, clustering of OTUs, and global search using USEARCH v8.0[78], to generate a table of 1026 samples × 1293 OTUs (39,710,336 reads). The representative sequence of each OTU was identified by a BLAST search against the curated, fungal specific UNITE database[79] and the NCBI database. A total 1070 OTUs were identified as fungal (34,541,758 reads), and 223 OTUs were non-fungal (5,168,591). Fungal OTUs were assigned into functional guilds using the FUNGuild v1.1[80].

Fungal biomass was estimated by quantitative PCR (qPCR) of the fungal small subunit rRNA (SSU or 18S) using the FF2 (GGTTCTATTTTGTTGGTTTCTA) and FR1 (CTCTCAATCTGTCAATCCTTATT) primers[81]. Analysis of qPCR mixtures was accomplished using a Real-Time PCR Detection System (Bio-Rad, Hercules, CA, USA) containing 1 μl of 5 ng/μl genomic DNA, 10 μl iTAQ SYBR Green Supermix with ROX (Bio-Rad, Hercules, CA, USA), 0.2 μl of 100 mg ml$^{-1}$ BSA, 0.15 μl of each 50 μM primer and water to 20 μl[82]. Thermal cycling conditions consisted of an initial denaturation at 95 °C for 3 min, followed by 40 cycles of 15 s of denaturation at 95 °C and 1 min of annealing and extension at 60 °C, finishing with a dissociation stage of 95 °C for 15 s, 60 °C for 30 s, and 95 °C for 15 s[82]. Standard curves were developed using a series of 10-fold dilutions of plasmids containing a fragment of an insert of the 18S gene of *Penicillium purpurogenum*[82].

Alternatively, fungal relative abundance was estimated by the percent of fungal transcripts in transcriptomes containing both sorghum plants and their fungi generated from 198 root samples and 197 leaf samples[28]. Using the BBsplit script in BBmap[83], fungal transcripts were mined from these transcriptomes, by referencing fungal genomes in MycoCosm[84] of *Cladosporium fulvum* v1.0, *Paraphoma chrysanthemicola* PD 92/468 v1.0, *Acremonium strictum* DS1bioAY4a v1.0, *Alternaria alternata* 133aPRJ v1.0, *Fusarium fujikuroi* IMI 58289, *Fusarium verticillioides* 7600 v2, *Fusarium oxysporum* f. sp. lycopersici 4287 v2, *Chaetomium globosum* v1.0, *Talaromyces marneffei* ATCC 18224, *Cryptococcus vishniacii* v1.0, *Ustilago maydis* 521 v2.0, and *Sporobolomyces roseus* v1.0.

In 2017, thirteen passive air samplers (empty petri dishes) were installed following a nested design (Supplementary Fig. 23A). Briefly, the origin sampler was located at the southwest corner of the sorghum field, and four samplers were located at the distance of 7.5, 15, 30, and 60 m along the two edges and diagonal (Supplementary Fig. 23A). The air samplers (empty petri dishes) were placed within a vessel with open holes to allow air circulate, which was protected by a conical roof from potential rainfall, and was placed one meter above the ground by a steel bar (Supplementary Fig. 23A). Sterile petri dishes were firstly placed at 15 August and collected at 13 September 2017, and secondly placed at 13 September 2017 and collected at 11 October 2017. Dust that had settled on the petri dishes was transferred to 2 ml DNA extraction tubes using sterile, DNA-free, swabs soaked with lysis buffer, and subjected to DNA extraction, ITS2 PCR amplification, library construction and Illumina Miseq sequencing as described above.

**Statistical methods.** The functional affiliation of fungal genera and their relative abundances of both unplanted soil and sorghum mycobiomes were depicted by Krona charts constructed using the ktImportText command of the KronaTool v2.7 (https://github.com/marbl/Krona). Fungal OTUs distributions among leaf, root, rhizosphere and soil (both unplanted and planted) were visualized by Ternary plots using the ggtern package[85] in R v3.5.1[86]. Indicator species for selected sampling times were identified and their threshold values were calculated using threshold indicator species analyses (TITAN) in the TITAN2 package[87] in R. Manhattan plots were constructed in the ggplot2 package[88] in R to visualize the P-values for the pairwise comparison of fungal OTUs between cultivar BTx642 and cultivar RTx430, and between control and either pre- or post- flowering drought. The most important age-discriminant fungal OTUs were identified from a random subset of 50% control samples using the random forest (RF) model in the RandomForest package[89] in R, and then visualized in a heatmap plot using the pheatmap package[90] in R. The sparse RF model of these age-discriminant fungal OTUs was used to predict the ages of drought samples and another subset of 50% control samples; and the discrepancies between predicted ages of control and drought samples were used to assess the extent of drought effect. To visualize the relative abundances of fungal functional guilds, fungal phyla, and common fungal OTUs, bar plots were constructed using the ggplot2 package[88] in R.

Recent recognition that microbiome data from HTS represents a random sample of the DNA molecules in an environment and not absolute counts of the molecules argues that the data be treated as compositional and not as counts[31], as commonly has been done. Therefore, where possible, we use both count and compositional methods to both permit comparisons with prior studies and to analyze the data as compositional. For the count approach (dataset 1), we rarefied the number of fungal sequences per sample to 362 (the smallest read number among all the 1026 samples) using the rrarefy command in package vegan[91]. Detail about the compositional method (dataset 2) can be found at our previous publication[11]. Briefly, the raw read data is transformed by zeros imputation and

centered log-ratio (CLR) conversion using the zCompositions package[92] and the CoDaSeq package (https://github.com/ggloor/CoDaSeq)[31]. Direct comparison of the two approaches is possible with permutational analysis of variance (PERM ANOVA), but not for other analyses because the statistical methods for compositional datasets are different from those for traditional count datasets, e.g., Bray–Curtis dissimilarity for counts v. Aitchison distance for compositional, and principal coordinate (PCo) analysis for counts v. principal component (PC) analysis for compositional[31]. For most of our analyses, methods are not yet available for compositional datasets, e.g., the DOC[24], the Simpson metric[33,93], the RCI[54], and the βNTI[53].

Bray–Curtis dissimilarities were calculated for dataset 1 to construct distance matrices of the fungal community (Hellinger transformed) using the vegdist command in vegan package[91] in R, and Aitchison distances were calculated for dataset 2[31]. PERM ANOVA was carried out to assess the effect of compartment (leaf, soil, rhizosphere or root), time period (weeks 1–17), cultivar (BTx642, RTx430) and drought treatment (control, pre-flowering drought, or post-flowering drought) on the fungal community variation either detected by Bray–Curtis dissimilarities or Aitchison distances using the adonis command in vegan package[91] in R. To visualize the variations in fungal community compositions, the Bray–Curtis dissimilarity was subjected to principal coordinate analysis using the pcoa command in the Ape package[94], and the Aitchison distance was subjected to principal component analysis in base package[86] in R. Euclidean dissimilarities were calculated to construct distance matrices of geographic and temporal distances in the vegan package[91] in R. Mantel tests were carried out to explore the correlations between geographic, temporal and community composition distance matrices in the vegan package[91] in R. To remove the effect of richness variation on fungal community composition, the Simpson metric of community dissimilarity was calculated based on presence/absence data using the beta.pair command in the betapart package[95] in R, and then ordinated by the principal coordinate analysis in the Ape package[94] in R. To test the homogeneity of the fungal community during succession[96], beta dispersions of Bray–Curtis, Jaccard and Simpson dissimilarities were explored by the betadisper function in the vegan package[91] in R. To explore the impact of richness on bacterial community compositional variances, we used betadispesion analysis of the Jaccard and Simpson dissimilarities for the rice datasets[35].

To assess stochasticity in fungal community assembly and temporal change, we calculated the Raup-Crick index (RCI) and the beta Nearest Taxon Index (βNTI) using the scripts of Chase et al.[54] and Stegen et al.[53] in R. Stochasticity was recognized from the proportions of community pairs that fell within |RCI| < 0.95 and |βNTI| < 2. Our calculation of βNTI from 1070 fungal ITS2 OTUs relied on a fungal phylogeny based on 18S + 28S rDNA sequences (taxonomy_to_tree.pl script of Tedersoo et al.[56]). To investigate potential links between stochasticity and ecological drift, relationships of the proportion of |RCI| < 0.95 and |βNTI| < 2 were compared with the size of fungal communities as estimated from: (i) the percent of fungal reads as compared to the total fungal and plant reads in ITS2 amplicon and Illumina sequencing, (ii) abundance of fungal SSU detected by qPCR and (iii) the percent of fungal transcripts as compared to the total fungal and plant transcripts in sorghum root and leaf transcriptomes. Meanwhile, we also explored the relationships between fungal richness and the proportion of |RCI| < 0.95 and |βNTI| < 2, to explore the link between stochasticity and ecological drift. These comparisons of stochasticity and fungal abundance and richness were explored by linear mixed-effects models, that included random effects of compartment type using the lme command in the lme4 package[97] in R. The variance explained (conditional $R^2$) by the mixed effect models was calculated by the r.squaredGLMM function in the MuMIn Package[98] in R. To depict the compositional variation in fungal communities, Fst measures were calculated using the script of Gilbert and Levine[18] in R.

To assess the universality in ecological dynamics across fungal communities, we used the DOC approach by constructing the curves using the DOC package (https://github.com/Russel88/DOC) in R. The DOC emerges by plotting, for each possible pair of communities, the dissimilarity on the y axis (root Jensen–Shannon divergence as calculated from only the OTUs shared by the two communities) against the fraction of taxa that overlap on the x axis. Where the DOC dips as the overlap grows, universality is supported and the level of support is proportional to the fraction of pairwise comparisons under the graph where the DOC slope is negative (termed the fraction negative slope, Fns). For the smoothed curve of a given DOC, the initiation of negative slope represents the median of initiation of negative slopes calculated from DOCs of 1000 bootstrapped data sets. To explore which OTUs tend to co-occur, a co-abundance network was constructed using the igraph package[99] in R. To avoid type I error of multiple comparisons, all statistical significance measures were corrected using the Bonferroni method[100].

**Reporting summary.** Further information on research design is available in the Nature Research Reporting Summary linked to this article.

## Data availability

All data that support the findings of this study have been deposited in (a) GenBank (representative read set) with accession codes MG008508 to MG008559, and MK018174 to MK019191; or (b) Sequence Read Archive (raw data) with the following accession codes: Bioproject PRJNA412410 and PRJNA494573; Biosample SAMN07711256 to

SAMN07711567, SAMN10176611 to SAMN10176624, SAMN10173923 to SAMN10174030, SAMN10173711 to SAMN10173818, SAMN10173573 to SAMN10173680, SAMN10173450 to SAMN10173557, SAMN10173164 to SAMN10173298, SAMN10173035 to SAMN10173160, and SAMN10172702 to SAMN10172707. The source data underlying Figs. 2–6 and Supplementary Figs. 1–29 are provided as a Source Data file.

## Code availability

The interactive Krona figures of fungi in unplanted soil and sorghum mycobiome are available as Supplementary Software 1. All scripts used in this study are available at GitHub (https://github.com/ChengGaoBerkeley/EPICON.Mycobiome).

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

## Acknowledgements

The research was supported by the program of Systems Biology Research to Advance Sustainable Bioenergy Crop Development of DOE BER (DE-FOA-0001207), under the fund #DE-SC0014081. The work conducted by the US DOE Joint Genome Institute is supported by the Office of Science of the US Department of Energy under Contract no. DE-AC02-05CH1123. We thank Steven E. Lindow, Catharine Adams, Claire E. Willing, Liang-Dong Guo, Lei Chen, and all members of Taylor and Bruns labs for insightful discussions and advice on the manuscript. We thank Rachel Adams for the protocol and reagents of qPCR. We thank Julie Sievert for assistance with the design and operation of the field experiment, sample collection, and plant phenotyping, Grady Pierroz for sample collection and management, and Barbara Alonso for assistance in project management.

## Author contributions

C.G. and J.W.T. conceived of and wrote the manuscript for the fungal portion of a larger project conceived of by P.G.L., D.C.D., E.P., J.V., J.A.D. and J.W.T. with P.G.L. as director. P.G.L., R.B.H., J.H., J.A.D. and M.M. coordinated the field work and sample storage. C.G., L.M. and L.X. performed the molecular analysis. C.G. performed the bioinformatics and statistical analyses. V.S. and J.V. performed the extraction of fungal transcription from sorghum transcriptome. E.P. and D.C.D. provided ideas about statistic and bioinformatics.

## Competing interests

The authors declare no competing interests.
