## [Peer Review File · Nature Communications]

Reviewers' comments:

Reviewer #1 (Remarks to the Author):

The authors used several state-of-the-art approaches in ecological statistics to draw inferences from an impressive data set of plant-associated fungal communities. The key outcome is the finding that plant-associated fungal communities associated with sorghum in this relatively homogeneous field are largely predictable following a brief period of more stochastic dynamics, particularly drift, that are associated with fungal communities containing relatively few individuals. There are also a few other outcomes, including that comparisons with bacterial community assembly dynamics suggest that fungal dynamics differ to a certain extent and that the timing of the imposition of drought influences the occurrence of drift in some compartments. The work is very descriptive - two hypotheses are tested (well), but the bulk of the manuscript is focussed on describing patterns among compartments, taxa and guilds. But I think that this is a strength of the current paper - it sets up a framework in which additional interesting ecological hypotheses can be tested in systems that are better suited to them (e.g., why the difference in turnover between leaves, roots and other compartments? The current study cannot answer this.) and, for the most part, does not over-interpret the data.

I'm not sure that I agree that there is support for the idea that the few studies that have used the approaches described in the paper suggest that universality is likely a general phenomenon for fungi, including in natural environments. E.g., dissimilarities in the current study (Figure 4a) were quite low for samples collected at or near the same time, many studies comparing communities among more heterogenous environments exhibit dissimilarities that are much higher - just to name a few:

Tedersoo et al., 2014. Global diversity and geography of soil fungi. *Science* 346, 1256688.

doi:10.1126/science.1256688

Davison et al., 2015. Global assessment of arbuscular mycorrhizal fungus diversity reveals very low endemism. *Science* 349, 970–973. doi:10.1126/science.aab1161

Powell et al., 2015. Deterministic processes vary during community assembly for ecologically dissimilar taxa. *Nature Communications* 6, 8444. doi:10.1038/ncomms9444

I know that it isn't a direct comparison so it doesn't invalidate the authors claims, but perhaps the current manuscript doesn't need to make this point about generality?

I wonder whether the sampling approach (combining ten samples from each plot to produce a composite sample) may lead to underestimation of stochastic processes. This would not invalidate the comparisons among compartments and through time, of course, but could be addressed in the manuscript where the sampling approach is briefly mentioned. I would also appreciate a brief description in the supplementary methods of how roots, rhizosphere and soil were sampled - currently there is only a citation to two other papers, I was able to access one of them but the methods section is large and took a long time to download.

The paper is very well written and the methodological details are (mostly) described thoroughly or appropriately cited. I've made a few suggestions in the attached annotated manuscript. The supplementary materials make it very easy to see the outcomes of the data analyses, including the Krona plots, so the work is transparent. However, to ensure reproducibility, I would have appreciated a series of markdown files that demonstrate the code used to perform analyses.

Reviewer #2 (Remarks to the Author):

Comments to Authors:

The authors present an ambitious work detailing the temporal, compartmental, and genotype effects of the mycobiome across an entire growing season of Sorghum. The amount of work that went into this MS is quite admirable. They test theories on the nature of fungal assembly and specifically address the role community drift (stochasticity) plays in the mycobiome. The results presented are important, but the MS lacks focus and it is very hard to follow. A careful rewrite will aid the reader in the story, especially being careful to reduce extraneous language to streamline the MS. Also, there are a number of methodological and statistical concerns that should be addressed (outlined below). Particularly troubling is the inclusion of Beta-NTI analyses (which the author's use numerous times to support their assertions). Beta-NTI requires a good tree to measure the average nearest taxon relationships. However, given that these were fungal ITS2 data (and highly diverse at that), a good tree cannot be made (poorly alignable data). So, how can the authors justify this method? There is not enough detail in the MS to glean if they found a work around to this or even any methods detailing how they generated the tree (even in the supplemental methods). This lack of detail and the inhering inappropriateness of the measure for ITS data casts doubt on these data. This reviewer suggests eliminating all beta-NTI references (focus on Beta RC instead) or VERY CAREFULLY alleviate my concerns. This work is of great interest and would appeal to the readership of Nature Communications, but only is acceptable if appropriate methodological approaches are implemented.

Abstract

Line 40: Comma seems out of place ("fungal, functional")

Introduction:

Lines 46-47: It is unclear how endophytes (which by the most definitions are generally commensal) might drive plant community structure, at least in the same vein as mycorrhizae or parasites/pathogens. Plant communities more likely drive endophytes rather than the reverse. So, I am not sure how well "it is well appreciated."

Line 49: Is "highlighted in the last decade" to reductionist? This has been a major study system for a long time.

Lines 49-51: Not sure this is a fair statement. One could easily argue that studies on fungi are more abundant than bacteria given the vast literature on mycorrhizal associations. This may be true for plant endophyte work (but I suspect not), but this statement says all studies of fungi-plant connections. Please omit or change.

Lines 51-53: This makes little sense.

Lines 57-59: This is making some assumptions that may not be justified. We do not know (or at least you do not present any evidence that we do know) that increased drought will affect plants via fungal endophyte-mediated plant responses negatively. One could easily argue that endophytes may confer drought resistance.

Lines 59-61: This is a fallacy. It is very possible to harness plant-associated fungi to aid modern agriculture without understanding assembly. If you see that one fungus is beneficial and enriched

in a particular treatment, it can be harnessed and utilized for a function, without understanding the mechanisms by which it was assembled.

Line 64: Nearly all possible combinations? Why nearly?

Lines 66-68: Seem irrelevant and unneeded.

Lines 70-72: Is it fair to say that selection is wholly deterministic, and drift is wholly stochastic. Largely, sure. Wholly? Perhaps not. This may be a bit reductionist.

Lines 69-76: This paragraph could be rewritten for clarity.

Lines 77-85: Sentence structure here and throughout is a bit odd. Recommend using more direct statements and shy away from over-comma use. This will do much to clarify the MS.

Line 86-88: Why do we think that these communities will be small during early development. I am not saying they will not be, but it would be useful to provide justification to this.

Lines 90-92: "when stress ... was relieved"- This makes little sense. Clarify please.

Lines 92-95: Please clarify this sentence. The words "negative correlation between the strength of stochasticity on one hand and community size on the other" makes little sense out of context. This would benefit from a rewrite.

Lines 104-107: I would tread carefully here, while it may be true that this MS is the first to use this method on non-AM systems to investigate 'universality', you are far from being the first to investigate the universality of fungal assembly rules. As written (though I know this is not what you are trying to say) it implies that you are the first to try this. Folks have been looking into universality of fungal assembly since the time of Clements. Choose your words carefully.

Line 108-109: Is this novel? Of course, fungal communities will be more similar when shared number of taxa increases. This is the whole point of most dissimilarity measures that ecologists have been using since the 1950s. Perhaps you are meaning this differently that is written. If so, clarify please.

Lines 117-119: This makes little sense. How can one take leave samples from soil? Please re-write to be explicit about what you did here.

Lines 115-127: This paragraph is oddly structured.

Lines 122-127: Should this be included here. This is just about what you would expect to find in any ag field. I would omit this here and expand the methods above.

Lines 128-144: Did the authors investigate any space x time interactions? Or, perhaps more interestingly, time within a particular "space" to investigate temporal effects within roots for examples. By doing two sets of analyses (whole model and individualized compartment models), the authors will allow for a more robust way to explore fungal community dynamics. But including the compartment (when one would naturally expect little overlap) in all analyses, that may parse so much variance out that the more subtle signals from other factors may be occluded.

Lines 149-152: Not sure if this is the most appropriate way to do this. Technically you are not supposed to run Mantel tests on non-distance data (BC is not true distance) but I understand that this is often done anyways. I would recommend either: 1) simply regressing BC values against time (weeks difference) to look for temporal shifts, or 2) using a Procrustes approach where you compare soil vs. lead (and every other combination) and regressing the Procrustes Associated Metric over time to ask which compartment is most responsive over time. Also looking at a full

interaction permanova would get at this as well.

Lines 157-158: Is this fair? It might not be assumed to be a plant driven community shift, but a propagule availability issue. This is an equally parsimonious explanation.

Lines 156-168: I Like this, but object to the phrase "stable community" here. Maybe replace it with "locally stable" or "temporarily stable"

Lines 167: This idea that rhizospheric communities have little turnover even in very early development casts doubt on previous statements that these communities are small, therefore open to drift (H1). I believe you but be careful with language.

Lines 169-186: This needs work. I like what you are getting at, but the writing is almost not linear. It is very difficult to interpret. Clean this up and make the point for the reader. What does this mean with respect to these communities.

Line 195: I would argue that "compositional variance" is not the same and "dissimilarity values"

Lines 208-215: I like the acknowledgement here, but I wonder how much this adds to the MS. I suggest reducing this paragraph and folding it into a different heading, I think it is not warranted to have a whole section.

Line 217: Genetical? This is very odd wording. You have used genotype throughout, it should probably be consistent.

Lines 220-223: This might be an issue on significant but unimportant (given the R-square values). However, this might be an issue where reanalysis as suggested above might give you more power. Also, what evidence are you using to make the claim that this genotype effect was due to these two OTUs? Just because they are differentially abundant, does not necessarily mean that these are the full drivers of this genotype effect.

Lines 229-232: Again, a reanalysis may give you more power to make these claims.

Lines 244-246: No sold that you adequately demonstrate this. Allochthonous dispersal is a constant issue with strong temporal and environmental effects on composition. I think dispersal is very likely and should be acknowledged. I fail to understand the argument that high richness and lack of local distance decay relationships nullify this concern. You may have lack of distance relationships due to mass broadcasting of fungal spores regionally.

Line 250: I would like to see an expansion of this line. Why MUST it be associated with size? It seems that if it is associated with community size, that in of itself would be a slight deterministic introduction. Why does the rate of randomness change with community size? Since this is a big part of your entire thesis, this should be expanded on and not just accepted. Justification is needed.

Lines 259-263: Did you do any more simple analyses such as estimated richness? Chao? To point (i), this should not be used as there are SO MANY issues based on individualized potential for biases from extraction, PCR efficiency, primer affinity, etc. Point (ii) is probably fine but copy number concerns could occlude this. Single copy genes should optimally be used.

Lines 263-271: This contradicts the statement made in lines 255-256. Which is it? That statistical tools exist or they do not.

Lines 274-276: Beta NTI requires a tree based on aligned data. Given that you used ITS2 data, how did you make this tree. I am suspicious of these data given that it is very unwise to align ITS

data. If you did align these, I would not trust these results as far as I can throw it (and this is a large part of your explanations of drift - JUSTIFY). Please clarify. I like the inclusion of RCI, but would not beta-RC be a better metric here?

Lines 282-292: Paragraph lacks focus.

Line 294-295: Priority effects can be stochastic as well, not just deterministic

Lines 293-310: Difficult to follow.

Lines 311-322: These explanations for initial colonizers are perhaps too much of a stretch.

Lines 353-354: I guess I am still a bit confused why releveling drought would be expected to favor stochasticity. With resource limitations (water availability) being alleviated, classic ecological theory would predict that this should favor specialists over generalists, this implies, to me at least, determinism, not stochasticity. Please clarify.

Lines 374-379: Inclusion of RF methods is nice, but this does not really explain why this was done or what expected results would tell us about the communities. Give that they did not add to the story, I wonder if it would be better to omit this analysis as I fear it makes the story more convoluted.

Figure 1: This is a good figure, but a bit difficult to follow when also reading the MS. I suggest expanding the MS to describe this setup better (but figure is good).

Figure 2: This is VERY busy. Are the OTU values to the right of A for B? I suggest remaking these so only the ten most abundant OTUs are represented and 'Other'. This will clean it up quite a bit. Figure SA is nice, but the points are too small to see the compartment icons.

Figure 3: This is a nice figure, but I fear it doesn't really tell us that much about the communities and is sort of redundant of other similar figures. I would recommend eliminating this figure (or moving to Supp.)

Figure 4: Beautiful!

Figure 5: Given that there was an overall small effect of cultivar (and largely dismissed in MS), it seems odd to dedicate a figure to this. BUT it is a gorgeous figure, it just doesn't belong in the main MS.

Figure 7: You have to do a lot to convince me that Beta NTI is appropriate here due to ITS2 data.

Supplemental Methods:

Lines 31-36: What primers were used?

Reviewer #3 (Remarks to the Author):

In the manuscript by Gao and colleagues, the researchers examine the temporal patterns of fungal colonization and establishment on sorghum plants. They use a factorial field experiment to test the effects of sorghum genotype (2 levels) and drought treatment (3 levels) on four components of

the plant fungi microbiome, with three replicates of each treatment grown in a plowed field. The main focus of this manuscript arises from their interpretation of the results – they posit that ecological drift is important in driving communities when the fungal communities are small, which occurs at the outset of the experiment.

I do not have a background in fungal microbiomes, and will leave the novelty of that part of the paper to other reviewers. My expertise lies in the community ecology inferences that the authors are making and, based on the evidence presented, I am unconvinced that they are real. I outline my concerns below, but want to emphasize that, even if all other issues could be addressed, the low sample size for estimating variances (3 observations) would still raise serious concerns for me. The exciting conclusion of this paper is that ecological drift plays an inordinately large role at the outset fungal community assembly. There are several reasons why this conclusion is unsupported, and they fall into a few main problem areas: misinterpretation of the ecological literature on drift and dispersal limitation, statistical problems / inappropriate tests, and experimental flaws.

1. Experimental flaws. There are two main problems with the experiment. First, the environment is not carefully controlled – it took place in a plowed field, and plowed fields are heterogeneous. If we imagine that the fungi undergo successional dynamics whereby the environment determines which are initially most abundant but over time the fungi-plant interaction selects for specific fungi, we would recreate the patterns the authors observed (without evoking drift). Second, the authors did not control for the initial abundances of fungi, which could exacerbate environmental effects or work independently of them. They seem to interpret the widespread, initial abundance of a few OTUs as evidence that dispersal limitation isn't present. This interpretation is incorrect – most plowed fields will be colonized instantly by thousands of dandelions, but that does not mean that no plant species are dispersal limited. An experiment designed to isolate the effects of drift would control both these aspects by using an homogeneous (sterile) soil with known or well-mixed inocula.

2. Misinterpretation of literature. i. The authors test for distance-dependent dissimilarity in fungal communities as a way to assess whether dispersal limitation is important. Classic work by Hurtt and Pacala (J. Theor Biol 1995) show that dispersal limitation (or fecundity limitation) can cause strong colonization effects in otherwise completely deterministic communities. Distance decay is not the only way that stochastic colonization effects can manifest – differences in abundance among species can be sufficient to cause these to happen. The data suggest that not all species are present at all locations (far from it, given the turnover stats), meaning that colonization differences are likely important. ii. The authors seem to be under the impression that a Raup-Crick index informs us about neutral stochasticity (I apologize if I am misinterpreting this but, if so, I think that the interpretation needs to be clearer in the paper). The Raup-Crick is a null model for random redistribution of individuals, which is very different from expectations for drift. The other metric (BNTI) has no relationship whatsoever.

3. Statistical problems. Apart from the issue with the Raup-Crick and BNTI methods and their interpretations, I was unsure of how the authors examine beta diversification. Their graphs seem to have more observations per time point than there were replicates. I imagine that this is an error in the presentation, but should be fixed. The fitting of the neutral model was also challenging – it appears that the authors compared AIC values between two datasets (their initial data split into two) rather than for nested models. This either needs to be fixed (if the authors did compare two datasets) or the authors should explain their methods more clearly. Beyond these issues, the authors should consider the limitations outlined above (concern #1) and think about if the temporal data could be used in a way that actually detects drift by decoupling random from directional changes in identities through time.

On a final note, I think that it would be completely reasonable for the authors to discuss drift as one mechanism driving the patterns observed, but the current manuscript does not convince me that this is the only (or even the most likely) possibility.

**Reviewer #1 (Remarks to the Author): (Our responses are in red)**

The authors used several state-of-the-art approaches in ecological statistics to draw
inferences from an impressive data set of plant-associated fungal communities. The key
outcome is the finding that plant-associated fungal communities associated with sorghum in
this relatively homogeneous field are largely predictable following a brief period of more
stochastic dynamics, particularly drift, that are associated with fungal communities
containing relatively few individuals. There are also a few other outcomes, including that
comparisons with bacterial community assembly dynamics suggest that fungal dynamics
differ to a certain extent and that the timing of the imposition of drought influences the
occurrence of drift in some compartments. The work is very descriptive - two hypotheses are
tested (well), but the bulk of the manuscript is focused on describing patterns among
compartments, taxa and guilds. But I think that this is a strength of the current paper - it sets
up a framework in which additional interesting ecological hypotheses can be tested in
systems that are better suited to them (e.g., why the difference in turnover between leaves,
roots and other compartments? The current study cannot answer this.) and, for the most
part, does not over-interpret the data.

**Response: We appreciate the reviewer's comments about the strength of our manuscript.**

I'm not sure that I agree that there is support for the idea that the few studies that have used
the approaches described in the paper suggest that universality is likely a general
phenomenon for fungi, including in natural environments. E.g., dissimilarities in the current
study (Figure 4a) were quite low for samples collected at or near the same time, many studies
comparing communities among more heterogenous environments exhibit dissimilarities that
are much higher - just to name a few:

Tedersoo et al., 2014. Global diversity and geography of soil fungi. Science 346, 1256688.
doi:10.1126/science.1256688

Davison et al., 2015. Global assessment of arbuscular mycorrhizal fungus diversity reveals very low
endemism. Science 349, 970–973. doi:10.1126/science.aab1161

Powell et al., 2015. Deterministic processes vary during community assembly for ecologically dissimilar
taxa. Nature Communications 6, 8444. doi:10.1038/ncomms9444

I know that it isn't a direct comparison so it doesn't invalidate the authors claims, but perhaps
the current manuscript doesn't need to make this point about generality?

**Response – Is universality general? We agree with the reviewer that more research will be**
**needed for more complex ecosystems and at larger scale to conclude that ‘universality is likely**
**a general phenomenon for fungi’. We added to the manuscript a statement to this effect.**

**Revised lines 443-447 of the MS: “These comparisons suggest the existence of general**
**universal ecological dynamics from human-associated ecosystems to agricultural ecosystems,**
**and from bacteria to fungi and fungal guilds, although it must be evaluated by more studies**
**in more complex ecosystems and at larger scale”.**

**Response – Communities with high dissimilarity - The three references noted by the reviewer,**
**while solid, do not bear on our work for two reasons. First, Tedersoo et al. focus on forest**
**trees that are ectomycorrhizal while we focus on arbuscular mycorrhizal plants. We do**
**reference the Tedersoo et al. publication in another study we are publishing on**
**ectomycorrhizal forest ecosystems. Interestingly, we find that ectomycorrhizal fungi are the**
**one group that does not show universality, whereas arbuscular mycorrhizal, saprotrophic,**
**and plant pathogenic fungi, as well as bacteria, do show universality. Second, the authors of**
**the two other publications use outdated methods to identify fungi that cannot discriminate**
**below the level of family or order [Davidson et al. 2015 use DNA sequence of 18S rDNA (see**

Bruns and Taylor 2016, 10.1126/science.aad4228) and Powell et al. 2015 use the tRFLP
method]. As a result, neither work can be used to meaningfully assess universality.

I wonder whether the sampling approach (combining ten samples from each plot to produce
a composite sample) may lead to underestimation of stochastic processes. This would not
invalidate the comparisons among compartments and through time, of course, but could be
addressed in the manuscript where the sampling approach is briefly mentioned. I would also
appreciate a brief description in the supplementary methods of how roots, rhizosphere and
soil were sampled - currently there is only a citation to two other papers, I was able to access
one of them but the methods section is large and took a long time to download.

**Response – pooling samples.** As the reviewer notes, our pooling ten samples to make a
composite sample is conservative in that it can only underestimate the contribution of
stochastic process. Given that one of our major results is detection of stochasticity, having
found it with a conservative approach only strengthens our observation of stochasticity. We
added these statements to the supplementary in the revised ms.

**Revised** lines 38-41 of supplementary: *“Our pooling ten samples to make a composite sample*
*is conservative in that it can only underestimate the contribution of stochastic process. Given*
*that one of our major results is detection of stochasticity, having found it with a conservative*
*approach only strengthens our observation of stochasticity.”*

We now provide more details about the sampling of root, rhizosphere and soil in the
supplementary material.

**Revised** Lines 26-38 of the supplementary: *“Briefly, roots were removed from the ten plants,*
*mixed together, transferred to 50 ml tubes with detergent-phosphate buffer (6.33*
*NaH₂PO₄•H₂O and 8.5 g Na₂HPO₄•anhydrous in 1 L water, autoclaved; cooled, 200µl Silwet-*
*77 added; pre-cooled in ice-water mixture), and vortexed at full speed for 2 min¹. The roots*

*were removed from the tube, the liquid-filled tube was saved, and the roots were transferred*
*to a 200-ml plastic cup with phosphate buffer without detergent (6.33 NaH₂PO₄•H₂O and 8.5*
*g Na₂HPO₄•anhydrous in 1 L water, autoclaved; pre-cooled in ice-water mixture), vortexed at*
*full speed for 1 min twice, dried by clean paper towels, put into aluminum packet and frozen*
*in liquid nitrogen¹. The saved, liquid-filled tube containing the rhizosphere was centrifuged at*
*full speed for 3 min, the buffer discarded and the rhizosphere pellet frozen in liquid nitrogen¹.*
*Simultaneously, soil at 6" depth was collected adjacent to the ten sampled plants using 6" soil*
*collection tubes¹. Ten samples were mixed, transferred to a 50-ml centrifuge tube, and frozen*
*in liquid nitrogen¹."*

The paper is very well written and the methodological details are (mostly) described
thoroughly or appropriately cited. I've made a few suggestions in the attached annotated
manuscript. The supplementary materials make it very easy to see the outcomes of the data
analyses, including the Krona plots, so the work is transparent. However, to ensure
reproducibility, I would have appreciated a series of markdown files that demonstrate the
code used to perform analyses.

*Response – Markdown files. We thank the reviewer for the effort and positive feedback. We*
*now provide the R markdown files on the github.*

*Revised in lines 511-513: 'All scripts used in this study will be available at GitHub*
*(<https://github.com/ChengGaoBerkeley/EPICON.Mycobiome>). Note the reviewers can login*
*the site with the Account: chenggaomyco@gmail.com; Password: g1thubg1thub.'*

**The followings are the edits made directly on the annotated manuscript by the reviewer 1**
**Line 40** change 'fungal, functional guilds' to 'functional guilds of fungi'.

**This is the original text in question:** *underlying ecological dynamics of the mycobiomes of the*
*different sorghum compartments and to fungal, functional guilds.*

**Revision in lines 42 of MS:** *'underlying ecological dynamics of the mycobiomes of the different*
*sorghum compartments and to functional guilds of fungi.'*

The reviewer also suggested that we remove the comma in 'fungal, functional guilds' in line
113.

This is the **original** text in question: "...across fungal functional guilds..."

**Revision in line 116 of MS:** "...across *functional guilds of fungi*..."

**Line 40-41:** Not a strong way to conclude the abstract. What is the main conclusion of the
current study?

This is the **original** text in question: *'We hope that our high-resolution approach to research*
*community assembly, developed in an agricultural system, can be extrapolated to more*
*complex, natural ecosystems'*

**Response:** Our response is to delete the sentence. We realize that we give our main
conclusions before this final sentence so there is no need to repeat them or end with a weak
sentence. Better to just delete it.

**Line 46:** delete 'individual'. Maybe don't open the can of worms of what is an 'individual'
fungus.

This is the **original** text in question: *'In nature, the intimate, symbiotic association of individual*
*fungi with living plants is well appreciated, whether as endophytes, mycorrhizal partners or*
*parasites, as is their role as drivers of plant community structure'*

**Revision in lines 45-47 of MS:** *In nature, the intimate, symbiotic association of fungi with living*
*plants is well appreciated, whether as mycorrhizal partners or parasites, as is their role as*
*drivers of plant community structure'*

**Line 62:** This hasn't been developed enough. What are the mechanisms that underlie
community assembly? What is a stochastic process and what is a deterministic process? What
theory/empirical support is there for these processes driving assembly.

I see now that this information is provided below. Please restructure the introduction so that
the information is provided in a more logical order.

**Response - We agree with the reviewer and have moved the paragraph that explains the**
**forces of community assembly up to provide the information directly after first mentioning**
**community assembly.**

**Line 65:** 'aggressive sampling'? Unclear. Intensive?

**This is the original text in question:** '*sorghum [Sorghum bicolor (L.) Moench] grown in*
*homogenous soil with aggressive sampling from thrice-replicated plots for nearly all possible*'

**Revision in line 72 of MS:** '*sorghum [Sorghum bicolor (L.) Moench] grown in homogenous soil*
*with intensive sampling from thrice-replicated plots for nearly all possible*'

**Line 113** 'fungal, functional guilds', remove the comma

**This is the original text in question:** : " *...to fungal, functional guilds*'

**Response:** We agree with the reviewer

**Revision in line 116 of MS:** '*...to functional guilds of fungi.*'

**Line 118** 'over the ensuing, seventeen-week period', remove the comma

**This is the original text in question:** '*over the ensuing, seventeen-week period*'

**Response:** We agree with the reviewer

**Revision in line 121 of MS:** '*...over the ensuing seventeen-week period.*'

**Line 246** (last submitted pdf) or line 250 (edited docx): Any distance effect is likely to be
swamped by the time effect. Shouldn't the distance decay be estimated for comparisons
made within each timepoint?

**Line 298-299** (last submitted pdf) or lines 303-304 (edited docx): Please see above comment
regarding within timepoint comparisons (lines 16-27 of this document).

This is the **original** text in question in line 246: '*...and the near absence of any effect of*
*distance on community dissimilarity*'

This is the **original** text in question in Line 298-299: '*we detected no effect of geographic*
*distance on fungal community dissimilarity*'

Response: As suggested by the reviewer, to explore the influence of time on the detection of
distance effect, we re-analyzed the relationship between geographic distance and community
dissimilarity for each time point (**Fig. S15**). Overall, similar to our results of pooling all time
points as a whole, the null hypothesis of no relationship between geographic distance and
community dissimilarity, while taking into account multiple testing [$P < 0.003$ (0.05/17)],
cannot be rejected for any leaf, root or rhizosphere sample, or for 15 of 17 soil samples
(excepting weeks 12 and 16). Thus, although there is an association at two time points
between community dissimilarity and distance for post-flowering soil, this effect is seen at
least five weeks after our detection of stochasticity in leaves (weeks 1-7) and roots (weeks 1-
3).

Geographic distance (m)

Line 290: '| β_{NTI} | < 2 and |RCI| < 0.95, is strongly', remove the comma

This is the original text in question: '|RCI| < 0.95, is strongly, negatively correlated'

Response: We agree with the reviewer

Revision in lines 299 of MS: '| β_{NTI} | < 2 and |RCI| < 0.95 is strongly negatively correlated'

**Fig. 4 (A)** Why show the result of a linear relationship? Looks non-linear, the regression is
probably underestimating the explained variance.

This is the original figure 4 (A):

Response - We removed the fit for the curves as seen in the new figure (Fig. 3A), below.

Calculation of R in Mantel test does not assume a linear relationship, hence the very similar
R values between the original Fig. 4A and new Fig. 3A (differences due to random fluctuations
in rarefaction and permutation). By removing the line fit to the data, we hope that readers
will understand that no linear relationship is assumed. In the new version, to enhance the
reproducibility, a seed was set when generating random number. As also suggested by the
reviewer 1, we shared all markdown files in lines 511-513 of the manuscript.

**Fig. 4 (B)** It's unclear how rates of turnover are inferred. Is this just by eye, looking at where
timepoints start to cluster?

This is the original figure 4 (B)

**Response:** Yes, it is just by eye to see where timepoints start to cluster

**Reviewer #2 (Remarks to the Author):**

Comments to Authors:

The authors present an ambitious work detailing the temporal, compartmental, and genotype
 effects of the mycobiome across an entire growing season of Sorghum. The amount of work
 that went into this MS is quite admirable. They test theories on the nature of fungal assembly
 and specifically address the role community drift (stochasticity) plays in the mycobiome. The
 results presented are important, but the MS lacks focus and it is very hard to follow. A careful
 rewrite will aid the reader in the story, especially being careful to reduce extraneous language
 to streamline the MS.

**Response:** The different response of reviewers 1 and 2 highlights the variation in reader
 appreciation of our prose. Reviewer 1 wrote, “The paper is very well written and the
 methodological details are (mostly) described thoroughly or appropriately cited.” Reviewer 2
 writes, “. . . the MS lacks focus and it is very hard to follow.” To walk this tightrope, we have
 polished our manuscript further as noted below for each reviewer comment.

**Also, there are** a number of methodological and statistical concerns that should be addressed
 (outlined below). Particularly troubling is the inclusion of Beta-NTI analyses (which the
 author’s use numerous times to support their assertions). Beta-NTI requires a good tree to
 measures the average nearest taxon relationships. However, given that these were fungal

ITS2 data (and highly diverse at that), a good tree cannot be made (poorly alignable data). So,
how can the authors' justify this method? There is not enough detail in the MS to glean if they
found a work around to this or even any methods detailing how they generated the tree (even
in the supplemental methods). This lack of detail and the inhering inappropriateness of the
measure for ITS data casts doubt on these data. This reviewer suggests eliminating all beta-
NTI references (focus on Beta RC instead) or VERY CAREFULLY alleviate my concerns. This
work is of great interest and would appeal to the readership of Nature Communications, but
only is acceptable if appropriate methodological approaches are implemented.

**Response: We deeply appreciate this criticism because it forced us to challenge our results**
**thoroughly with two new analytic strategies, both of which we are pleased to report, resulted**
**in consistent conclusions that mirrored our original result. In the first strategy, we eliminated**
**beta-NTI references and only focused on RCI (as suggested by the reviewer). In the second**
**strategy, we calculated beta-NTI using a strongly-supported 18S+28S phylogenetic tree, in this**
**case a 18S+28S tree built from the taxonomy of our ITS OTUs using the taxonomy_to_tree.pl**
**script of Tedersoo et al 2018. We compared the two new analyses to our original calculation**
**of beta-NTI using an ITS2 tree. All three analyses returned similar results, as depicted by the**
**following two figures. We prefer to present the most thorough analysis suggested by the**
**reviewer, that is, RCI and beta-NTI using the 18S+28S phylogeny in the revised manuscript**
**(Fig. 5, Figs. S21, S25).**

**The following figure showing stochasticity** is detected from three strategies: (A) Raup-Crick
index (RCI) alone, (B) RCI and beta NTI (β NTI) calculated from ITS2 OTUs and a phylogeny
based on 18S+28S rDNA sequence, and (C) RCI and β NTI calculated from ITS2 OTUs and a
phylogeny based on the same region. Where (A) $|RCI| < 0.95$ alone or (B-C) $|RCI| < 0.95$ and
$|\beta NTI| < 2$ (red bars), compositional variance is most likely due to stochasticity, which in this

range is independent from dispersal and selection. Note that stochasticity is dominant in early
 time points in leaves and roots.

The following figure showing ecological drift and community size (abundance) or richness.

Ecological drift can be evidenced when the stochastic component of compositional variance

is negatively correlated with community size (abundance). Here, we demonstrate this

correlation when stochasticity is estimated in three different ways: $|RCI| < 0.95$ alone, $|RCI| <$

0.95 and $|\beta NTI| < 2$ where βNTI was calculated from ITS2 OTUs using a phylogeny based on

18S+28S rDNA sequence, and $|RCI| < 0.95$ and $|\beta NTI| < 2$ where βNTI was calculated from

ITS2 OTUs and a phylogeny using the same region. Community size (**abundance**) is estimated

in three different ways: fungal rDNA, internal transcribed spacer (ITS2) reads as a percentage

of total ITS2 reads amplified from fungal plus sorghum host DNA, fungal 18S rDNA amount
 assessed by real time or quantitative PCR amplification and fungal RNA as a percentage of
 fungal plus plant reads found in the transcriptomes of sorghum leaves and roots. Ecological
 drift is also evidenced by a positive correlation between fungal richness and the stochastic
 component of compositional variance, as shown here. Note that R^2 values are not drastically
 affected by the method of analysis.

 The following figure (Fig. 5) showing the results of RCI and beta-NTI using the 18S+28S
 phylotaxy presented in the revised MS.

Abstract

**Line 40:** Comma seems out of place (“fungal, functional”)

**This is the original text in question:** ‘...underlying ecological dynamics of the mycobiomes of
 the different sorghum compartments and to fungal, functional guilds.’

Response: We agree with the reviewer

**Revision in line 42:** ‘...to functional guilds of fungi.’

Introduction:

**Lines 46-47:** It is unclear how endophytes (which by the most definitions are generally
 commensal) might drive plant community structure, at least in the same vein as mycorrhizae
 or parasites/pathogens. Plant communities more likely drive endophytes rather than the
 reverse. So, I am not sure how well “it is well appreciated.”

**This is the original text in question:** ‘In nature, the intimate, symbiotic association of individual
 fungi with living plants is well appreciated, whether as endophytes, mycorrhizal partners or
 parasites.’

**Response:** We agree with the reviewer that it is unclear how endophytes might drive plant
 community structure. We remove ‘endophytes’ from this sentence.

**Revision in line 46:** '*...whether as mycorrhizal partners or parasites...*'.

**Line 49:** Is "highlighted in the last decade" to reductionist? This has been a major study system
for a long time.

**This is the original text in question:** '*...their role in ecosystem carbon and nitrogen cycling has
been highlighted in the last decade*'

**Response:** We agree with the reviewer that it is reductionist to state '*highlighted in the last
decade*'. We now remove "*in the last decade*".

**Revision in line 48:** '*...their role in ecosystem carbon and nitrogen cycling has been highlighted.*

**Lines 49-51:** Not sure this is a fair statement. One could easily argue that studies on fungi are
more abundant than bacteria given the vast literature on mycorrhizal associations. This may
be true for plant endophyte work (but I suspect not), but this statement says all studies of
fungi-plant connections. Please omit or change.

**This is the original text in question:** '*In contrast, studies of the communities of fungi associated
with plants are few and far fewer than those of bacteria.*'

**Response:** We write that studies of communities of fungi are fewer than studies of
communities of bacteria, which we think is accurate. Studies of mycorrhizae are often on one
mycorrhizal fungus guild and not of total fungal communities. We make this point clearer by
adding the word 'total'.

**Revision in line 48-50:** '*In contrast, studies of the total communities of fungi associated with
plants are few and far fewer than those of total communities of bacteria*'.

**Lines 51-53:** This makes little sense.

**Response:** We guess that the reviewer refers to the text identified below. To our mind, it does
make sense. We have rewritten it as two, shorter sentences.

**This is the original text in question:** *'Here we tackle this gap in knowledge of fungal*
*communities associated with plants by adding the fungal mycobiome to studies of the*
*bacterial microbiome using an approach that accounts for the variation in plant compartment*
*(leaf, root, rhizosphere and soil), plant development (from seedling emergence to grain*
*maturation), physical environment (irrigation and drought), and host genotype (drought*
*response to either retain or suppress photosynthesis) as summarized in Tables S1 and S2.'*

**Revision in line 51-57:** *"Here we tackle this gap in knowledge of fungal communities*
*associated with plants by adding the fungal mycobiome to studies of the bacterial*
*microbiome^{10,11}. We use an approach that accounts for the variation in plant compartment*
*(leaf, root, rhizosphere and soil), plant development (from seedling emergence to grain*
*maturation), physical environment (irrigation and drought), and host genotype (drought*
*response to either retain or suppress photosynthesis) as summarized in Tables S1 and S2."*

**Lines 57-59:** This is making some assumptions that may not be justified. We do not know (or
least you do not present any evidence that we do know) that increased drought will affect
plants via fungal endophyte-mediated plant responses negatively. One could easily argue that
endophytes may confer drought resistance.

**This is the original text in question:** *'We chose drought as the key environmental variable for*
*our study because it will be a defining feature of this century, and will not only directly affect*
*plants and their fungal communities, but also indirectly affect plants through changes in the*
*fungal community. Positive, indirect effects of fungal communities on plants could be*
*harnessed by modern agriculture, but only if the mechanisms of community assembly are*
*understood.'*

Response: We do not write that effects could be negative, although they could be. We do
write that the effects can be positive, as the reviewer argues. We prefer to keep this sentence
as is.

**Lines 59-61:** This is a fallacy. It is very possible to harness plant-associated fungi to aid modern
agriculture without understanding assembly. If you see that one fungus is beneficial and
enriched in a particular treatment, it can be harnessed and utilized for a function, without
understanding the mechanisms by which it was assembled.

This is the **original text in question:** *'Positive, indirect effects of fungal communities on plants*
*could be harnessed by modern agriculture, but only if the mechanisms of community assembly*
*are understood.'*

Response: We are aware of many studies concerning the addition of arbuscular mycorrhizal
fungal inoculum to crops. However, we are not aware of studies where it was shown that the
added fungus altered the community or even persisted throughout the experiment.

We consulted a recent review article on AM Fungi as biofertilizers, which reveals this lack of
research, (Berruti, A., Lumini, E., Balestrini, R., & Bianciotto, V. (2016). Arbuscular mycorrhizal fungi as natural
biofertilizers: let's benefit from past successes. *Frontiers in microbiology*, 6, 1559.) In it, we find
supplementary table 2 which lists 164 studies. Limiting these studies to field studies reduces
the number to 39 studies. Limiting further to studies where differences in roots shoots and
yield were significant following addition of AMF inoculum, two studies remain. We were
unable to locate one of the two studies, Li et al. 2013, *Biology and Fertility of Soils*, neither in the
references of Berruti et al. nor in the references to the supplementary material. In the second
study, Gaur, A., & Adholeya, A. (2002). Arbuscular-mycorrhizal inoculation of five tropical fodder crops and
inoculum production in marginal soil amended with organic matter. *Biology and Fertility of Soils*, 35(3), 214-
218), we find that the authors did not measure yield, as claimed in Supplementary Table 2 in

Berruti et al. 2016. Thus, we still know of no field study where the community of AMF was
assessed prior to addition of AM fungal inoculum, during crop growth or after harvest.
Nevertheless, we have altered this sentence by adding the word “best.”

**Revision** in lines 60-61: *‘Positive, indirect effects of fungal communities on plants could best*
*be harnessed by modern agriculture if the mechanisms of community assembly were better*
*understood.’*

**Line 64:** Nearly all possible combinations? Why nearly?

**This is the original text in question:** *‘...homogenous soil with aggressive sampling from thrice-*
*replicated plots for nearly all possible...’.*

**Response:** We agree with the reviewer to remove the word ‘nearly’.

**Revision in line 72:** *‘homogenous soil with intensive sampling from thrice-replicated plots for*
*all possible’*

Lines 66-68: Seem irrelevant and unneeded.

**This is the original text in question:** *‘Our study complements recent reports from the same*
*agricultural system on the bacterial microbiome (756 samples) ¹³ and the arbuscular*
*mycorrhizal (AM) mycobiome (312 samples) ^{14’}*

**Response:** We believe that we should include this sentence because data based on the same
DNA samples examined by us have been published by us for AM fungi and bacteria. We use
the AM fungal results in this manuscript, and refer to results from the bacterial study.

**Lines 70-72:** Is it fair to say that selection is wholly deterministic, and drift is wholly stochastic.
Largely, sure. Wholly? Perhaps not. This may be a bit reductionist.

**This is the original text in question:** *‘While selection is wholly deterministic, both dispersal and*
*diversification have stochastic and deterministic components, and drift is wholly stochastic.’*

Response: By definition in the field of microbial community assembly, selection is wholly
deterministic and drift is wholly stochastic. These definitions were published in the recent
review of stochastic community assembly (Zhou J, Ning D. 2017. Stochastic community assembly: Does
it matter in microbial ecology? *Microbiol Mol Biol R* 81, e00002-00017). The four forces are selection,
dispersal, divergence and drift. Selection is defined as wholly deterministic, drift as wholly
stochastic, and the other two as a blend of both selection and determinism. We feel that we
cannot change the definitions published in MMBR because doing so would confuse readers.

**Lines 69-76:** This paragraph could be rewritten for clarity.

**This is the original text in question:** *'The assembly of communities rests on the activity of the*
*four processes that influence constituent species: selection, drift, diversification and dispersal.*
*While selection is wholly deterministic, both dispersal and diversification have stochastic and*
*deterministic components, and drift is wholly stochastic. Of the four processes, drift, the*
*stochastic extinction caused by random species abundance fluctuation, is the most difficult to*
*demonstrate because one must first rule out the others, two of which have stochastic*
*components and only one of which—the evolutionary process of diversification—can*
*reasonably be ignored for fungi over a period as short as a season.'*

**Response:** We have rearranged the paragraph to improve its clarity.

**Revision in lines 62-69:** *'The assembly of communities rests on the activity of the four*
*processes that influence constituent species: selection, drift, diversification and dispersal*¹⁶.
*Selection is wholly deterministic and drift is wholly stochastic*¹⁷, *but both dispersal and*
*diversification have stochastic and deterministic components*¹⁷. *Of the four processes, drift,*
*the stochastic extinction caused by random species abundance fluctuation*¹⁸, *is the most*
*difficult to demonstrate because one must first rule out the other three processes, two of*

which have stochastic components, *although one of these*—the evolutionary process of
*diversification—can reasonably be ignored for fungi over a period as short as a season.'*

**Lines 77-85:** Sentence structure here and throughout is a bit odd. Recommend using more
direct statements and shy away from over-comma use. This will do much to clarify the MS.

**This is the original text in question:** *'In previous studies, ecological drift has been estimated*
*as the 'unexplained' compositional variation in ecological model fitting, or as the dispersion*
*of beta diversity (also termed the compositional variance) ¹⁷. However, neither of these*
*measures necessarily represent the consequences of ecological drift due to possible*
*undermeasurement of the processes of selection, dispersal and diversification ¹⁷. Despite the*
*difficulty of its demonstration, drift is thought to strongly influence community assembly when*
*(i) communities are small or when (ii) they have recently been released from selection imposed*
*by stress such as drought ^{18, 19}; and due to this difficulty, demonstration of drift in these two*
*situations in nature remains rare ^{16, 20, 21, 22}.'*

**Response:** We have revised this paragraph to improve its clarity.

**Revision** in line 78-86: *'Ecological drift is estimated in ecological model fitting as the*
*'unexplained' compositional variation, or is estimated from empirical data as the dispersion*
*of beta diversity (an approach also termed compositional variance) ¹⁸. However, neither of*
*these methods of estimation necessarily represent the consequences of ecological drift due to*
*possible undermeasurement of the processes of selection, dispersal and diversification ¹⁸. Drift*
*is thought to most strongly influence community assembly when (i) communities are small or*
*when (ii) they have recently been released from selection imposed by a stress such as drought*
*^{19, 20}; However, due to the difficulty of detecting drift, even in these two situations its*
*demonstration in nature remains rare ^{17, 21, 22, 23}.'*

**Line 86-88:** Why do we think that these communities will be small during early development.

I am not saying they will not be, but it would be useful to provide justification to this.

**This is the original text in question:** *'We test two hypotheses concerning fungal communities*
*of sorghum plants; H1, that drift will be important when fungal communities are small, as*
*expected early in the development of sorghum plants'*

**Response:** We think that it is reasonable to assume that newly emerged roots and leaves will
have few or no microbes in their tissues or on their surfaces. It has been a consensus in the
study of microbial succession that the sizes of early communities are small (Fierer et al 2010
*Research in Microbiology: 10.1016/j.resmic.2010.06.002*).

**Revision in lines 87-90:** *'We test two hypotheses concerning fungal communities of sorghum*
*plants; H₁, that drift will be important when fungal communities are small, as expected early*
*in the development of sorghum plants when microbes should be rare on newly formed roots*
*and leaves*²⁴*'.*

**Lines 90-92:** "when stress ... was relieved"- This makes little sense. Clarify please.

**This is the original text in question:** *'To test these hypotheses we characterized, in the field,*
*fungal communities of sorghum plants from their emergence as seedlings to senescence under*
*conditions of regular irrigation and when stress imposed by pre-flowering drought was*
*relieved.'*

**Response:** We have revised the sentence to improve clarity.

**Revisions in lines 91-94:** *'To test these hypotheses we characterized fungal communities*
*associated with sorghum plants growing in the field from seedling emergence to plant*
*senescence under conditions of regular irrigation and also when stress imposed by pre-*
*flowering drought was relieved by resuming irrigation.*

**Lines 92-95:** Please clarify this sentence. The words “negative correlation between the
strength of stochasticity on one hand and community size on the other” makes little sense
out of context. This would benefit from a rewrite.

**This is the original text in question:** *‘Here, as hypothesized in H_1 , we present evidence*
*demonstrating a significant role for drift early in fungal community assembly by*
*demonstrating a negative correlation between the strength of stochasticity on one hand and*
*community size on the other.’*

**Response:** We have revised the sentence to improve clarity.

**Revision in lines 95-97:** *‘We present evidence that demonstrates, as hypothesized (H_1), a*
*significant role for drift early in fungal community assembly by observing a negative*
*correlation between the strength of stochasticity on one hand and community size on the*
*other.’*

**Lines 104-107:** I would tread carefully here, while it may be true that this MS is the first to use
this method on non-AM systems to investigate ‘universality’, you are far from being the first
to investigate the universality of fungal assembly rules. As written (though I know this is not
what you are trying to say) it implies that you are the first to try this. Folks have been looking
into universality of fungal assembly since the time of Clements. Choose your words carefully.

**This is the original text in question:** *‘To our knowledge, universality has been the subject of*
*only two studies of fungal mycobiomes, both of them AM fungal communities that showed*
*both universal and unique ecological dynamics depending on ecosystem type (nature v.*
*agricultural) and phosphorus availability (low v. high)^{25, 26}.’*

**Response:** We wholeheartedly agree that we are not the first to investigate universality of
fungal assembly rules. Here the universality is tested for population dynamics across
communities. The universality of population dynamics has only been recently addressable

due to the invention of DOC method (Bashan et al 2016 Nature). The DOC method has not
been used outside the fields of bacteria and AM fungi, to our knowledge.

**Revision** in lines 107-111: *'Although many attempts have been made to assess the universality*
*of microbial community assembly, assessment of universality by the DOC method has been*
*applied to just two types of fungal mycobiomes, both of which are AM fungal communities.*
*These studies found both universal and unique ecological dynamics depending on ecosystem*
*type (nature v. agricultural) and phosphorus availability (low v. high)^{25, 26.}'*

**Line 108-109:** Is this novel? Of course, fungal communities will be more similar when shared
number of taxa increases. This is the whole point of most dissimilarity measures that
ecologists have been using since the 1950s. Perhaps you are meaning this differently that is
written. If so, clarify please.

**This is the original text in question:** *'Here, where DOC analysis shows an increased similarity*
*among fungal communities as the fraction of shared taxa increases, we find that . . .*

**Response:** What is novel about the DOC approach is that the calculation of similarity is based
solely on the shared taxa (Bashan et al. 2016 Nature), whereas the traditional measures of
similarity include all taxa, including those that are unique to each sample. It is not necessarily
the case that dissimilarity will drop as the proportion of shared taxa increases, because
dissimilarity involves relative abundance of taxa (Bashan et al. 2016 Nature). Where
dissimilarity does drop as the proportion of shared taxa increases, universality can be claimed
only if this reduction is supported by a preponderance of the pairwise comparisons (Bashan
et al. 2016 Nature).

**Revision in lines 111-114:** *'Here, where DOC analysis shows a reduced dissimilarity among*
*fungal communities as the fraction of shared taxa increases and where this reduced*
*dissimilarity is supported by the bulk of all possible pairwise comparisons, we find . . .*

**Lines 117-119:** This makes little sense. How can one take leave samples from soil? Please re-
write to be explicit about what you did here.

**This is the original text in question:** *'From samples of soil, rhizosphere, roots and leaves taken*
*from a field that had never previously experienced sorghum and most recently had been*
*planted to oats, we recognized 1070 fungal operational taxonomic units (OTUs)'*

**Response:** We believe that our sentence did not imply that leaves were taken from soil, rather
than they were taken from a field. We feel that readers will understand that roots and leaves
were taken from plants.

**Lines 115-127:** This paragraph is oddly structured.

**This is the original text in question:** *'First, we characterized overall fungal communities in an*
*agricultural field of sorghum from samples taken before sorghum seed was sown and over the*
*ensuing, seventeen-week period from seedling emergence to grain maturation (Fig. 1). From*
*samples of soil, rhizosphere, roots and leaves taken from a field that had never previously*
*experienced sorghum and most recently had been planted to oats, we recognized 1070 fungal*
*operational taxonomic units (OTUs) (Table S3) detected from DNA sequence of fungal internal*
*transcribed spacer 2 (ITS2) amplified by dual-barcoded, fungal specific primers, and sequenced*
*by Illumina Miseq (34 541 758 reads; Fig. S1). The fungal communities were composed,*
*phylogenetically, of Ascomycota, Basidiomycota, Chytridiomycota, Mucoromycota*
*(Glomeromycotina, Mortierellomycotina and Mucoromycotina) and Zoopagomycota*
*(Entomophthoromycotina, Kickxellomycotina and Zoopagomycotina), and composed,*
*functionally, of filamentous fungi that grow invasively and function as arbuscular*
*mycorrhizae, endophytes, plant pathogens, and saprobes, as well as yeasts (Fig. S2-S3).'*

**Response:** We have revised the paragraph to improve clarity

**Revision** in lines 120-133: *'First, we characterized overall fungal communities in an*
*agricultural field of sorghum from samples taken before sorghum seed was sown and over the*
*ensuing seventeen-week period from seedling emergence to grain maturation (Fig. 1). We*
*characterized fungal operational taxonomic units (OTUs) from samples of soil, rhizosphere,*
*roots and leaves taken from a field that had never previously experienced sorghum and most*
*recently had been planted to oats. In total, we recognized 1070 OTUs (Table S3) detected from*
*DNA sequence of fungal internal transcribed spacer 2 (ITS2) amplified by dual-barcoded,*
*fungal specific 5.8SFun and ITS4Fun primers²⁹, and sequenced by Illumina Miseq (34 541 758*
*reads; Fig. S1). The fungal communities were composed, phylogenetically, of Ascomycota,*
*Basidiomycota, Chytridiomycota, Mucoromycota (Glomeromycotina, Mortierellomycotina*
*and Mucoromycotina) and Zoopagomycota (Entomophthoromycotina, Kickxellomycotina and*
*Zoopagomycotina), and composed, functionally, of filamentous fungi that grow invasively and*
*function as arbuscular mycorrhizae, endophytes, plant pathogens, and saprobes, as well as*
*yeasts (Fig. S2-S3).'*

**Lines 122-127:** Should this be included here. This is just about what you would expect to find
in any ag field. I would omit this here and expand the methods above.

**This is the original text in question:** *'The fungal communities were composed,*
*phylogenetically, of Ascomycota, Basidiomycota, Chytridiomycota, Mucoromycota*
*(Glomeromycotina, Mortierellomycotina and Mucoromycotina) and Zoopagomycota*
*(Entomophthoromycotina, Kickxellomycotina and Zoopagomycotina), and composed,*
*functionally, of filamentous fungi that grow invasively and function as arbuscular*
*mycorrhizae, endophytes, plant pathogens, and saprobes, as well as yeasts (Fig. S2-S3)'*

Response: We disagree. The methods are published and routine and need not be expanded
 in the main text. Our results, however, are specific to our study and can only be obtained in
 our report. We prefer to keep the results.

**Lines 128-144:** Did the authors investigate any space x time interactions? Or, perhaps more
 interestingly, time within a particular “space” to investigate temporal effects within roots for
 examples. By doing two sets of analyses (whole model and individualized compartment
 models), the authors will allow for a more robust way to explore fungal community dynamics.
 But including the compartment (when one would naturally expect little overlap) in all
 analyses, that may parse so much variance out that the more subtle signals from other factors
 may be occluded.

Response: We conducted and now provide (Table S4) results of our permanova analysis of (i)
 all samples including the interactions, and (ii) samples of each compartment. Note that, for
 all samples, compartment (leaf, root, rhizosphere, or soil) still explains most of the variance
 ($R^2=0.434$) with sampling time ($R^2=0.061$) and irrigation treatment ($R^2=0.02$) well behind. No
 interaction explains as much variance as does time point, and those that explain more than
 irrigation treatment involve compartment.

Table S4 Permutational analysis of variance (PERM ANOVA) showing association of fungal community composition with compartment, time point, drought treatment, sorghum cultivar and their interactions

Samples	Variables	Df	Sums Of Sqs	Mean Sqs	F	R2	P
All samples	Plant compartment (Co)	3	106.596	35.532	370.320	0.434	0.001
	Time point (TP)	1	15.098	15.098	157.360	0.061	0.001
	Drought treatment (DT)	2	4.849	2.424	25.270	0.020	0.001
	Plant cultivar (Cu)	1	0.574	0.574	5.980	0.002	0.001
	Co:TP	3	11.107	3.702	38.590	0.045	0.001
	Co:DT	6	5.826	0.971	10.120	0.024	0.001
	TP:DT	2	1.396	0.698	7.280	0.006	0.001
	Co:Cu	3	0.915	0.305	3.180	0.004	0.001
	TP:Cu	1	0.305	0.305	3.180	0.001	0.011
	DT:Cu	2	0.383	0.192	2.000	0.002	0.029
	Co:TP:DT	6	2.660	0.443	4.620	0.011	0.001

	Co:TP:Cu	3	0.547	0.182	1.900	0.002	0.024
	Co:DT:Cu	6	0.695	0.116	1.210	0.003	0.194
	TP:DT:Cu	2	0.293	0.146	1.520	0.001	0.105
	Co:TP:DT:Cu	6	0.595	0.099	1.030	0.002	0.386
	Residuals	978	93.839	0.096		0.382	
Leaf	TP	1	10.448	10.448	108.555	0.287	0.001
	DT	2	0.943	0.472	4.899	0.026	0.001
	Cu	1	0.185	0.185	1.924	0.005	0.067
	TP:DT	2	1.073	0.537	5.575	0.030	0.001
	TP:Cu	1	0.135	0.135	1.401	0.004	0.155
	DT:Cu	2	0.229	0.115	1.192	0.006	0.253
	TP:DT:Cu	2	0.243	0.122	1.262	0.007	0.226
	Residuals	240	23.099	0.096		0.635	
Root	TP	1	5.642	5.642	69.293	0.169	0.001
	DT	2	5.281	2.640	32.427	0.158	0.001
	Cu	1	0.572	0.572	7.026	0.017	0.001
	TP:DT	2	1.586	0.793	9.741	0.048	0.001
	TP:Cu	1	0.309	0.309	3.791	0.009	0.001
	DT:Cu	2	0.257	0.129	1.579	0.008	0.059
	TP:DT:Cu	2	0.197	0.098	1.207	0.006	0.182
	Residuals	240	19.541	0.081		0.585	
Rhizosphere	TP	1	7.870	7.870	78.341	0.207	0.001
	DT	2	3.801	1.900	18.915	0.100	0.001
	Cu	1	0.543	0.543	5.408	0.014	0.001
	TP:DT	2	0.991	0.496	4.933	0.026	0.001
	TP:Cu	1	0.253	0.253	2.521	0.007	0.007
	DT:Cu	2	0.294	0.147	1.461	0.008	0.079
	TP:DT:Cu	2	0.235	0.118	1.170	0.006	0.217
	Residuals	240	24.111	0.101		0.633	
Soil	TP	1	2.215	2.215	21.100	0.071	0.001
	DT	2	0.680	0.340	3.239	0.022	0.001
	Cu	1	0.188	0.188	1.794	0.006	0.005
	TP:DT	2	0.406	0.203	1.934	0.013	0.001
	TP:Cu	1	0.157	0.157	1.497	0.005	0.033
	DT:Cu	2	0.297	0.149	1.415	0.010	0.015
	TP:DT:Cu	2	0.213	0.107	1.015	0.007	0.401
	Residuals	258	27.088	0.105		0.867	

**Lines 149-152:** Not sure if this is the most appropriate way to do this. Technically you are not
supposed to run Mantel tests on non-distance data (BC is not true distance) but I understand
that this is often done anyways. I would recommend either: 1) simply regressing BC values
against time (weeks difference) to look for temporal shifts, or 2) using a Procrustes approach

where you compare soil vs. leaf (and every other combination) and regressing the Procrustes
Associated Metric over time to ask which compartment is most responsive over time. Also
looking at a full interaction permanova would get at this as well.

**This is the original text in question:** *'Subsequent temporal change in community composition,*
*measured using a Mantel test of Bray-Curtis community dissimilarity, was seen throughout*
*the ensuing 17 weeks and was strongest in leaf, strong in both root and rhizosphere and*
*weakest in soil (Fig. 4A; Fig. S7).'*

**Response:** Now, we simply plot the Bray-Curtis dissimilarity and temporal distance, and no
longer fitting a curve (see lines 163 to 174 in this document). However, as noted by the
reviewer, it is common practice to apply the Mantel test, so we would prefer to report the R
value to invite comparison of our results with those of other studies.

**We also provided a full interaction permanova as described in lines 516-528 of this document**
**Lines 157-158:** Is this fair? It might not be assumed to be a plant driven community shift, but
a propagule availability issue. This is an equally parsimonious explanation.

**This is the original text in question:** *'Given that the developing sorghum plant is involved in*
*driving these temporal changes in community composition'*

**Response:** As we discuss later in our manuscript (lines 312-315), and as we responded to
Reviewer 1 above (lines 139-158 of this document), in our homogenous agricultural
ecosystem, propagule availability is not likely to drive this succession pattern over just 17
552 weeks and not in comparison to the developing plant, which is providing all of the energy for
all of the fungi.

**Lines 156-168:** I Like this, but object to the phrase "stable community" here. Maybe replace
it with "locally stable" or "temporarily stable"

**This is the original text in question:** *'The length of time that it takes for this effect to reach a*
*stable community can be used to gauge the strength and timing of sorghum's influence (Figs*
*4B; Fig. S8-S9).'*

Response: We agree with the reviewer

**Revision in line 165** *'The length of time that it takes for this effect to reach a temporarily stable*
*community can be used to gauge the strength and timing of sorghum's influence (Figs 4B; Fig.*
*S8-S9).'*

**Lines 167:** This idea that rhizospheric communities have little turnover even in very early
development casts doubt on previous statements that these communities are small,
therefore open to drift (H1). I believe you but be careful with language.

**This is the original text in question:** *'The situation in rhizosphere and soil is different with little*
*turnover from weeks 1 through 7 or 9, and again from weeks 13 through 17, the two periods*
*of relative stability bridged by one of turnover (Figs 4B; Fig. S8-S9).'*

**Response:** In line 166-167 we shift from discussion of leaves (in which communities are small)
to discussion of soil and rhizosphere (where communities are large). We now clarify that shift.

**Revision in lines 173-175:** *'In contrast to leaves and roots, the situation in rhizosphere and soil*
*is different with little turnover from weeks 1 through 7 or 9, and again from weeks 13 through*
*17, the two periods of relative stability bridged by one of turnover (Figs 3B; Fig. S8-S9).'*

**Lines 169-186:** This needs work. I like what you are getting at, but the writing is almost not
linear. It is very difficult to interpret. Clean this up and make the point for the reader. What
does this mean with respect to these communities.

**This is the original text in question:** *'We also found evidence for turnover using a second*
*analytical approach, threshold indicator taxa analysis (TITAN), which is based on increases*
*(z+) or decreases (z-) in the abundance of indicator fungal OTUs 31. With the leaf mycobiome,*

*our results showed that indicator fungal OTUs showed both z+ and z- beginning in the 7th to*
*8th weeks which largely ceased by the 9th week (Fig. 4C; Fig. S10). In roots, indicator fungal*
*OTUs showed significant z+ and z- from the third week until the 12th week, while in the*
*rhizosphere obvious temporal turnover did not begin until the 8th week and continued until*
*the 12th week (Fig. 4C). The five week delay in the initiation of turnover of rhizosphere*
*compared to roots was not seen in the bacterial or AM fungal communities of the same*
*sorghum plants, where the bacterial microbiome of both root and rhizosphere stabilized after*
*the 6th week 13, and AM fungal communities of both root and rhizosphere continued to turn*
*over from the 1st week to the 17 week. Having compared our analysis of total fungal*
*communities with those previously made from the same samples for bacterial and AM fungal*
*communities, we now turn to results from unrelated studies of bacteria. In a study of rice, the*
*bacterial community turnover started early (1st week) and stabilized after 8 or 9 weeks, when*
*vegetative plant growth ceased, again with root and rhizosphere following the same temporal*
*pattern. Why turnover should cease earlier for bacterial microbiomes than fungal mycobiomes*
*is not clear, although the better taxonomic precision for fungal ITS compared to bacterial 16S*
*may play a role.*

**Response:** We revised by beginning sentences with compartments to make the text more
straightforward.

**Revision in lines 176-193:** *'We also found evidence for turnover using a second analytical*
*approach, threshold indicator taxa analysis (TITAN), which is based on increases (z+) or*
*decreases (z-) in the abundance of indicator fungal OTUs³³. With the leaf mycobiome, our*
*results showed that indicator fungal OTUs showed both z+ and z- beginning in the 7th to 8th*
*weeks which largely ceased by the 9th week (Fig. 3C; Fig. S10). In roots, indicator fungal OTUs*
*showed significant z+ and z- from the third week until the 12th week, while in the rhizosphere*

*obvious temporal turnover did not begin until the 8th week and continued until the 12th week*
*(Fig. 3C). In published analyses of bacterial and AM fungal communities of these same*
*sorghum plants, the five-week delay in the initiation of turnover of rhizosphere compared to*
*roots was not seen in and, instead, the bacterial microbiome of both root and rhizosphere*
*stabilized after the 6th week ¹⁰, and AM fungal communities of both root and rhizosphere*
*continued to turn over from the 1st week to the 17th week ¹¹. Turning to studies of bacteria in*
*other systems, compared to our fungal results, in a bacterial community associated with rice,*
*turnover started early (1st week) and stabilized after 8 or 9 weeks when vegetative plant*
*growth had ceased, again with root and rhizosphere following the same temporal pattern ³⁴.*
*Why turnover should cease earlier for bacterial microbiomes than fungal mycobiomes is not*
*clear, although the better taxonomic precision for fungal ITS compared to bacterial 16S may*
*play a role.'*

**Line 195:** I would argue that “compositional variance” is not the same and “dissimilarity
values”

**This is the original text in question:** ‘...metrics also showed higher compositional variance
(pairwise dissimilarity values) in early than...’

**Response:** We do not mean that compositional variance is the same as dissimilarity values,
rather that dissimilarity values are used to assess compositional variance.

**Revision in line 203-204:** ‘...higher compositional variance (assessed from pairwise
dissimilarity values)...’

**Lines 208-215:** I like the acknowledgement here, but I wonder how much this adds to the MS.
I suggest reducing this paragraph and folding it into a different heading, I think it is not
warranted to have a whole section.

This is the **original text in question**: 'While acknowledging that comparison among studies is
never simple, comparison of our results with those two studies may help explain the more than
two-fold difference in *Populus* compartment effects. The strong compartment effects seen in
our study and the first of the *Populus* studies rely, in part, on an abundance of yeasts in leaves,
which was smaller in the second *Populus* study (Fig. S2). The gap between the very strong
compartment effect in the first *Populus* study and our results may be explained by the low
overlap of EM fungi OTUs in *Populus* roots and soil compared to the greater overlap of AM
fungi in these two compartments in sorghum.'

**Response**: We would prefer to keep this comparison here because these *Populus* studies are
about the only studies that characterize fungi associated with plants as thoroughly as our
study.

**Line 217**: Genetical? This is very odd wording. You have used genotype throughout, it should
probably be consistent.

This is the **original text in question**: 'Host effect on fungal community along the genetical
dimension.'

**Response**: Agreed.

**Revision in line 226**: 'Effect of plant genotype on fungal community composition.'

**Lines 220-223**: This might be an issue on significant but unimportant (given the R-square
values). However, this might be an issue where reanalysis as suggested above might give you
more power. Also, what evidence are you using to make the claim that this genotype effect
was due to these two OTUs? Just because they are differentially abundant, does not
necessarily mean that these are the full drivers of this genotype effect.

This is the **original text in question**: '...(R² = 0.021, P < 0.001) but not in leaf (R² = 0.004, P =
0.284) (Fig. 5). This host genotype effect in sorghum roots was due to the presence of two

pathogens (*OTU19_Sarcocladium* and *OTU20_Monosporascus*) and one saprotroph
(*OTU34_Achroistachys*), all of which were significantly more abundant in the roots of
sorghum cultivar BTx642 than in the roots of cultivar RTx430'

**Response:** Our analyses are for each compartment, alone, as suggested by the reviewer in the
comments regarding lines 128-144 (lines 516-528 of this document). In this case, we analyze
root communities and base the genotype effect on four OTUs, not just two. As these four are
the only OTUs showing significantly different abundance between the two genotypes, they
constitute the entire effect. We do not know how the host influences the different abundance
of the four OTUs, but as the host genotype is the only variable, it seems safe to ascribe the
effect to host genotype.

**Lines 229-232:** Again, a reanalysis may give you more power to make these claims.

**This is the original text in question:** 'Although we found no effect of host genotype on leaf
fungal communities when considering all 17 time points as a whole ($R^2 = 0.004$, $P > 0.05$), we
did find a significant effect of host genotype on leaf fungal communities for the subset of
665 weeks 10-17 ($R^2 = 0.038$, $P = 0.002$; Fig. 5A).'

**Response:** Again, we had analyzed our data using one compartment at a time, as suggested
by the reviewer in the comment regarding lines 128-144 (lines 516-528 of this document).

**Lines 244-246:** Not sold that you adequately demonstrate this. Allochthonous dispersal is a
constant issue with strong temporal and environmental effects on composition. I think
dispersal is very likely and should be acknowledged. I fail to understand the argument that
high richness and lack of local distance decay relationships nullify this concern. You may have
lack of distance relationships due to mass broadcasting of fungal spores regionally.

**This is the original text in question:** 'Also as mentioned at the outset, over a 17-week period,
we can ignore evolutionary divergence for fungi, and dispersal is unlikely to be of significant

*importance in our local, agricultural field due both to high richness (Fig. 6D) and the near*
*absence of any effect of distance on community dissimilarity (Fig. S14).'*

**Response – dispersal:** As noted in our response to Reviewer 1 on the same issue (lines 139-
158 in this document), to remove the influence of time on the detection of distance effect,
we re-analyzed the relationship between geographic distance and community dissimilarity
for each time point. Overall, similar to our results of pooling all time points as a whole (Fig.
S15), the null hypothesis of no relationship between geographic distance and community
dissimilarity cannot be rejected [$p < 0.002$ (0.05/17)] for any leaf, root or rhizosphere sample,
or for 15 of 17 soil samples (excepting weeks 12, 16). Thus, although there is an association
at two time points between community dissimilarity and distance for post-flowering soil, the
R is low (0.320-0.372) and this effect is seen at least five after our detection of stochasticity
in leaves and roots, whether before flowering or after the relief of drought stress.

**Response, mass broadcasting.** Of course, mass broadcasting of fungal spores does occur
regionally, but this broadcasting is not going to result in different fungi impacting leaves or
roots over the distance of our plots. Data that we now provide from air sampling shows no
distance decay among air samples taken in our plots (**Fig. S16**). Again, our finding of drift in
early leaves and roots is not affected by regional dispersal.

Fig. S15

**Fig. S16** Evidence of the absence of distance decay of air fungal community dissimilarity. (A)
air samplers were located in the sorghum field with a nested design. (B) Air fungal community
composition in each sample. (C) Principal coordinate (PCo) analysis of air fungal community
in terms of direction and location of samplers. (D) Flat relationship between geographic
distance and Bray-Curtis dissimilarity of air fungal communities.

**Line 250:** I would like to see an expansion of this line. Why MUST it be associated with size?
It seems that if it is associated with community size, that in of itself would be a slight
deterministic introduction. Why does the rate of randomness change with community size?
Since this is a big part of your entire thesis, this should be expanded on and not just accepted.
Justification is needed.

**This is the original text in question:** *‘However, because stochasticity can also contribute to*
*dispersal and divergence, attributing it to drift requires that the stochasticity also be shown*
*to be negatively correlated with community size under the observation that species in*
*communities with few individuals are prone to extinction by chance*¹⁷ *and assumptions that*
*the probability of extinction is expected to increase as the community size shrinks*^{44, 45, 46, 47.}

**Response.** We have revised the text to replace “requires” with “strengthened.”

**Revised in line 259-264:** *‘However, because stochasticity, over one season, can also contribute*
*to dispersal, arguments attributing it to drift are strengthened when stochasticity is negatively*
*correlated with community size. Whereas stochastic aspects of dispersal are expected to be*
*unaffected by community size, drift is clearly enhanced in small communities when individuals*
*are prone to extinction by chance*¹⁸ *and the probability of extinction is expected to increase*
*as the community size shrinks*^{46, 47, 48, 49}

**Lines 259-263:** Did you do any more, simple analyses such as estimated richness? Chao? To
point (i), this should not be used as there are SO MANY issues based on individualized

potential for biases from extraction, PCR efficiency, primer affinity, etc. Point (ii) is probably
fine but copy number concerns could occlude this. Single copy genes should optimally be
used.

**This is the original text in question:** *'...estimating community size using three different*
*methods, all of which were strongly correlated ($R > 0.8$, $P < 0.001$; Fig. S15): (i) the percentage*
*of fungal reads found in PCR amplifications of ITS2 from fungal and host DNA, (ii) the fungal*
*abundance as assessed by real time PCR amplification of rDNA small subunit (SSU) and (iii) the*
*percentage of fungal reads found in the transcriptomes of sorghum leaves and roots (Fig. S16-*
*S18).'*

**Response:** We feel that simple methods to assess richness would not help because we are
measuring abundance of fungi, not richness. We agree that there are many issues regarding
the measurement of abundance, which is why we employed three different methods to
measure abundance (rDNA ITS2 counts, rDNA 18S qPCR and mRNA counts). We use rDNA
because we need to account for all fungi, which would be technically impossible with single
copy regions where primer design would be a nightmare. We do sample single copy genes in
the transcriptome comparisons. We believe that our measurement of abundance is as
thorough as is now technically feasible.

**Lines 263-271:** This contradicts the statement made in lines 255-256. Which is it? That
statistical tools exist or they do not.

**This is the original text in question:** *'...and (2) the lack of statistical tools that can retrieve the*
*stochastic component of compositional variance.'*

**Response:** Agreed.

**Revision in lines 266-267:** *'and (2) the lack, until recently, of statistical tools that can retrieve*
*the stochastic component of compositional variance.'*

**Lines 274-276:** Beta NTI requires a tree based on aligned data. Given that you used ITS2 data,
how did you make this tree. I am suspicious of these data given that it is very unwise to align
ITS data. If you did align these, I would not trust these results as far as I can throw it (and this
is a large part of your explanations of drift - JUSTIFY). Please clarify. I like the inclusion of RCI,
but would not beta-RC be a better metric here?

**This is the original text in question:** *'Early in the season, when fungi were shown to be rare on*
*newly emerged leaves (weeks 1-7) and roots (weeks 1-3), we found that stochasticity shaped*
*the fungal community as shown when $|\beta NTI| < 2$ and $|RCI| < 0.95$ (Fig. 7A-D).'*

**Response:** We again thank the reviewer for catching this problem which we address beginning
on lines 194-242 in this document.

**Lines 282-292:** Paragraph lacks focus.

**This is the original text in question:** *'Intuitively, the presences of many small communities,*
*each containing just a few species that show little overlap with the others (resulting in a high*
*combined richness), implicates stochastic community assembly. Formally, having observed*
*stochasticity in early leaves and roots, we ascribe it to drift because the percentage of sample*
*pairs showing $|\beta NTI| < 2$ and $|RCI| < 0.95$, is strongly, negatively correlated with fungal*
*community size as detected by amplicon-, qPCR- and transcriptome- based methods (Fig. 7E-*
*G). Stochasticity can also be ascribed to ecological drift if it is positively correlated with*
*richness when community sizes are small, because the probability of extinction is expected to*
*increase when the total species pool is large relative to the local community size 51. Again,*
*formally, in line with this expectation and in support of drift, the strength of stochasticity was*
*found to be positively correlated with mean fungal richness for six replicate samples (Fig. 7H).'*

**Revision** in lines 295-306: *'Intuitively, stochastic community assembly would be suspected*
*when one is confronted by many small communities, each containing just a few species that*

*show little overlap with the others (resulting in a high combined richness). Quantitatively,*
*having observed stochasticity in early leaves and roots, we ascribe it to drift because the*
*percentage of sample pairs showing $|\beta NTI| < 2$ and $|RCI| < 0.95$ is strongly negatively*
*correlated with fungal community size as detected by amplicon-, qPCR- and transcriptome-*
*based methods (Fig. 5E-G). Stochasticity can also be ascribed to ecological drift if it is positively*
*correlated with richness when community sizes are small, because the probability of extinction*
*is expected to increase when the total species pool is large relative to the local community size*
*⁵⁴. Again, quantitatively, in line with this expectation and in support of drift, the strength of*
*stochasticity was found to be positively correlated with mean fungal richness for the six*
*replicates that constitute each of the samples (week, compartment, treatment) (Fig. 5H).'*

**Line 294-295:** Priority effects can be stochastic as well, not just deterministic.

**This is the original text in question:** *'...events that shape communities, such as, colonization*
*by stochastic dispersal and priority effects due to deterministic selection...'*

**Response:** Agreed.

**Revision in lines 308-309:** *"...events that shape communities, such as, colonization by*
*stochastic dispersal and priority effects..."*

**Lines 293-310:** Difficult to follow.

**This is the original text in question:** *'Complicating detection of drift during a field experiment*
*are the earlier, historical events that shape communities, such as, colonization by stochastic*
*dispersal and priority effects due to deterministic selection, both of which make it unlikely that*
*a replay of evolutionary time would produce new communities identical to those in place.*
*Therefore, we asked if the stochastic component of compositional variances observed here in*
*early leaf and root fungal communities could be attributed to dispersal limitation or historical*
*contingency. Concerning dispersal limitation, we detected no effect of geographic distance on*

*fungal community dissimilarity given the flat slope of the change in dissimilarity over distance*
*($R < 0.035$ in leaves and roots; Fig. S14). Concerning stochastic colonization in early fungal*
*communities, we found no support for this process because the first leaf and root fungal*
*communities were dominated in replicate samples by a single species, either OTU42*
*(Actinomucor) for leaves or OTU17 (Acrophialophora) for roots (Fig. 2B), two OTUs that were*
*rare in soil sampled prior to planting. Concerning priority effects involving selection in early*
*fungal communities, none were indicated because OTU42 and OTU17, although abundant in*
*all replicates in the 1st week, TP01, were largely replaced by other fungi by the 2nd week,*
*TP02 (Fig. 2B). Thus, we infer fungal communities are not ruled by historical contingency and*
*dispersal limitation in our local, homogenous, agricultural field. Of course, we cannot*
*comment on the strength of these processes in larger, heterogenous ecosystems.'*

**Response: We have revised the text to make it more direct.**

**Revision in lines 307-323:** *'Complicating detection of drift during a field experiment are the*
*earlier, historical events that shape communities, such as, colonization by stochastic dispersal*
*and priority effects, both of which make it unlikely that a replay of evolutionary time would*
*produce new communities identical to those in place ¹. Therefore, we asked if the stochastic*
*component of compositional variances observed here in early leaf and root fungal*
*communities could be attributed to dispersal limitation or historical contingency. We find no*
*evidence for dispersal limitation because we detected no effect of geographic distance on*
*fungal community dissimilarity as shown by the flat slope of the change in dissimilarity over*
*distance ($R < 0.263$, $P > 0.11$ in leaves and roots of each week; Fig. S15). We found no support*
*for stochastic colonization in early fungal communities because the first leaf and root fungal*
*communities were dominated in replicate samples by a single species, either OTU42*
*(Actinomucor) for leaves or OTU17 (Acrophialophora) for roots (Fig. 2B; Fig. S3), and both of*

*these OTUs were rare in soil sampled prior to planting (Fig. S5A). Neither did we detect any*
*priority effects involving selection in early fungal communities because OTU42 and OTU17,*
*although abundant in all replicates in the 1st week (TP01), were largely replaced by other fungi*
*by the 2nd week (TP02) (Fig. 2B; Fig. S3). Thus, we infer fungal communities are not ruled by*
*historical contingency or dispersal limitation in our local, homogenous, agricultural field.'*

**Lines 311-322:** These explanations for initial colonizers are perhaps too much of a stretch.

**This is the original text in question:** *'The ability of early colonizing fungi to produce large*
*amounts of hydrolytic enzymes might help explain their selection as first colonizers of roots*
*and leaves. The early colonizers, OTU42 (Actinomucor) and OTU17 (Acrophialophora), are*
*known to produce large amounts of enzymes that hydrolyze the protein, lipid, starch, and*
*xylan components of sorghum seeds 52; i.e., protease, lipase and amylase from Actinomucor*
*and xylanase from Acrophialophora 53, 54. The ability to produce abundant hydrolytic*
*enzymes suggests that these fungi are adapted to quickly mobilize nutrients released from*
*seeds at germination, but not to persist once the nutrients are exhausted. In the case of one*
*of these initially abundant leaf fungi, OTU42 (Actinomucor, Mucoromycota), its large*
*phylogenetic distance from the Ascomycota and Basidiomycota fungi that dominate other*
*time points (Fig. S3) explains why leaf samples of the first week are substantially different*
*from later samples in the phylogenetic-based ordination (Fig. S21).'*

**Response:** We would prefer to keep this section for the following reason. When we obtained
reviews of our manuscript from colleagues in our department prior to submitting it to *Nature*
*Communications*, the most thorough reviewer (from a microbiologist who is in our national
academy) highlighted these lines as his favorite because they dealt with function.

**Lines 353-354:** I guess I am still a bit confused why releveling drought would be expected to
favor stochasticity. With resource limitations (water availability) being alleviated, classic

ecological theory would predict that this should favor specialists over generalists, this implies,
to me at least, determinism, not stochasticity. Please clarify.

**This is the original text in question:** *'Contrary to expectations, the hypothesis (H2) that release*
*from pre-flowering drought would favor stochasticity was not supported because*
*stochasticity...'*

**Response.** We refer, in lines 347 to 350 of the previous submission (lines 360-363 of the new
MS), to a number of studies where release from stress has been shown to lead to stochastic
community assembly. *"Relief from stress has been shown to lead to a rise in compositional*
*variance among replicate microbial communities, whether the stress was due to drought*^{60, 61,}
*62, salinity*⁵⁹, *pH*⁶³, *nutrient limitation*^{64, 65}, *removal of perennial plant functional groups*⁶⁶,
*or predation*⁶⁷" Our hypothesis is based on observations in these publications that release
from stress leads to stochasticity. The explanation for these observations is that stress
provides a strong, deterministic force and in the absence of this deterministic force,
stochasticity can take a larger role in community assembly. However, in our system, we find
that release from drought simply strengthens the deterministic force provided by the plant.
Conversely, during drought, when the plant cannot provide a strong deterministic force, we
see evidence of stochastic community assembly.

**Lines 374-379:** Inclusion of RF methods is nice, but this does not really explain why this was
done or what expected results would tell us about the communities. Give that they did not
add to the story, I wonder if it would be better to omit this analysis as I fear it makes the story
more convoluted.

**This is the original text in question:** *'To compare the effects of pre- and post- flowering*
*drought on fungal community composition over time, we turned to Random Forest (RF)*
*models. Given that the key difference was most likely to be temporal, we needed to identify*

*the most important age-discriminant fungal OTUs (Fig. S24) to use them to develop a more*
*accurate sparse RF model, i.e., the trained model. To identify the most important age*
*discriminant fungal OTUs and use them to develop a trained model...'*

*Response: We prefer to keep the section on the temporal effect of drought of fungal*
*community assembly because the story differs from that published for bacteria (based on the*
*same samples) in that both types of drought affect fungi similarly, but pre-flowering drought*
*affects bacteria more strongly than post-flowering drought.*

**Figure 1:** This is a good figure, but a bit difficult to follow when also reading the MS. I suggest
expanding the MS to describe this setup better (but figure is good).

*Response: We agree and added more description in supplementary.*

*Revision in lines 26-38 of the supplementary "Briefly, Roots were removed from the ten plants,*
*mixed together, transferred to 50 ml tubes with detergent-phosphate buffer (6.33*
*NaH₂PO₄•H₂O and 8.5 g Na₂HPO₄•anhydrous in 1 L water, autoclaved; cooled, 200µl Silwet-*
*77 added; pre-cooled in ice-water mixture), and vortexed at full speed for 2 min. The roots*
*were removed from the tube, the liquid-filled tube was saved, and the roots were transferred*
*to a 200-ml plastic cup with phosphate buffer without detergent (6.33 NaH₂PO₄•H₂O and 8.5*
*g Na₂HPO₄•anhydrous in 1 L water, autoclaved; pre-cooled in ice-water mixture), vortexed at*
*full speed for 1 min twice, dried by clean paper towels, put into aluminum packet and frozen*
*in liquid nitrogen. The saved, liquid-filled tube containing the rhizosphere was centrifuged at*
*full speed for 3 min, the buffer discarded and the rhizosphere pellet frozen in liquid nitrogen.*
*Simultaneously, soil at 6" depth was collected adjacent to the ten sampled plants using 6" soil*
*collection tubes. Ten samples were mixed, transferred to a 50-ml centrifuge tube, and frozen*
*in liquid nitrogen. Our pooling ten samples to make a composite sample is conservative in that*
*it can only underestimate the contribution of stochastic process. Given that one of our major*

*results is detection of stochasticity, having found it with a conservative approach only*
 *strengthens our observation of stochasticity.”*

**Figure 2:** This is VERY busy. Are the OTU values to the right of A for B? I suggest remaking
 these so only the ten most abundant OTUs are represented and ‘Other’. This will clean it up
 quite a bit. Figure SA is nice, but the points are too small to see the compartment icons.

*Response. Figure 2b. Agreed. We remade these figures to show the only ten most abundant*
 *OTUs and clarify that the OTU values to the right are for Fig. 2B.*

 *Response Figure 2a, Agreed. To improve the figure to better show the different among the*
 *four compartments*

**Figure 3:** This is a nice figure, but I fear it doesn't really tell us that much about the
 communities and is sort of redundant of other similar figures. I would recommend eliminating
 this figure (or moving to Supp.)

**Response:** Agreed. We moved this Figure to the supplementary as Fig. S6.

**Figure 4:** Beautiful!

**Thanks!**

**Figure 5:** Given that there was an overall small effect of cultivar (and largely dismissed in MS),
 it seems odd to dedicate a figure to this. BUT it is a gorgeous figure, it just doesn't belong in
 the main MS.

**Response:** Agreed. We moved this Figure to the supplementary as Fig. S14.

**Figure 7:** You have to do a lot to convince me that Beta NTI is appropriate here due to ITS2
 data.

**Response:**

Agreed. We now calculate Beta NTI based on the 18S+28S tree developed by
taxonomy_to_tree method, as detailed in lines 194-242 of this document.

Supplemental Methods:

**Lines 31-36:** What primers were used?

**Response:** We now identify the primers as 5.8SFun and ITS4Fun in line 46 of supplementary.

**Reviewer #3 (Remarks to the Author):**

In the manuscript by Gao and colleagues, the researchers examine the temporal patterns of
fungal colonization and establishment on sorghum plants. They use a factorial field
experiment to test the effects of sorghum genotype (2 levels) and drought treatment (3
levels) on four components of the plant fungi microbiome, with three replicates of each
treatment grown in a plowed field. The main focus of this manuscript arises from their
interpretation of the results – they posit that ecological drift is important in driving
communities when the fungal communities are small, which occurs at the outset of the
experiment.

I do not have a background in fungal microbiomes, and will leave the novelty of that part of
the paper to other reviewers. My expertise lies in the community ecology inferences that the
authors are making and, based on the evidence presented, I am unconvinced that they are
real. I outline my concerns below, but want to emphasize that, even if all other issues could
be addressed, the low sample size for estimating variances (3 observations) would still raise
serious concerns for me.

**Response, Low Sample Size.**

**We failed to make it clear in our manuscript that, because we show that the effect of host
genotype is negligible, we are able to use six replicates in all of our analyses. To calculate a
dissimilarity matrix, six observations provide for 15 within-group sample pairs, which is a**

reasonable number to estimate compositional variance in our high-resolution agricultural
system. We now make this clear in the introduction and the legend of Figure 1.

**Added** words in lines 74-75: *'The discovery that the effect of host genotype is negligible*
*allowed us to use six replicates in most of our analyses'*

**The exciting conclusion** of this paper is that ecological drift plays an inordinately large role at
the outset fungal community assembly. There are several reasons why this conclusion is
unsupported, and they fall into a few main problem areas: misinterpretation of the ecological
literature on drift and dispersal limitation, statistical problems / inappropriate tests, and
experimental flaws.

1. Experimental flaws. There are two main problems with the experiment. First, the
environment is not carefully controlled – it took place in a plowed field, and plowed fields are
heterogeneous. If we imagine that the fungi undergo successional dynamics whereby the
environment determines which are initially most abundant but over time the fungi-plant
interaction selects for specific fungi, we would recreate the patterns the authors observed
(without evoking drift).

**Response:** We agree with the reviewer that our data show that, later in the season, the plant
selects for specific fungi. Early in the season, however, we disagree that the environment
determines which fungi are initially most abundant. If the reviewer were right, the
environment would have to be heterogenous early in the season. We provide a new PCA
analysis of fungal communities in the four compartments for the first two sampling times, TP0
and TP1 (Fig. S24b, below). From this analysis, it is clear that the replicate communities of the
soil and rhizosphere for these first two time points (TP00 and TP01) are more similar than are
the communities of leaves or roots for just the second time point. Thus, our data show that
the environment is not heterogenous early in the season, therefore, early fungal communities

of the plant roots and leaves cannot be determined by a heterogenous environment. This
result is consistent with our claim that drift is responsible for the assembly of the early fungal
communities of sorghum leaves and roots but that, later in the season, the plant selects the
fungal community.

**Fig. S24B PCo plots of fungal communities in unplanted soil and first week samples.**
Replicate communities of the soil and rhizosphere for these first two time points (TP00 and
TP01) are more similar than are the communities of leaves or roots for just the second time
point.

**Second, the authors** did not control for the initial abundances of fungi, which could
exacerbate environmental effects or work independently of them. They seem to interpret the
widespread, initial abundance of a few OTUs as evidence that dispersal limitation isn't
present. This interpretation is incorrect – most plowed fields will be colonized instantly by
thousands of dandelions, but that does not mean that no plant species are dispersal limited.
An experiment designed to isolate the effects of drift would control both these aspects by
using an homogeneous (sterile) soil with known or well-mixed inocula.

Response – Measurement of abundance and composition. One strength of our research is
that it is conducted in an agricultural field and not a greenhouse (where soil can be sterilized
and well-mixed inocula provided) because the results obtained in greenhouse experiments
rarely translate to the field. What makes our research possible are the recent advances in
characterization of fungal microbiomes. These methods allowed us to measure the
abundance of fungi and the composition of fungal communities at the start of our experiment
and throughout the growing season. We do not assume the absence of dispersal limitation
based on the initial abundance of a few OTUs, rather we are unable to find evidence for
dispersal limitation when we compare community dissimilarity with the distance between
communities. To measure fungal abundance, we use three quantitative methods, all three of
which proved to be highly correlated (see lines 231 to 235 in this document).

Response – dispersal. We do not state that “dispersal limitation isn’t present.” Instead, we
conclude that ‘dispersal limitation cannot account for the observed higher stochasticity in
early leaves and roots’ for three reasons: (i) we find no significant relationship between
geographic distance and Bray-Curtis dissimilarity of fungal communities in either roots or
leaves in any of the 17 weeks. (ii) After accounting for false discovery [$P = 0.003 (0.05/17)$], a
significant relationship between geographic distance and Bray-Curtis dissimilarity is observed
in soil only at week 12 and week 16, well past the time when stochasticity was observed in
leaves and roots. (iii) Throughout the growing season, we find no relationship between
geographic distance and airborne fungal communities. We are adding to the supplementary
material a figure showing dissimilarity by distance for all compartments over all weeks (as
described above in response to Reviewers 1 and 2, Fig. S15), and a second, new figure
presenting data from air sampling over the growing season (Fig. S16)(see lines 668 to 699 in
this document).

Bray-Curtis dissimilarity

Geographic distance (m)

Fig. S15

**Fig. S16** Evidence of the absence of distance decay of air fungal community dissimilarity. (A)
air samplers were located in the sorghum field with a nested design. (B) Air fungal community
composition in each sample. (C) Principal coordinate (PCo) analysis of air fungal community
in terms of direction and location of samplers. (D) Flat relationship between geographic
distance and Bray-Curtis dissimilarity of air fungal communities.

**2. Misinterpretation of literature.** i. The authors test for distance-dependent dissimilarity in
fungal communities as a way to assess whether dispersal limitation is important. Classic work
by Hurtt and Pacala (J. Theor Biol 1995) show that dispersal limitation (or fecundity limitation)
can cause strong colonization effects in otherwise completely deterministic communities.
Distance decay is not the only way that stochastic colonization effects can manifest –
differences in abundance among species can be sufficient to cause these to happen. The data
suggest that not all species are present at all locations (far from it, given the turnover stats),
meaning that colonization differences are likely important.

**Response – Dispersal limitation.** The reviewer argues that dispersal limitation can cause
stochastic colonization in communities shaped primarily by determinism. As noted above, if
dispersal limitation were occurring, the methods we use to assess fungal mycobiomes would
observe distance-dependent dissimilarity among fungal communities. However, we observe
none. Therefore, we have no evidence that dispersal limitation is important in our system.

**Response – Stochastic colonization.** If, as suggested by the reviewer, stochastic colonization
was important, some of our observations would not be possible.

(1) The first leaf and root fungal communities were dominated in replicate samples by a
single species, either OTU42 (Actinomucor) for leaves or OTU17 (Acrophialophora) for
roots (Fig. 2B, Fig. S3), two OTUs that were rare in soil sampled prior to planting. If

colonization were stochastic, the different samples would not be dominated by the
same species.

(2) We observed a strong negative correlation between stochasticity and community size.
If stochastic colonization were important, its effect would not be related to
community size. The negative correlation between stochasticity and community size,
however, is a characteristic of ecological drift (stochastic extinction).

We do not deny any contribution of stochastic colonization, in fact, the role of ecological drift
(stochastic extinction) can be amplified by stochastic colonization. What our data show is that
ecological drift is occurring.

**ii. The authors seem to** be under the impression that a Raup-Crick index informs us about
neutral stochasticity (I apologize if I am misinterpreting this but, if so, I think that the
interpretation needs to be clearer in the paper). The Raup-Crick is a null model for random
redistribution of individuals, which is very different from expectations for drift. The other
metric (BNTI) has no relationship whatsoever.

**Response – Raup-Crick, BNTI.** The reviewer is mistaken about the use of Raup-Crick and BNTI
to detect stochasticity. S/he is misinterpreting the common use of Raup-Crick and bNTI to
detect stochasticity, uses that are widely accepted in recent ecological publications and
reviews of the field (Chase et al 2011 Ecosphere, Stegen et al 2012, 2013 ISMEJ, Dini-Andreote
et al 2015 PNAS; Zhou & Ning 2017 Microbiol. Mol. Biol. Rev).

Raup-Crick and bNTI compare the matrix of dissimilarities calculated for observed
communities to those calculated from communities randomly assembled by sampling, with
replacement, from the pool of all observed taxa. The position of the matrix of dissimilarity
calculated from observed data with respect to the distribution of those calculated from
multiple rounds of resampling determines the probability that the null hypothesis of

stochasticity can be falsified. The two tools that we employed are the beta Nearest Taxon
Index (β NTI) and the Raup-Crick Index (RCI), which indicate stochasticity when the $|\beta$ NTI| < 2
(reflecting stochastic turnover in phylogenetic composition) and when the $|RCI| < 0.95$
(reflecting stochastic turnover in species composition).

**3. Statistical problems.** Apart from the issue with the Raup-Crick and BNTI methods and their
interpretations, I was unsure of how the authors examine beta diversification. Their graphs
seem to have more observations per time point than there were replicates. I imagine that this
is an error in the presentation, but should be fixed.

**Response:** Again, we apologize for not making it clear that, owing to the lack of host genotype
effect, we are able to combined the two genotypes and use six replicates in calculating beta
diversity. Therefore, there is no error in the presentation. We now make it clear that we have
six replicates.

**The fitting of the neutral** model was also challenging – it appears that the authors compared
AIC values between two datasets (their initial data split into two) rather than for nested
models. This either needs to be fixed (if the authors did compare two datasets) or the authors
should explain their methods more clearly. Beyond these issues, the authors should consider
the limitations outlined above (concern #1) and think about if the temporal data could be
used in a way that actually detects drift by decoupling random from directional changes in
identities through time.

**This is the original Figure of last submission**

Response: The Sloan Neutral Community Model (SNCM) is carried out for six samples of each
 time point of each treatment and each compartment. There are no temporal directional
 changes in identities through time within each dataset. Due to the large number of datasets
 involved, we only plotted the AIC value (and fitting curve) against time (Fig. SA).

To examine the difference between early leaves and later leaves, we split the leaf data into
 two datasets and carried out the analysis for each dataset (Fig. SB). But this dataset is involved
 temporal directional changes, thus we remove it (Fig B) from the new MS. The following is
 the new Fig. S23

On a final note, I think that it would be completely reasonable for the authors to discuss drift

as one mechanism driving the patterns observed, but the current manuscript does not

convince me that this is the only (or even the most likely) possibility.

Response: We are heartened to read that the reviewer considers drift to be one mechanism
for explaining the early communities in leaves and roots. Drift's influence is short lived
because, as time goes on and the plant grows, we find that deterministic forces displace drift.
We hope that our explanations and revisions will let this reviewer accept the view provided
by our analyses of the fungal mycobiome.

Reviewers' comments:

Reviewer #1 (Remarks to the Author):

I am satisfied by the authors' responses to my comments on the original version of the manuscript.

I do quibble with their response on lines 39-49 of their response letter. Their criticisms of the studies that I cited should lead to those studies underestimating dissimilarities even more than what was observed, further indicating how the low dissimilarities in the current study might limit their ability to make general conclusions. It is also implied that the sequencing methods used in the study are 'meaningful', whatever that means, but even those have issues with variable resolution that can lead one to challenge their ecological relevance. But this is just me ranting and hoping that they are less likely to be so dismissive in the future.

Reviewer #3 (Remarks to the Author):

In the manuscript by Gao and colleagues, the researchers examine the temporal patterns of fungal colonization and establishment on sorghum plants. They use a factorial field experiment to test the effects of sorghum genotype (2) and drought treatment (3) on four components of the plant fungi microbiome, with three replicates of each treatment grown in a plowed field. The main focus of this manuscript arises from their interpretation of the results – they posit that ecological drift is important in driving communities when the fungal communities are small, which occurs at the outset of the experiment.

This is my second review of this paper and, unfortunately, I am no more convinced of its conclusions with respect to ecological drift than I was with the first version. The authors have attempted to address several of my concerns, but have failed to address the central issues of the appropriateness of the statistics used and their relevance for detecting ecological drift.

My main concerns are as follows.

1. The environment was not carefully controlled – it took place in a plowed field, and plowed fields are very heterogeneous. This is not a strength, as the authors state, but rather a fundamental flaw. To attribute variation in community composition to drift requires removing deterministically-driven variation. This includes environmental variation and, for this study, removal of genetic differences.

2. Misinterpretation of literature. The authors test for distance-dependent dissimilarity in fungal communities, and assume that this is the way to assess dispersal limitation. This assumption is correct for neutral models, but is incorrect for virtually all spatial models of biodiversity.

3. The authors misinterpret the Raup-Crick index. The Raup-Crick is a null model for random redistribution of individuals, which is very different from expectations for drift. The other metric (BNTI) has no relationship whatsoever. The authors contend that these are valid metrics of drift, but this comes from a misunderstanding of drift – drift is the random change through time in species relative abundances that occur from demographic stochasticity. It is not a random redistribution of species. As I stated in my previous review, there are ways to calculate expectations for neutral drift through time – the work by Egbert Leigh and colleagues is particularly relevant here (e.g., Gilbert et al. 2006, *American Naturalist*; Leigh et al. 1993, *Evolutionary Ecology*).

4. In addressing my earlier concerns, the authors state that they have 15 within-group sample pairs, and argue that this is a high level of replication. The number of replicates is 3 (potentially six, although there would need to be a more convincing demonstration that genotype really is unimportant). Using pairwise distances (and a Mantel approach) has been shown to be statistically flawed and cannot increase the number of independent observations. The authors should look to multivariate methods that calculate distance from the centroid (in R, the *betadisper* function in the

vegan package does this. Similarly, in Gilbert and Levine (2017, Proceedings B) the authors outline how to calculate these distances using an FST metric that may be appropriate here.

5. The fitting of the neutral model was done incorrectly. The authors compared AIC values among datasets rather than for nested models. This is simply wrong – as the number of species increases in a dataset, the AIC will go down even if the model fits each observation equally well.

As I previously stated, I think that it would be completely reasonable for the authors to discuss drift as one mechanism driving the patterns observed, but this is just one possibility for their results.

We deeply appreciate both this opportunity to appeal the decision about our manuscript and the effort that reviewers and editors have expended, all of which have improved our manuscript. In particular, we appreciate the comments of Reviewer #3 because these comments likely represent those of many researchers in the field and allowed us to treat the criticisms as hypotheses that we could address with our data set. As we reiterate in the conclusion of this document, our fondest hope is that our manuscript stimulates discussion about the forces that act to assemble fungal communities. In this regard, receipt of the comments of Reviewer #3 was particularly welcome before our work could appear in print.

Reviewer #1 (Remarks to the Author): (Our responses are in red)

I am satisfied by the authors' responses to my comments on the original version of the manuscript.

Response: We appreciate the reviewer's comments about the effort of our revision.

I do quibble with their response on lines 39-49 of their response letter. Their criticisms of the studies that I cited should lead to those studies underestimating dissimilarities even more than what was observed, further indicating how the low dissimilarities in the current study might limit their ability to make general conclusions. It is also implied that the sequencing methods used in the study are 'meaningful', whatever that means, but even those have issues with variable resolution that can lead one to challenge their ecological relevance. But this is just me ranting and hoping that they are less likely to be so dismissive in the future.

Response: We did not intend to deny the merit of these previous studies. We cited these papers in our manuscript and continue to do so. We have added, in this second revision, a stronger call for more and larger efforts (68, 69, 70).

Revised texts: *"These comparisons suggest the existence of general universal ecological dynamics from human-associated ecosystems to agricultural ecosystems, and from bacteria to fungi and fungal guilds. Still, given the small number and scope of the studies, universality needs to be evaluated by more studies in more complex ecosystems and at larger scales^{68, 69, 70}."*

Reviewer #3 (Remarks to the Author):

In the manuscript by Gao and colleagues, the researchers examine the temporal patterns of fungal colonization and establishment on sorghum plants. They use a factorial field experiment to test the effects of sorghum genotype (2) and drought treatment (3) on four components of the plant fungi microbiome, with three replicates of each treatment grown in a plowed field. The main focus of this manuscript arises from their interpretation of the results – they posit that ecological drift is important in driving communities when the fungal communities are small, which occurs at the outset of the experiment.

This is my second review of this paper and, unfortunately, I am no more convinced of its conclusions with respect to ecological drift than I was with the first version. The authors have attempted to address several of my concerns, but have failed to address the central issues of the appropriateness of the statistics used and their relevance for detecting ecological drift. My main concerns are as follows.

1. The environment was not carefully controlled – it took place in a plowed field, and plowed fields are very heterogeneous. This is not a strength, as the authors state, but rather a fundamental flaw. To attribute variation in community composition to drift requires removing

deterministically-driven variation. This includes environmental variation and, for this study, removal of genetic differences.

Reviewer #3 voiced three opinions in his/her first review and maintained them in his/her second review. We treated each opinion as a hypothesis and we provided data to test these hypotheses in both our original manuscript and our first revision. Here we explain how we used our data to address each of Reviewer #3's opinions.

Response: Opinion in Reviewer 3's statement that "plowed fields are very heterogeneous." Our data (Fig. S23B) show that fungal communities in the unplanted soil in the field are homogenous. Therefore, we feel that the reviewer is mistaken in his/her opinion that fields are heterogeneous with regard to fungal communities. Our research challenged this opinion and our data show that an agricultural field, prior to planting, is homogeneous with respect to fungal communities. The homogeneity of the field is likely the product of at least 57 years of intensive cultivation at our research site. In our first revision we added data on airborne fungi that settle on the field, again finding that analysis of two monthly samples from 13 samplers placed in the field in a nested square design showed no heterogeneity (Figure S16).

Response: Opinion in Reviewer 3's comment about "removal of [host] genetic differences." Our data (Fig. S14) show no difference in fungal OTUs on leaves of the two sorghum cultivars prior to flowering when stochasticity is dominant. Again, we feel that the reviewer is mistaken because he/she holds an opinion that different plant genotypes must harbor different fungal communities. Again, our research challenged this opinion and our data show that there are no differences in the fungal communities of the two cultivars early in the season when we detect ecological drift.

2. Misinterpretation of literature. The authors test for distance-dependent dissimilarity in fungal communities, and assume that this is the way to assess dispersal limitation. This assumption is correct for neutral models, but is incorrect for virtually all spatial models of biodiversity.

Response: The reviewer writes that dispersal limitation cannot be assessed using distance-dependent dissimilarity. Our data (Fig. S15) show no distance-dependent dissimilarity among fungal communities in the field. If there is no distance-dependent dissimilarity, then we felt that there could be no dispersal limitation. At the June 2019, American Society for Microbiology meeting we raised this point with other microbial ecologists and received this response: distance-dependent dissimilarity is a necessary condition to claim dispersal limitation and its absence is sufficient to reject dispersal limitation. We feel that this opinion is related to the reviewer's opinion that plowed fields cannot have homogenous fungal communities. Again, our research addresses this opinion and our data show that plowed fields, prior to planting, have a homogeneity of fungal communities (Fig. S23B).

3. The authors misinterpret the Raup-Crick index. The Raup-Crick is a null model for random redistribution of individuals, which is very different from expectations for drift. The other

metric (β NIT) has no relationship whatsoever. The authors contend that these are valid metrics of drift, but this comes from a misunderstanding of drift – drift is the random change through time in species relative abundances that occur from demographic stochasticity. It is not a random redistribution of species. As I stated in my previous review, there are ways to calculate expectations for neutral drift through time – the work by Egbert Leigh and colleagues is particularly relevant here (e.g., Gilbert et al. 2006, *American Naturalist*; Leigh et al. 1993, *Evolutionary Ecology*).

Response: The reviewer writes that neither Raup-Crick nor β NIT can be used to detect ecological drift. We use the Raup-Crick index and β NIT to detect stochasticity. Where we find stochasticity, we then take extra steps, described below, to address drift. As the reviewer notes, drift is the random change through time in the relative abundance of species that occur from demographic stochasticity. Therefore, having demonstrated stochasticity, we tested the hypothesis that random species extinction should be higher in smaller communities and, finding that stochasticity is higher in smaller communities, we then infer drift.

Our use of the Raup-Crick index and β NIT to detect stochasticity follows the recent, pioneering work of many researchers who use these two approaches to detect stochasticity (Dini-Andreote et al 2015 PNAS; Zhou & Ning 2017 *Microbiol. Mol. Biol. Rev.*; Stegen et al 2012, 2013 *ISMEJ*).

4. In addressing my earlier concerns, the authors state that they have 15 within-group sample pairs, and argue that this is a high level of replication. The number of replicates is 3 (potentially six, although there would need to be a more convincing demonstration that genotype really is unimportant). Using pairwise distances (and a Mantel approach) has been shown to be statistically flawed and cannot increase the number of independent observations. The authors should look to multivariate methods that calculate distance from the centroid (in R, the `betadisper` function in the `vegan` package does this. Similarly, in Gilbert and Levine (2017, *Proceedings B*) the authors outline how to calculate these distances using an F_{ST} metric that may be appropriate here.

Response: Reviewer 3 argues that genetic differences between our two host plants make it impossible to combine the three replicates of each host genotype to bring the total to six. We feel that we do have six replicates because analysis of fungal communities associated with the two sorghum cultivars shows that host genotype explains no variance prior to flowering (when we detect drift) and almost none (0.2%) over the entire growing season. Because host genotype accounts for no variance when we detect drift, we feel that it is correct to combine the replicates of the two cultivars.

Response: The reviewer argues that we should use multivariate and F_{ST} methods to calculate distance. Both the original and revised version of our manuscript contain our analyses using both the multivariate method to calculate distance from the centroid (beta dispersion; Figs. S8; S11; S12; S13) and the F_{ST} metric (Fig. S22) of Gilbert and Levine (2017, *Proceedings B*).

5. The fitting of the neutral model was done incorrectly. The authors compared AIC values

among datasets rather than for nested models. This is simply wrong – as the number of species increases in a dataset, the AIC will go down even if the model fits each observation equally well.

Response: We accept Reviewer 3's opinion. We used the neutral model to attempt to reject our finding of drift. Although the result of the neutral model failed to reject our finding of drift, thereby supporting our conclusions, removing it will not affect our conclusions and we are happy to take the reviewer's advice and remove it.

Text in lines 298-300 of the last submitted manuscript: '~~Given that stochasticity is one of the consequences of neutral process 16, our data best fit the Sloan Neutral Community Model (SNCM) in early leaves, again as expected (Fig. S23).~~'

As I previously stated, I think that it would be completely reasonable for the authors to discuss drift as one mechanism driving the patterns observed, but this is just one possibility for their results.

Response. We appreciate the reviewer's position. We feel that we already state that drift is just one of four mechanisms driving community assembly (introduction lines 62-69). We then investigate two situations where drift would be expected to occur, that is, early developing leaves and plants released from stress, and find that our approach can both detect drift in early leaves (H_1) and reject its influence following stress relief (H_2 ; noted both in our abstract (L34-L38) and manuscript (L95-L100; H_1 beginning L251; H_2 beginning L352). We also acknowledge the action of deterministic forces, by reporting that, apart from early and drought-stressed leaves, fungal community assembly involves selection.

We feel that what makes our research appealing to reviewers and readers is our novel finding of drift in the assembly of fungal communities in an agricultural field setting. We feel that our result is due to our use of new molecular and analytical tools to assess communities of fungi. These new tools are allowing researchers to challenge ideas that emerged from traditional approaches. As clichéd as it is, our fondest hope is for our manuscript to stimulate critical discussion about fungal community assembly.

Reviewers' comments:

Reviewer #1 (Remarks to the Author):

I was asked to review the authors responses to reviewer #3's comments. I believe that all of the responses are reasonable and should address the concerns that were raised. The authors use multiple, state-of-the-art lines of evidence to support their conclusions that drift can be an important process during the assembly of plant-associated fungal communities. I don't think that it is entirely accurate to say that the soils were 'homogeneous' since at some timepoints there were significant relationships between dissimilarity and distance (including at the start - Figure S15) but the slope of the relationship is so flat that it is probably true in effect. Thus I am still supportive of the manuscript being accepted for publication.

Reviewer #4 (Remarks to the Author):

I was asked to review this manuscript in place of a reviewer (reviewer 3) who could no longer do so, and to evaluate the extent to which that reviewer's comments had been addressed by the authors. In order to avoid bias, I read through the manuscript first and formulated my own opinions, as I would if I were reviewing a paper for the first time, and then read through Reviewer 3's comments afterwards. The short version is that my comments were very similar to those of Reviewer 3, which in my opinion went largely (but not entirely) unaddressed.

In this paper, the authors seek to understand the community assembly processes that result in fungal community variability in several different compartments of sorghum plants (soil, rhizosphere, roots, leaves). The authors focus on selection, dispersal, and drift, discounting speciation as unlikely because of the time scales involved (a decision I support). The authors have two main hypotheses, which are H1: drift is important when fungal communities are "small", and H2: drift is important after drought stress is relieved.

The authors infer that drift occurs via the following logic: A: drift is a stochastic process and generates random variability in community composition. B: drift makes sense if "stochasticity" in community structure is negatively correlated with community size, since less populous communities are more vulnerable to drift. C: low-population high-richness communities are expected to exhibit the most drift, because per-species risk is highest.

I have no problems with the logic above, except that I think it is poorly articulated in the paper and these arguments are spread throughout using inadequate and imprecise language. Subpoints:

- Stochasticity is an attribute of a process, not of data. Drift as a process is stochastic, but the result of drift is randomness in community composition (hence my quotes around stochasticity above). This point is largely semantic and is less important than the next:
- The authors' logic for drift is difficult to understand. Part is found around lines 304-310, and another part is found around lines 264-266.

However, the logic above is contingent on the authors actually observing the results of drift. However, I do not believe the authors' metric (proportion of samples where both β NTI and RCI do not deviate from null expectation) is appropriate for measuring drift, for several reasons:

- $|\beta$ NTI < 2 just means that the observed β NTI for a pair of samples is within two standard deviations of the null β NTI distribution. That distribution is created by permuting identities in the phylogenetic tree. A negative value indicates phylogenetic underdispersion, i.e. the two communities are more phylogenetically similar than expected by chance. A positive value indicates overdispersion. An empirical β NTI value that falls within the null distribution (2 SD is a bit arbitrary but OK) represents a pair of communities that are phylogenetically homogenous. Reviewer 3 had a

very similar comment (#3).

- $|RCI| > 0.95$ also means homogeneity, but of composition (OTUs) instead of phylogeny. For RCbray, the null distribution is created by resampling communities under the assumption of homogeneity, i.e. under the assumption that any community is composed of the same proportion of OTUs found in its surroundings (i.e. metacommunity), with some randomness thrown in as a random multinomial draw of OTUs with probabilities proportional to the total count of each OTU. Again, homogeneity, not drift.
- Findings consistent with a null hypothesis ($|\beta NTI| < 2$ or $|RCI| < 0.95$) are not evidence FOR the null hypothesis. This is stats 101.
- The null RCI distribution may be what is expected if drift is the ONLY community assembly process acting between a pair of samples, but homogenizing dispersal (how communities got where they are in the first place under high mixture) happened at some point, and is a community assembly process in and of itself. So drift doesn't explain why the two communities look the same (why RCI and βNTI look like the null), that's because of homogenizing dispersal. Drift is just all that's gone on AFTER that initial assembly, and the intensity of that drift is not indicated by RCI or βNTI . I suspect this is a misinterpretation the authors have of Stegen et al. 2013.
- The absence of spatial patterns in beta-diversity (bray) observed by the authors is just MORE evidence of homogeneity. However, this analysis is largely superficial since individual OTUs can exhibit spatial structure even when overall community composition shows little or no pattern. This is especially true since bray-curtis is a weighted metric. Let's say one or two OTUs are fairly rare, but each has a spatial pattern. e.g. an east-west abundance gradient, or patch dynamics. That spatial structuring would be swamped out by a beta-diversity mantel style analysis. Again, Reviewer 3 had a very similar comment (#2, although they explained very little). While distance-dependent similarity is a necessary to claim dispersal limitation OF A SPECIES, the inverse is not necessarily true as stated above, especially in regard to broad-scale community variability.
- Unexplained variability is not drift. The authors say they observed "high stochasticity" on line 304, and it's tempting to ascribe unexplained variability in community structure to stochastic processes. However hidden variables are impossible to rule out in an uncontained real-world experiment such as this sorghum field. This was mentioned by Reviewer 3 in their first comment (#1).

I was told not to make to many minor points, but:

- Taxonomic composition of fungal communities (129-134) seems superfluous.
- Calling plant compartment "space" (142) is confusing.
- Critical typo: position vs. proportion (283).
- What constitutes a "small" community size for microbes? The authors don't seem to have an a priori idea of how "small" a community would need to be in order to be vulnerable to drift. I think it's fine to discuss this relationship in a correlative framework (as in most of the paper), but around line 309 this is not the case.

Finally, my broad impression of the paper is that it has an interesting experiment, but the authors push their results too far. I do not think the authors need to be so focused on definitively showing "drift", especially because nobody (myself especially) will believe that they can observe drift in a natural system. However, like Reviewer 3 wrote in their review, "I think that it would be completely reasonable for the authors to discuss drift as one mechanism driving the patterns observed, but this is just one possibility for their results". Again, I would like to re-iterate that these opinions are my own, and were not informed in any way by those of Reviewer 3. The high overlap between my comments and theirs, however, indicates that their comments were not adequately addressed.

Response to Reviewers, September 27, 2019

We identify three main points raised by reviewers #1, #3 and #4 that must be addressed: 1.
Alternatives to drift as an explanation for stochasticity. 2. The use of RCI and β NTI to detect
stochasticity, and 3 The belief that soil fungal communities cannot be homogeneous. Short
summaries of our responses are just below and full responses follow that include Reviewers #1
and 4 but focus on the criticisms of Reviewer #4.

- 1. Alternatives to drift as an explanation for stochasticity. Here, we now agree with
Reviewers #3 and #4 that there are two possible explanations for stochasticity, drift
(which is wholly stochastic) and the stochastic component of dispersal. We now note
both explanations throughout our manuscript.
- 2. The use of RCI and β NTI to detect stochasticity. RCI and β NTI are routinely used to
detect stochasticity, as noted in our citations. Just this month **Nature Communications**
(Xun et al, 2019, Nat. Comm) published a study of bacterial communities that used β NTI
to detect stochasticity. We feel that we are on solid ground here, but we realize from
the reactions of two reviewers show that we need more justification for all readers,
which we now provide.
- 3. The feeling that soil fungal communities cannot be homogeneous. All of our soil
community data show that fungal communities in our field are homogeneous. Again, we
feel that we are on solid ground here. At the request of Reviewer #4 we added analysis
using a measure of community dissimilarity that accounts for abundance and still find
that our soil, fungal communities are homogenous. We also have data on airborne
fungal propagules (spores and hyphal fragments), which also show homogeneity. Again,
seeing that two reviewers have difficulty believing that soil, fungal communities can be
homogeneous, we realize that we must provide more background for our results, that
is, emphasizing the point that a 50-year-old, well-ploughed field with one crop plant is
not a truly wild system and could be expected to have homogenous fungal communities.

Responses to full reviewers' comments: comments from the reviewer are **black**, our responses
are **red** and our revised text is **blue**.

Reviewer #1 (Remarks to the Author):

I was asked to review the authors responses to reviewer #3's comments. I believe that all of the
responses are reasonable and should address the concerns that were raised. The authors use
multiple, state-of-the-art lines of evidence to support their conclusions that drift can be an
important process during the assembly of plant-associated fungal communities. I don't think
that it is entirely accurate to say that the soils were 'homogeneous' since at some timepoints
there were significant relationships between dissimilarity and distance (including at the start -
Figure S15) but the slope of the relationship is so flat that it is probably true in effect. Thus I am
still supportive of the manuscript being accepted for publication.

**Response:** We appreciate the reviewer's comments about the effort of our revision. Regarding
homogeneity, our investigation of community dissimilarity and geographic distance involves
multiple tests for each compartment, so we feel that P-value for significance should be at 0.01.
If the bar for significance is set at $P = 0.01$, of the 69 samplings for the four compartments of
soil, rhizosphere, root and leaf, only two show a significant relationship between community
dissimilarity and geographic distance, soil at week 12 and soil at week 16, well after our
detection of stochasticity in young leaves or roots. In all of the 69 samplings, we agree with the
reviewer that the relationships, even the two significant ones, are "so flat that it [homogeneity]
is probably true in effect."

Reviewer #4 (Remarks to the Author):

I was asked to review this manuscript in place of a reviewer (reviewer 3) who could no longer
do so, and to evaluate the extent to which that reviewer's comments had been addressed by
the authors. In order to avoid bias, I read through the manuscript first and formulated my own
opinions, as I would if I were reviewing a paper for the first time, and then read through
Reviewer 3's comments afterwards. The short version is that my comments were very similar to
those of Reviewer 3, which in my opinion went largely (but not entirely) unaddressed.

In this paper, the authors seek to understand the community assembly processes that result in
fungal community variability in several different compartments of sorghum plants (soil,
rhizosphere, roots, leaves). The authors focus on selection, dispersal, and drift, discounting
speciation as unlikely because of the time scales involved (a decision I support). The authors
have two main hypotheses, which are H1: drift is important when fungal communities are

“small”, and H2: drift is important after drought stress is relieved.

The authors infer that drift occurs via the following logic: A: drift is a stochastic process and
generates random variability in community composition. B: drift makes sense if “stochasticity”
in community structure is negatively correlated with community size, since less populous
communities are more vulnerable to drift. C: low-population high-richness communities are
expected to exhibit the most drift, because per-species risk is highest.

I have no problems with the logic above, except that I think it is poorly articulated in the paper
and these arguments are spread throughout using inadequate and imprecise language. Subpoints:

Reviewer bullet point A

• Stochasticity is an attribute of a process, not of data. Drift as a process is stochastic, but the
result of drift is randomness in community composition (hence my quotes around stochasticity
above). This point is largely semantic and is less important than the next:

**Response:** The difference between ‘stochasticity’ and ‘randomness’ is subtle, as two words are
interchangeable in most cases: [https://math.stackexchange.com/questions/114373/whats-the-](https://math.stackexchange.com/questions/114373/whats-the-difference-between-stochastic-and-random)
[difference-between-stochastic-and-random](https://math.stackexchange.com/questions/114373/whats-the-difference-between-stochastic-and-random)

For studies using RCI and β NTI to detect stochasticity/randomness (such as Dini-Andreote et al
2015 PNAS, Zhou & Ning MMBR 2017), the word ‘stochasticity’ is commonly used, and the word
randomness is rarely used. Therefore, for consistency in our manuscript and the field, we prefer
to use ‘stochasticity’ and have removed randomness throughout our manuscript.

Reviewer bullet point B

• The authors’ logic for drift is difficult to understand. Part is found around lines 304-310, and
another part is found around lines 264-266.

**Response:** Both lines 264-266 and lines 304-310 are subsets of our section dealing with
ecological drift part (lines 257-343). We now see that we can simply delete the lines 264-266
without affecting the meaning.

**Revised text:**

As noted in the introduction, four ecological forces shape fungal community composition,
selection, dispersal, evolutionary divergence, and drift, only one of which is wholly
deterministic, selection¹⁶. Also, as mentioned at the outset, one of the forces with a stochastic
component, divergence, can be ignored for fungi over a 17-week period⁴⁵, leaving two forces
with stochastic components, wholly stochastic drift and the stochastic part of dispersal. Thus,
should we detect stochasticity, it could be attributed to drift or stochastic dispersal in the
period of initial colonization. Whereas stochastic aspects of dispersal, in this case the initial

colonization, are expected to be unaffected by community size, drift is clearly enhanced in small
communities when individuals are prone to extinction by chance¹⁸ and the probability of
extinction is expected to increase as the community size shrinks^{47, 48, 49, 50}.

Reviewer bullet point C

However, the logic above is contingent on the authors actually observing the results of drift.

However, I do not believe the authors' metric (proportion of samples where both β NTI and RCI
do not deviate from null expectation) is appropriate for measuring drift, for several reasons:

**Response:** As we now say in the revised text just above, we use β NTI and RCI to detect
stochasticity, not drift. Where we find stochasticity, we then take extra steps, described below
in our responses to Reviewer bullet points D and E, to address drift.

Revised text:

Quantitatively, having observed stochasticity in early leaves and roots, we ascribe a significant
fraction of it to drift because the percentage of sample pairs showing $|\beta$ NTI| < 2 and |RCI| <
0.95 is strongly negatively correlated with fungal community size as detected by amplicon-,
qPCR- and transcriptome- based methods (Fig. 5E-G). Stochasticity can also be ascribed to
ecological drift if it is positively correlated with richness when community sizes are at their
smallest, because the probability of extinction is expected to increase when the total species
pool is large relative to the local community size⁵⁵. Additionally, a consequence of ecological
drift is high beta diversity, which can lead to high, joint richness (Fig. 5H). Again, we find that
the strength of stochasticity is positively correlated with mean fungal richness for the six
replicates that constitute each of the 171 samples (week, compartment, treatment) (Fig 5H).

Having ascribed a significant fraction of stochasticity to drift, can we rule out the action
of stochastic dispersal in the colonization of emerging plants? No clear evidence in support of
stochastic colonization was found because the first leaf and root fungal communities were
dominated in replicate samples by a single species, either OTU42 (*Actinomyces*) for leaves or
OTU17 (*Acrophialophora*) for roots (Fig. 2B; Fig. S3), and both of these OTUs were rare in soil
sampled prior to planting (Fig. S5A). However, despite the dominance of OTU42 and OTU17 in
the first week, we cannot rule out the stochastic colonization of other OTUs, nor can we rule
out any stochastic colonization that might occurred during the 2 weeks between planting and
our first sampling. Concerning events that occur prior to sampling, we did not detect any
priority effects involving selection in early fungal communities because OTU42 and OTU17,
although abundant in all replicates in the 1st week (TP01), were largely replaced by other fungi
by the 2nd week (TP02) (Fig. 2B; Fig. S3).

Reviewer bullet point D

• $|\beta\text{NTI}| < 2$ just means that the observed βNTI for a pair of samples is within two standard
deviations of the null βNTI distribution. That distribution is created by permuting identities in
the phylogenetic tree. A negative value indicates phylogenetic underdispersion, i.e. the two
communities are more phylogenetically similar than expected by chance. A positive value
indicates overdispersion. An empirical βNTI value that falls within the null distribution (2 SD is a
bit arbitrary but OK) represents a pair of communities that are phylogenetically homogenous.
Reviewer 3 had a very similar comment (#3).

**Response:** We are using βNTI to detect stochasticity by determining if βNTI for our observed
dataset can be distinguished from the distribution of βNTI for a thousand, randomized datasets.
We find that, for early leaves and roots, βNTI from our observed data is not significantly
different from that seen with randomized data. Therefore, we cannot reject the null hypothesis
of stochasticity, which we feel allows us to infer stochasticity and take the next step of
comparing stochasticity to community size to assess drift.

We agree with the reviewer's explanations of under- and over-dispersion, but we feel that
these concepts would have been important only if we had been able to reject the null
hypothesis of stochasticity. In that case, our data could have shown over-dispersion due to
heterogeneity of community composition corrected for phylogenetic relationships or under-
dispersion due to homogeneity of community composition corrected for phylogenetic
relationships. However, because we could not reject the null hypothesis of stochasticity, we
feel that we cannot say anything about the over- or under-dispersion of our data, nor about
their homo- or heterogeneity. All that we can say that our observed data cannot be
distinguished from randomized data.

We urge reference to Dini-Andreote et al 2015 PNAS where the usage of three βNTI categories
to classify phylogenetical heterogeneity ($\beta\text{NTI} > 2$), stochasticity ($|\beta\text{NTI}| < 2$), and homogeneity
($\beta\text{NTI} < -2$) has been clearly elaborated.

Reviewer bullet point E

• $|\text{RCI}| > 0.95$ also means homogeneity, but of composition (OTUs) instead of phylogeny. For
RC_{bray} , the null distribution is created by resampling communities under the assumption of
homogeneity, i.e. under the assumption that any community is composed of the same
proportion of OTUs found in its surroundings (i.e. metacommunity), with some randomness
thrown in as a random multinomial draw of OTUs with probabilities proportional to the total
count of each OTU. Again, homogeneity, not drift.

**Response:** As with βNTI , we use RCI to detect stochasticity by determining if RCI for our
observed dataset can be distinguished from the distribution of RCI for a thousand, randomized

datasets. The difference between β NTI and RCI is that the former accounts for phylogenetic
relationships among OTUs and the latter does not. We find that, for early leaves and roots, the
community dissimilarity signal from our observed data is not significantly different from that
seen with randomized data. Therefore, we cannot reject the null hypothesis of stochasticity,
which, we feel, allows us to take the next step of comparing stochasticity to community size.

With RCI, the concepts of homogeneity and heterogeneity would have been important if we
had been able to reject the null hypothesis of stochasticity. In that case, our observed data
could have shown compositional homogeneity or heterogeneity compared the randomized
datasets. However, because we could not reject the null hypothesis of stochasticity, we feel
that we cannot say anything about the homo- or heterogeneity of our observed data. All that
we can say is that our observed data cannot be distinguished from randomized data.

Reviewer bullet point F

• Findings consistent with a null hypothesis ($|\beta$ NTI|<2 or $|RCI|<0.95$) are not evidence FOR the
null hypothesis. This is stats 101.

**Response:** We agree with the reviewer that inability to reject a null hypothesis is never
evidence for a null hypothesis. However, our use of RCI and β NTI to attempt to reject the null
hypothesis of stochasticity follows the recent, pioneering work of many researchers who use
these two approaches to address the question of stochasticity (Xun et al 2019 Nat Com; Dini-
Andreote et al 2015 PNAS; Zhou & Ning 2017 Microbiol. Mol. Biol. Rev; Stegen et al 2012, 2013
ISMEJ).

Revised text:

Please see Reviewer bullet point C, above.

Reviewer bullet point G

• The null RCI distribution may be what is expected if drift is the ONLY community assembly
process acting between a pair of samples, but homogenizing dispersal (how communities got
where they are in the first place under high mixture) happened at some point, and is a
community assembly process in and of itself. So drift doesn't explain why the two communities
look the same (why RCI and β NTI look like the null), that's because of homogenizing dispersal.
Drift is just all that's gone on AFTER that initial assembly, and the intensity of that drift is not
indicated by RCI or β NTI. I suspect this is a misinterpretation the authors have of Stegen et al.
2013.

**Responses:**

*Drift and RCI and β NTI.* Again, we agree with the reviewer that the inability to reject the null
hypothesis of stochasticity does not mean that the observed data are solely the result of

ecological drift. Based on the results of RCI and β NTI, which did not reject the null hypothesis of
stochasticity, we infer that our observed data are stochastic. We then take the extra step of
considering community size to address drift.

*Did homogenizing dispersal confound drift detection?* We do not agree with the reviewer that
homogenizing dispersal could result in a null RCI distribution. Instead, homogenizing dispersal
should result in $RCI < -0.95$ & $|\beta NTI| < 2$, which could be distinguished from RCI of a null
distribution. This result has been statistically supported by 83-91% of simulations in Stegen
2015 *Frontier in Microbiology*.

*Initial assembly.* We agree with the reviewer that drift occurs after initial assembly. Initial
assembly had to involve colonization from the homogenous soil communities and this
colonization could have been both deterministic and stochastic. In fact, we have evidence of
deterministic colonization by OTU42 on leaves and OTU17 on roots, but this deterministic
colonization is easily distinguished from stochasticity and should not confound assessment of
drift. In contrast, stochastic dispersal (which we term stochastic colonization because it occurs
on newly emerging leaves or roots) could confound the detection of drift, but colonization will
not be affected by community size whereas drift will be enhanced in small communities when
individuals are prone to extinction by chance and the probability of extinction is expected to
increase as the community size shrinks. Our correlation of high stochasticity when community
size is small provides our evidence for drift.

We agree that we cannot rule out the stochastic colonization that might occurred from seed
germination to the first week of seedling development, a time during which sampling was not
carried out in this study. Also, although we can ascribe the dominance of OTU42 and OTU17 to
deterministic selection, we cannot rule out stochastic colonization of some other OTUs.

Revised text

Please see Reviewer bullet point C, above.

Reviewer bullet point H

• The absence of spatial patterns in beta-diversity (bray) observed by the authors is just MORE
evidence of homogeneity. However, this analysis is largely superficial since individual OTUs can
exhibit spatial structure even when overall community composition shows little or no pattern.
This is especially true since bray-curtis is a weighted metric. Let's say one or two OTUs are fairly
rare, but each has a spatial pattern. e.g. an east-west abundance gradient, or patch dynamics.
That spatial structuring would be swamped out by a beta-diversity mantel style analysis. Again,
Reviewer 3 had a very similar comment (#2, although they explained very little). While distance-

dependent similarity is a necessary to claim dispersal limitation OF A SPECIES, the inverse is not
necessarily true as stated above, especially in regard to broad-scale community variability.

**Response:** Responding to the reviewer's concern of that Bray-Curtis is a weighted metric, we
re-analyzed the data using an unweighted metric, the Jaccard metric. As shown in the figure at
the end of this document, the relationships between geographic distance and Jaccard
dissimilarity were also flat in each time point in leaf, root, rhizosphere and soil.

We agree with the reviewer that our results cannot rule out the potential dispersal limitation of
one or two rare species. However, we argue that the lack of dispersal limitation in the
community as a whole is enough for us to conclude that our observed patterns are not driven
by dispersal limitation of the community as a whole.

**Revised text:**

We find no evidence for heterogeneity ($\beta\text{NTI} > 2$ or $\text{RCI} > 0.95$), whether caused by variable
selection or dispersal limitation, as expected by our finding of stochasticity and homogeneity⁴⁶.
In line with these results, we detected no substantial effect of geographic distance on fungal
community dissimilarity as shown by the flat slope of the change in dissimilarity over distance
in leaves, roots, rhizosphere, soil and air (Fig. S15-S16). Although we detect no dispersal
limitation for communities, we cannot rule out dispersal limitation of particular species,
although these would not be numerous or abundant.

Reviewer bullet point I

• Unexplained variability is not drift. The authors say they observed "high stochasticity" on line
304, and it's tempting to ascribe unexplained variability in community structure to stochastic
processes. However hidden variables are impossible to rule out in an uncontained real-world
experiment such as this sorghum field. This was mentioned by Reviewer 3 in their first
comment (#1).

**Response:** We agree with the reviewer that hidden variables are impossible to rule out in an
uncontained real-world experiment. However, our real-world experiment is as contained as
possible, with soil fungal communities that show no dissimilarity over our field and just one
crop species. As a result, our measured variables explained as much as 61.8% of total variance
in this study (Table S4), which is remarkable for an uncontained, real-world experiment.

Additionally, as indicated by Stegen et al 2015, stochasticity measured based on both $|\text{RCI}| <$
0.95 and $|\beta\text{NTI}| < 2$ is independent from selection (whether variable or homogeneous) and
dispersal (whether homogenous or from dispersal limitation). Therefore, we feel that the
stochasticity detected by $|\text{RCI}| < 0.95$ & $|\beta\text{NTI}| < 2$ is not likely to be explainable by hidden
variables.

**Revised text:**

The stochasticity indicated by $|\beta\text{NTI}| < 2$ & $|\text{RCI}| < 0.95$ is obtained when communities are not
dominated by either the two dimensions of dispersal (homogenous dispersal, dispersal
limitation) or the two dimensions of selection (variable selection, homogeneous selection)⁴⁶.
Thus, the stochasticity detected by $|\text{RCI}| < 0.95$ & $|\beta\text{NTI}| < 2$ is not likely to be explainable by
hidden variables. When the null hypothesis can be rejected, observations smaller than the null
estimations indicate under dispersion of phylogenetic ($\beta\text{NTI} < -2$) and species ($\text{RCI} < -0.95$)
composition, in which community pairs are homogenous, and observations larger than the null
estimations indicate over dispersion of phylogenetic ($\beta\text{NTI} > 2$) and species ($\text{RCI} > 0.95$)
composition, in which community pairs are heterogenous^{51, 52, 53}.

I was told not to make to many minor points, but:

Reviewer minor point a

• Taxonomic composition of fungal communities (129-134) seems superfluous.

**Response:** We agree and have moved taxonomic composition to the supplements.

Reviewer minor point b

• Calling plant compartment “space” (142) is confusing.

**Response:** We agree to use compartment and have removed all references to space.

Reviewer minor point c

• Critical typo: position vs. proportion (283).

**Response:** We thank the reviewer for catching this critical typo and have changed ‘position’ to
‘proportion’.

Reviewer minor point d

• What constitutes a “small” community size for microbes? The authors don’t seem to have an
a priori idea of how “small” a community would need to be in order to be vulnerable to drift. I
think it’s fine to discuss this relationship in a correlative framework (as in most of the paper),
but around line 309 this is not the case.

**Response:** We agree with the reviewer that we do not have a priori idea of how “small” a
community would need to be in order to be vulnerable to drift. We agree with the reviewer we
should discuss this part in a correlative framework.

**Revised text:**

Stochasticity can also be ascribed to ecological drift if it is positively correlated with richness
when community sizes are at their smallest, because the probability of extinction is expected to
increase when the total species pool is large relative to the local community size⁵⁵. Additionally,
a consequence of ecological drift is high beta diversity, which can lead to high, joint richness
(Fig. 5H). Again, we find that the strength of stochasticity is positively correlated with mean
fungal richness for the six replicates that constitute each of the 171 samples (week,
compartment, treatment) (Fig 5H).

Reviewer's final comment

Finally, my broad impression of the paper is that it has an interesting experiment, but the
authors push their results too far. I do not think the authors need to be so focused on
definitively showing "drift", especially because nobody (myself especially) will believe that they
can observe drift in a natural system. However, like Reviewer 3 wrote in their review, "I think
that it would be completely reasonable for the authors to discuss drift as one mechanism
driving the patterns observed, but this is just one possibility for their results". Again, I would
like to re-iterate that these opinions are my own, and were not informed in any way by those of
Reviewer 3. The high overlap between my comments and theirs, however, indicates that their
comments were not adequately addressed.

**Response:** We thank the reviewer for pointing out the value of our work.

We understand the belief that the complexity of natural systems hides the action of drift. We
hope that readers will abandon their beliefs and accept our data and analyses arguing that we
detect drift in leaf and root, fungal community assembly. We understand that to achieve this
goal we need to emphasize the simplicity and uniformity of our natural system, which made it
possible for us to detect drift.

We now add stochastic dispersal (or stochastic colonization in the case of newly emerging
leaves and roots) to drift as stochastic forces responsible for the assembly of fungal
communities in our sorghum system.

We revised our conclusion according to the reviewer's comment.

**Revised text:**

In early leaves and roots, we detected stochasticity ($|\beta_{NTI}| < 2$ and $|RCI| < 0.95$) characteristic
of communities lacking dominant effects of dispersal (homogenous dispersal, dispersal
limitation) or selection (variable selection, homogeneous selection)⁴⁶. This stochasticity,
negatively correlated with fungal community size, was most likely caused by ecological drift,
but we cannot rule out a contribution by the initial, stochastic aspects of dispersal, i.e.,

stochastic colonization. Our detection of ecological drift in sorghum mycobiome almost
certainly rests on our ability to capture and detect small fungal communities in early leaves and
roots by weekly sampling beginning with seedling emergence, the choice of an extremely
simple agricultural system with one species of plant and mechanical homogenization of soil,
and the recent development of statistical methods that allow us to retrieve the stochastic
component of compositional variance⁴⁶. Our detection of drift in a very simple system raises
the possibility that drift is also important in community assembly in truly natural systems.
However, in these systems, which are complex, detection of drift would be far more difficult.

**References**

- 1. Xun W, *et al.* Diversity-triggered deterministic bacterial assembly constrains community
functions. *Nat Com* **10**, 3833 (2019).
- 2. Dini-Andreote F, Stegen JC, van Elsas JD, Salles JF. Disentangling mechanisms that
mediate the balance between stochastic and deterministic processes in microbial
succession. *Proc Natl Acad Sci U S A* **112**, E1326–E1332 (2015).
- 3. Stegen JC, Lin X, Konopka AE, Fredrickson JK. Stochastic and deterministic assembly
processes in subsurface microbial communities. *ISME J* **6**, 1653 (2012).
- 4. Stegen JC, *et al.* Quantifying community assembly processes and identifying features
that impose them. *ISME J* **7**, 2069–2079 (2013).
- 5. Stegen JC, Lin XJ, Fredrickson JK, Konopka AE. Estimating and mapping ecological
processes influencing microbial community assembly. *Front Microbiol* **6**, 370 (2015).
- 6. Chase JM, Kraft NJB, Smith KG, Vellend M, Inouye BD. Using null models to disentangle
variation in community dissimilarity from variation in α -diversity. *Ecosphere* **2**, art24
(2011).
- 7. Zhou J, Ning D. Stochastic community assembly: Does it matter in microbial ecology?
*Microbiol Mol Biol R* **81**, e00002-00017 (2017).

**Figure**
 Correlation of community dissimilarity and geographic distance using an unweighted metric,
 Jaccard dissimilarity. Only two tests show significance ($P < 0.01$), Soil at week 14 and week 17,
 well after our evidence for stochasticity in leaves at weeks 1 to 7 and in roots at weeks 1 to 3.

REVIEWERS' COMMENTS:

Reviewer #4 (Remarks to the Author):

Discussing results more as stochasticity and less as drift has been a positive change to this manuscript, and has addressed many of my previous comments.

However. The authors still present drift as a primary community assembly mechanism in their experiment, but in their statistical analyses drift is a null hypothesis. In their response to my previous comments, the authors agree that the inability to reject a null hypothesis is not evidence for that null hypothesis. However, their manuscript still presents drift as a result of their analyses, when in fact it is not. My main issue with this manuscript has not been addressed. I think it's fine to describe the results of beta-NTI and RCI as 'stochasticity', but making the leap to a definitive observation of drift is not appropriate. Again, there is a clear path forward for the authors to discuss drift as a possible mechanism, but the definitive language the authors use currently is not warranted. For a glaring example in the current manuscript, see lines 486, 491-492, and 802.

Additionally, I do not think drift is the only explanation for stochasticity corresponding to community size. It's true that small communities (small meaning fewer individuals) are more vulnerable to drift. This is indisputable. But those small communities are also vulnerable to dispersal, the other stochastic community assembly process. For example, imagine jars of multicolored marbles on the floor. Some jars have hundreds of marbles, others have as few as 3 or 4. Now you randomly add a couple red marbles to each jar, simulating a stochastic dispersal of microbes across the jars. The community composition of jars that had few marbles will see the greatest variability in change (stochasticity), as a function of the random number of marbles that were added. But jars that had many marbles will all change very little, and therefore experience low variability in change. Note that this 1-species example is extremely reductive, but I'm sure the authors can see how a multi-species example would result similarly. Putting my juvenile example aside, stochastic dispersal is also extremely difficult to account for in natural systems. This would not be detectable via Mantel analysis, since we're talking about UN-limited dispersal, not dispersal limitation. Thus, I think stochastic dispersal is as-good of an explanation as drift for stochasticity in microbial community structure and its relationship to community size (as estimated by qPCR).

Really I think this manuscript would be fine if the authors discuss drift as a possible mechanism, rather than a definitive conclusion.

REVIEWERS' COMMENTS:

Reviewer #4 (Remarks to the Author):

Discussing results more as stochasticity and less as drift has been a positive change to this manuscript, and has addressed many of my previous comments.

However. The authors still present drift as a primary community assembly mechanism in their experiment, but in their statistical analyses drift is a null hypothesis. In their response to my previous comments, the authors agree that the inability to reject a null hypothesis is not evidence for that null hypothesis. However, their manuscript still presents drift as a result of their analyses, when in fact it is not. My main issue with this manuscript has not been addressed. I think it's fine to describe the results of beta-NTI and RCI as 'stochasticity', but making the leap to a definitive observation of drift is not appropriate. Again, there is a clear path forward for the authors to discuss drift as a possible mechanism, but the definitive language the authors use currently is not warranted. For a glaring example in the current manuscript, see lines 486, 491-492, and 802.

Response: We revised our manuscript thoroughly to accommodate the reviewer's suggestion that our detection of stochasticity do not definitely mean drift. In our current revision, we either replaced 'drift' with 'stochasticity' or replaced 'detection of drift' with 'likely detection of drift'.

Original texts in lines 485-486: This stochasticity, negatively correlated with fungal community size, was most likely caused by ecological drift, but we cannot rule out a contribution by the initial, stochastic aspects of dispersal, i.e., stochastic colonization. Our detection of ecological drift in sorghum...

Revised: This stochasticity, negatively correlated with fungal community size, **was likely caused** by ecological drift, but we cannot rule out a contribution by the initial, stochastic aspects of dispersal, i.e., stochastic colonization. Our **likely** detection of ecological drift in sorghum...

Original texts in lines 491-492: Our detection of drift in a very simple system raises the possibility that drift is also important in community assembly in truly natural systems.

Revised: Our **likely** detection of drift in a very simple system raises the possibility that drift is also important in community assembly in truly natural systems."

Original texts in line 802: Ecological drift and richness. Ecological drift is evidenced by a positive correlation between fungal richness and the...

Revised: **Stochasticity** and richness. Ecological drift is **suggested** by a positive correlation between fungal richness and the...

Additionally, I do not think drift is the only explanation for stochasticity corresponding to community size. It's true that small communities (small meaning fewer individuals) are more vulnerable to drift. This is indisputable. But those small communities are also vulnerable to

dispersal, the other stochastic community assembly process. For example, imagine jars of multicolored marbles on the floor. Some jars have hundreds of marbles, others have as few as 3 or 4. Now you randomly add a couple red marbles to each jar, simulating a stochastic dispersal of microbes across the jars. The community composition of jars that had few marbles will see the greatest variability in change (stochasticity), as a function of the random number of marbles that were added. But jars that had many marbles will all change very little, and therefore experience low variability in change. Note that this 1-species example is extremely reductive, but I'm sure the authors can see how a multi-species example would result similarly. Putting my juvenile example aside, stochastic dispersal is also extremely difficult to account for in natural systems. This would not be detectable via Mantel analysis, since we're talking about UN-limited dispersal, not dispersal limitation. Thus, I think stochastic dispersal is as-good of an explanation as drift for stochasticity in microbial community structure and its relationship to community size (as estimated by qPCR).

Response: we agree with the reviewer that the stochastic dispersal can be also negatively correlated with community size.

Added text: "Given the same level of stochastic dispersal, greater variability in change of species compositions is expected for smaller communities than larger communities"

Really I think this manuscript would be fine if the authors discuss drift as a possible mechanism, rather than a definitive conclusion.

Response: we thank the reviewer for pointing out the alternative explanations of our results.